

# Evaluation of five dry particle deposition parameterizations for incorporation into atmospheric transport models

Tanvir R. Khan, Judith A. Perlinger

Department of Civil and Environmental Engineering, Michigan Technological University,
Houghton, Michigan 49931, United States

*Correspondence to*: Judith A. Perlinger (jperl@mtu.edu)

**Abstract.** Despite considerable effort to develop mechanistic dry particle deposition parameterizations for atmospheric transport models, current knowledge has been inadequate to propose quantitative measures of the relative performance of available parameterizations. In this study, we evaluated the performance of five dry particle deposition parameterizations developed by Zhang et al. (2001) (*Z01*), Petroff and Zhang (2010) (*PZ10*), Kouznetsov and Sofiev (2012) (*KS12*), Zhang and He (2014) (*ZH14*), and Zhang and Sao (2014) (*ZS14*), respectively. The evaluation was performed in three dimensions: model ability to reproduce observed deposition velocities, $V_d$ (*accuracy*), the influence of imprecision in input parameter values on the modeled $V_d$ (*uncertainty*), and identification of the most influential parameter(s) (*sensitivity*). The accuracy of the modeled $V_d$ was evaluated using observations obtained from five land use categories (LUCs): grass, coniferous and deciduous forests, natural water, and ice/snow. To ascertain the uncertainty in modeled $V_d$, and quantify the influence of imprecision in key model input parameters, a Monte Carlo uncertainty analysis was performed. The Sobol' sensitivity analysis was conducted with the objective to determine the parameter ranking, from the most to the least influential. Comparing the normalized mean bias factors (indicator of accuracy), we find that the *ZH14* parameterization is the most accurate for all LUCs except for coniferous forest, for which it is second most accurate ($B_{NMBF}$ = -2.31). From Monte Carlo simulations, the estimated mean normalized uncertainties in the modeled $V_d$ obtained for seven particle sizes (ranging from 0.005 to 2.5 µm) for the five LUCs are 17%, 12%, 13%, 16%, and 27% for the *Z01*, *PZ10*, *KS12*, *ZH14*, and *ZS14* parameterizations, respectively. From the Sobol' sensitivity results, we suggest that the parameter rankings vary by particle size and LUC for a given parameterization. Overall, for $d_p$ = 0.001 to 1.0 μm, friction velocity was one of the three most influential parameters in all parameterizations. For giant particles ($d_p$ = 10 μm), relative humidity was the most influential parameter. Because it is the least complex of the five parameterizations, and it has the greatest accuracy and least uncertainty, we propose that the *ZH14* parameterization is currently superior for incorporation into atmospheric transport models.

## 1. Introduction

Of the processes involved in the atmosphere-surface exchange of gaseous species and particles, dry deposition refers to the process by which aerosols are removed from the atmosphere in the absence of precipitation through vertical turbulent transport and interaction with the earth's surface. Dry deposition is a complex process that is influenced by the chemical properties of aerosols and their sources, meteorological conditions, and surface characteristic features. The transference of particles from the atmosphere to the earth's surface is controlled by forcings such as frictional drag and terrain induced flow modification (Giorgi, 1986; Stull, 1988). Understanding the processes and factors controlling dry deposition is necessary to accurately estimate the residence time of atmospheric particles, which governs their atmospheric transport distance, trans-boundary fluxes, and potential climate effects (IPCC, 2001; Nemitz et al., 2002; Pryor et al., 2008). An accurate estimation of dry deposition is also needed to quantify the atmospheric loads of particle containing sulfate, nitrate, and ammonium that contribute to acidification and eutrophication of ecosystems, toxic elements such as Pb, Zn, and Cd, and base cations such as $Na^+$, $K^+$, $Ca^{2+}$, and $Mg^{2+}$ that alter the nutrient cycling in soil (Ruijgrok et al., 1995; Petroff et al., 2008a).



Over the last three decades, several indirect and direct methods were developed to measure dry particle deposition flux to

ecosystems (McMahon and Denisot, 1979; Sehmel, 1980; Gallagher et al., 1997; Zhang and Vet, 2006; Pryor et al., 2008). In
particle flux measurements, the timescale for dry deposition ($\tau_{dep}$) for a given measurement height $z$ is estimated as:

$$\tau_{dep}(z) = \frac{z}{V_d}, \tag{1}$$

where the dry deposition velocity $V_d$ at height $z$ is defined as the ratio of the total flux $F(z)$ divided by the particle concentration
at the same height $C(z)$ (Pryor et al., 2013; Rannik et al., 2016) and is mathematically expressed as:

$$V_d = -\frac{F(z)}{C(z)}, \tag{2}$$

One of the major limitations of direct flux measurement is limited spatial coverage because the measurement stations are
confined to only a limited number of sites (Nemitz et al., 2002). The application of spatially and temporally resolved 3-D
atmospheric transport models, from regional to global scale, can produce estimates of the dry deposition fluxes for a suite of
atmospheric species over various natural surfaces such as bare soil, grass, forest canopies, water, and ice/snow. To predict the dry

deposition fluxes using atmospheric transport models, a parameterization/scheme that can adequately account for the major
physical processes of particle deposition (e.g., turbulent diffusion, gravitational settling, interception, impaction, and Brownian
diffusion) must be embedded in the host model.

Many dry particle deposition models have been developed for scientific research and operational purposes (see model
review by Petroff et al., 2008a). In mechanistic or process-based dry deposition models, an electrical resistance based approach is

widely used to parameterize the dry particle deposition velocity (Venkatram and Pleim, 1999). In this approach, dry particle
deposition occurs via two parallel pathways: turbulent diffusion (expressed as aerodynamic resistance) and gravitational settling
(expressed as resistance due to gravitation). In addition, particle collection by surfaces via Brownian diffusion, interception, and
impaction are represented using separate surface resistance terms (Slinn, 1982; Hicks et al., 1987; Wesely and Hicks, 2000; Zhang
et al., 2001; Seinfeld and Pandis, 2006; Petroff and Zhang, 2010; Zhang and He, 2014).

Despite considerable efforts in developing dry particle deposition parameterizations of varying complexity, there remains
considerable gaps in systematic performance evaluation of existing schemes with reliable field measurements. We note that the
evaluation of dry deposition parameterizations with field measurements is very limited and not up to date. Van Aalst (1986)
evaluated the performance of six dry deposition parameterizations against field measurements, and reported large discrepancies in
terms of the modeled deposition velocities. He reported that over water surfaces the modeled deposition velocities for 1.0 µm

particles by the Williams (1982) scheme were factors of 10 to 50 higher than those predicted by the Sehmel and Hodgson (1978)
scheme. For forest canopy, the Wiman and Agren (1985) model over-predicted the deposition velocities of the Slinn (1982) model
by a factor of five. Studies also suggest that in many dry deposition parameterizations, the largest uncertainty exists for 0.1-1.0 µm
particles because of the differing treatments of some key particle deposition processes such as Brownian diffusion (Van Aalst,
1986; Petroff and Zhang, 2010; Zhang and Sao, 2014).

Uncertainty in modeled dry deposition velocities is an area that requires a thorough investigation. Only a few studies have
been conducted in quantifying the uncertainties in dry particle deposition parameterizations. Ruijgrok et al. (1992) performed an
uncertainty evaluation of the Slinn (1982) model by assessing the variabilities in nine input parameters to the model outputs. Using
Slinn's model, Gould and Davidson (1992) determined the influence of uncertainties in the size of the collection elements,
roughness length, canopy wind profile and wind speed on the modeled deposition velocities. As far as we know, a detailed

uncertainty analysis to address the influence of varying particle size, meteorological conditions, and surface features has not been
performed on existing dry deposition parameterizations. The results from an uncertainty analysis could be used as one of the



model's performance indicators, and help guide the modeling community to adequately account for uncertainties in the modeled deposition fluxes of pollutants to ecosystems.

Sensitivity analysis is often performed to determine the most influential parameters to the model outputs. Typically, a dry particle deposition model incorporates a large number of input parameters, which are subject to variability. In addition to identifying the most sensitive parameter(s), a sensitivity analysis can provide important insight as to the processes that control the overall deposition process, and identify those that may require further improvement. However, a detailed sensitivity test that encompasses exploring the entire parameter spaces of the input parameters of a dry deposition parameterization has not yet been

performed. Some researchers conducted one-at-a-time (OAT) sensitivity analysis test (SA) (Ruijgrok et al., 1997; Zhang et al., 2001) of dry deposition models. In OAT-SA, the effect of varying one model input parameter is tested at a time while keeping all others fixed (Salteli and Annoni, 2010). Because in reality the variabilities in a set of model input parameters are expected to occur simultaneously, a OAT-SA is not a useful tool to determine the most influential parameter(s) in the deposition models. Rather, a global sensitivity test approach is needed. In global sensitivity analysis, the potential effects from simultaneous variabilities of model input parameters over their plausible range is considered (Lilburne and Tarantola, 2008).

In the present study, five dry particle deposition parameterizations, developed by Zhang et al. (2001), Petroff and Zhang (2010), Kouznetsov and Sofiev (2012), Zhang and He (2014), and Zhang and Sao (2014), are selected for an intercomparison of performance in terms of accuracy, uncertainty, and sensitivity. Throughout this paper, these models are referred to as *Z01*, *PZ10*, *KS12*, *ZH14*, and *ZS14*, respectively. The objectives of this study are threefold. The first objective is to evaluate the accuracy of five dry particle deposition parameterizations using measured dry deposition velocities obtained from field observations. Data of

measured deposition velocities were collected from the literature, which comprised of measurements conducted over land use categories (LUCs) including grass, coniferous and deciduous forests, natural waters, and ice/snow. The second objective is to perform an uncertainty analysis of the modeled dry deposition velocities related to imprecision in model input parameter values. The third objective is to quantify the most influential parameters in the modeled dry deposition velocities by applying a global variance-based sensitivity analysis.

## 2. Background

The five dry particle deposition schemes used in this paper are described briefly below. For each scheme, the major expressions/equations used to compute the dry particle deposition velocities are provided.

### 2.1. The *Z01* scheme

The *Z01* scheme estimates dry particle deposition velocity as a function of particle size and density, meteorological

variables, and surface properties. In the *Z01* scheme, the dry particle deposition velocity ($V_d$) is expressed as:

$$V_d = V_g + \frac{1}{R_a + R_s} \tag{3}$$

where $V_g$ is the gravitational settling velocity, $R_a$ is the aerodynamic resistance above the canopy, and $R_s$ is the surface resistance. The expression for gravitational settling velocity ($V_g$) is given as:

$$V_g = \frac{\rho d_p^2 g C}{18 \eta_V}, \tag{4}$$

where $\rho$ is the dry density of the particle, $d_p$ is the particle aerodynamic diameter, $g$ is the gravitational acceleration, $C$ is the Cunningham correction factor, and $\eta_V$ is the temperature dependent viscosity coefficient of air. The correction factor $C$ is applied to account for the molecular structure of the air and is expressed as:

$$C = 1 + \frac{2\lambda}{d_p}\left(1.257 + 0.4 e^{-\frac{0.55 d_p}{\lambda}}\right), \tag{5}$$





where $\lambda$ is the mean free path of air molecules.

The aerodynamic resistance ($R_a$) is calculated as:

$$R_a = \frac{\ln\left(\frac{z_R}{z_0}\right) - \psi_H}{\kappa u_*},$$     (6)

where $z_R$ is the reference height where $V_d$ is typically computed, $z_0$ is the roughness height, $\kappa$ is the von Kármán constant, $u_*$ is

the friction velocity, and $\psi_H$ is the stability function for heat. The expression for $\psi_H$ is: $\psi_H = 2ln[0.5(1 + (1 - 16x)^{0.5}]$ when

$x \in [-2; 0]$, and $\psi_H = -5x$ when $x \in [0; 1]$. Here, $x = z/L_O$, where $z$ is the measurement height and $L_O$ is the Monin-Obukhov

length.

The surface resistance term, $R_s$ in Eq. 3, is a function of particle collection efficiencies due to Brownian diffusion ($E_B$),

impaction ($E_{IM}$), and interception ($E_{IN}$). Accordingly, $R_s$ is parameterized as:

$$R_s = \frac{1}{\varepsilon_0 u_* (E_B + E_{IM} + E_{IN}) R_1},$$     (7)

where $\varepsilon_0$ is an empirical constant and its value is taken as 3 for all LUCs, and $R_1$ is the correction factor for particle rebound,

which is included to modify the collection efficiencies at the surface. $R_1$ is parameterized as a function of Stokes number ($St$) as:

$$R_1 = \exp(-St^{-0.5}).$$     (8)

The parameterizations for $E_B$, $E_{IM}$, and $E_{IN}$ are expressed by Eqs. (9-11). The particle collection efficiency ($E_B$) is

parameterized as a function of Schmidt number ($Sc$) as:

$$E_B = Sc^{-\gamma},$$     (9)

where $Sc$ is the ratio of kinematic viscosity of air ($v$) to the particle Brownian diffusivity ($D$). $\gamma$ is a LUC dependent variable, and

the typical values of $\gamma$ range from 0.54 to 0.56 for rough surfaces and from 0.50 to 0.56 for smooth surfaces. Brownian diffusivity

($D$) is calculated as:

$$D = \frac{C k_B T}{3\pi\mu d_p},$$     (10)

where $C$ is the Cunningham correction factor as expressed by Eq. (5), $k_B$ is the Boltzmann's constant ($1.38 \times 10^{-23}$ J K$^{-1}$), and $\mu$ is

the dynamic viscosity of air at temperature $T$.

For smooth surfaces, particle collection efficiency by impaction ($E_{IM}$) is parameterized as:

$$E_{IM} = 10^{-\frac{3}{St}}.$$     (11)

And, for rough surfaces,

$$E_{IM} = \left(\frac{St}{\alpha + St}\right)^\beta,$$     (12)

where $\alpha$ and $\beta$ are constants; values of $\alpha$ are LUC dependent, and $\beta$ is taken as 2. In Eqs. (11-12), the Stokes number ($St$) is

expressed as:

$$St = \frac{V_g u_*}{g A} \quad \text{(for vegetative surfaces)},$$     (13)

$$St = \frac{V_g u_*^2}{v} \quad \text{(for smooth surfaces)},$$     (14)

where $A$ is the characteristic radius of the surface collector elements. The values of $A$ are given for different LUCs for various

seasons by Zhang et al. (2001).

Collection efficiency by interception ($E_{IN}$) is calculated as:

$$E_{IN} = \frac{1}{2}\left(\frac{d_p}{A}\right)^2.$$     (15)

Growth of particles under humid conditions is considered in the *Z01* scheme by replacing the $d_p$ with a wet particle

diameter ($d_w$), which is calculated as:





$$d_w = \left[ \frac{C_1 \left(\frac{d_p}{2}\right)^{C_2}}{C_3 \left(\frac{d_p}{2}\right)^{C_4} - logRH} + \left(\frac{d_p}{2}\right)^{C_3} \right]^{1/3},$$
(16)

where $C_1$, $C_2$, $C_3$, and $C_4$ are the empirical constants (values given in Table 1 of Zhang et al., 2001), and $RH$ is the relative humidity.

**2.2 The *PZ10* scheme**

Petroff and Zhang (2010) parameterized dry particle deposition velocity using an expression similar to Eq. (3) with some improvements of the surface resistance and collection efficiency terms. In the *PZ10* scheme, the effect of gravity and drift forces

(e.g., phoretic effects) were taken into account by introducing the term drift velocity $(V_{drift})$. Thus, dry deposition velocity $(V_d)$ at a reference height $(z_R)$ is given as:

$$V_d = V_{drift} + \frac{1}{R_a + R_s}.$$
(17)

Here, the drift velocity $V_{drift}$ is equal to the sum of gravitational settling velocity and phoretic velocity, and the expression of $V_{drift}$ is:

$$V_{drift} = V_g + V_{phor}.$$
(18)

$V_g$ is calculated using Eq. (4). The LUC dependent values of $V_{phor}$ were given by Petroff and Zhang (2010).

Surface resistance $(R_s)$ is commonly expressed as an inverse of the surface deposition velocity, $V_{ds}$ (i.e., $R_s = 1/V_{ds}$). In the *PZ10* scheme, $V_{ds}$ is parameterized as:

$$\frac{V_{ds}}{u_*} = E_g \frac{1 + \left[\frac{Q}{Q_g} - \frac{\alpha}{2}\right] \frac{\tanh(\eta)}{\eta}}{1 + \left[\frac{Q}{Q_g} + \alpha\right] \frac{\tanh(\eta)}{\eta}}.$$
(19)

The parameters (e.g., $Q$, $Q_g$, $\alpha$, and $\eta$) used in Eq. (19) are dependent on the aerodynamic and surface characteristic features. The parameterization of the total particle collection efficiency on the ground below the vegetation $(E_g)$ has two components: (i) collection by Brownian diffusion $(E_{gb})$ and (ii) collection by turbulent impaction $(E_{gt})$. In the *PZ10* scheme, formulation of $E_{gb}$ is expressed as:

$$E_{gb} = \frac{Sc^{-2/3}}{14.5} \left[ \frac{1}{6} ln \frac{(1+F)^2}{1-F+F^2} + \frac{1}{\sqrt{3}} Arctan \frac{2F-1}{\sqrt{3}} + \frac{\pi}{6\sqrt{3}} \right]^{-1},$$
(20)

where $F$ is a function of the Schmidt number $(Sc)$ and is expressed as $F = Sc^{\frac{1}{3}}/2.9$.

Collection efficiency by turbulent impaction, $E_{gt}$, is a function of dimensionless particle relaxation time $(\tau_{ph}^+)$ and a coefficient $C_{IT}$ (taken as 0.14). In the *PZ10* scheme, $E_{gt}$ is parameterized as:

$$E_{gt} = 2.5 \times 10^{-3} C_{IT} \tau_{ph}^{+2}.$$
(21)

$\tau_{ph}^+$ is calculated as $\tau_{ph}^+ = \tau_p u_f^2 / \nu$. The local friction velocity $(u_f)$ is expressed as:

$$u_f = u_* e^{-\alpha},$$
(22)

where $\alpha$ is the aerodynamic extinction coefficient and is expressed as:

$$\alpha = \left( \frac{k_x LAI}{12\kappa^2 \left(1 - \frac{d}{h}\right)^2} \right)^{1/3} \Phi_m^{2/3} \left( \frac{h-d}{L_O} \right).$$
(23)

In Eq. (21), $k_x$ is the inclination coefficient of canopy elements, $LAI$ is the leaf area index, $d$ is the zero-plane displacement height, $h$ is the height of the canopy, $L_O$ is the Monin-Obukhov length, and $\Phi_m$ is the non-dimensional stability function for momentum.

The expressions for $\Phi_m$ is, $\Phi_m(x) = (1 - 16x)^{-1/4}$ when $x \in [-2:0]$ and $\Phi_m(x) = (1 + 5x)^{-1/4}$ when $x \in [0:1]$.

In Eq. (19), the non-dimensional time-scale parameter, $Q$, is defined as the ratio the turbulent transport time scale to the vegetation collection time scale. The magnitude of $Q$ can be used to characterize the dominant mechanism of the vertical transport





of particles to the surface. For particle deposition over a canopy, $Q \ll 1$ describes a condition when homogeneous concentration of Aitken and accumulation mode particles prevails throughout the canopy. This condition occurs when turbulent mixing is very

efficient and transfer of particles is limited by the collection efficiency on leaves. In contrast, $Q \gg 1$ characterizes a situation in which an inhomogeneous particle concentration within the canopy prevails, which is typical for coarse mode particles. Under such conditions, efficient collection of particles by leaves takes place and transfer to the surface is usually limited by the turbulent transport.

In the *PZ10* scheme, $Q$ and $Q_g$ are parameterized using Eqs. (24) and (25), respectively:

$$Q = \frac{LAI E_T h}{l_{mp}(h)}, \tag{24}$$

$$Q_g = \frac{E_g h}{l_{mp}(h)}, \tag{25}$$

where $E_T$ is the total particle collection efficiency by various physical processes and $l_{mp}(h)$ is the mixing height for the particles. The mixing height for particles, $l_{mp}(h)$, is calculated as:

$$l_{mp}(h) = \frac{\kappa(h-d)}{\Phi_h\left(\frac{h-d}{LO}\right)}, \tag{26}$$

where $\Phi_h$ is the stability function for heat and expressed as: $\Phi_h(x) = (1 - 16x)^{-1/2}$ when $x \in [-2; 0]$ and $\Phi_h(x) = 1 + 5x$ when $x \in [0; 1]$.

The total collection efficiency ($E_T$) is expressed as:

$$E_T = \frac{U_h}{u_*}(E_B + E_{IN} + E_{IM}) + E_{IT}, \tag{27}$$

where $U_h$ is the horizontal wind speed at canopy height $h$, and $E_B$, $E_{IN}$, $E_{IM}$, and $E_{IT}$ are the collection efficiencies by Brownian

diffusion, interception, impaction, and turbulent impaction, respectively. Note that the physical meaning of the first three efficiency terms are similar to those of the *Z01* scheme. However, the parameterizations of these terms differ from the *Z01* scheme. The term describing turbulent impaction efficiency ($E_{IT}$) is absent in the *Z01* scheme.

Parameterization of deposition efficiencies (i.e., $E_B$, $E_{IN}$, $E_{IM}$, and $E_{IT}$) are given below according to the *PZ10* scheme:

Particle collection efficiency by Brownian diffusion ($E_B$):

$$E_B = C_B Sc^{-2/3} Re_h^{-1/2}. \tag{28}$$

In Eq. (28), $C_B$ is the LUC dependent coefficient, $Re_h$ is the Reynolds number of the horizontal air flow calculated at top of the canopy height $h$ as $Re_h = \frac{U_h L}{\nu}$. Here, $L$ is the LUC dependent characteristic length of the canopy obstacle elements.

Particle collection efficiency by interception ($E_{IN}$):

$$E_{IN} = C_B \frac{d_p}{L} \quad (for \; needle - like \; obstacle), \tag{29}$$

$$E_{IN} = C_B \frac{d_p}{L}\left[2 + ln\frac{4L}{d_p}\right] \quad (for \; leaf \; of \; plane \; obstacle). \tag{30}$$

In Eqs. 29-30, $C_B$ is the LUC dependent coefficient.

Particle collection efficiency by impaction ($E_{IM}$):

$$E_{IM} = C_{IM}\left(\frac{St_h}{St_h + \beta_{IM}}\right)^2. \tag{31}$$

In Eq. (31), $St_h$ is the Stokes number on top of the canopy, which is calculated as $St_h = \frac{\tau_p U_h}{L}$. $\tau_p$ is the particle relaxation time

calculated as $\tau_p = V_g/g$. $C_{IM}$ and $\beta_{IM}$ are LUC dependent coefficients.

Particle collection efficiency by turbulent impaction ($E_{IT}$) is parameterized as:

$$E_{IT} = 2.5 \times 10^{-3} C_{IT} \tau_{ph}^{+2} \qquad if \; \tau_{ph}^+ \leq 20, \tag{32}$$



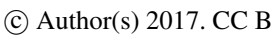

$$E_{IT} = C_{IT} \qquad\qquad if\ \tau_{ph}^{+} \geq 20, \tag{33}$$

In Eqs. (32-33), the dimensionless particle relaxation time, $\tau_{ph}^{+} = \tau_p u_*^2 / \nu$.

The term $\eta$ in Eq. (19) is taken as:

$$\eta = \sqrt{\frac{\alpha^2}{4} + Q}. \tag{34}$$

For non-vegetative surfaces, such as bare soil, natural water and ice/snow, a modified form of Eq. (17) is used in the form of Eq. (35), which is expressed as:

$$V_d = V_{drift} + \frac{1}{R_a + 1/(E_{gb} u_*)}. \tag{35}$$

**2.3. The *KS12* scheme**

Kouznetsov and Sofiev (2012) developed a dry particle deposition parameterization by extending the conventional resistance-based analogy using the exact solution of the steady-state equation for aerosol flux. According to the *KS12* scheme, for rough surfaces, dry deposition velocity ($V_d$) is computed as:

$$V_d = V_{diff} + V_{int} + V_{imp} + V_g, \tag{36}$$

where $V_{diff}$, $V_{int}$, $V_{imp}$, and $V_g$ are the velocities for the depositing particles due to Brownian diffusion, interception, impaction, and gravitational settling, respectively. The parameterizations for these terms are provided below.

$V_{diff}$ was parameterized as:

$$V_{diff} = 2u_* Re_*^{-1/2} Sc^{-2/3}, \tag{37}$$

where $Re_*$ is the canopy Reynolds number given by

$$Re_* = \frac{u_* a}{\nu}, \tag{38}$$

where $a$ is the length scale for different LUCs.

$V_{int}$ is parameterized as:

$$V_{int} = u_* Re_*^{1/2} \left(\frac{d_p}{a}\right)^2, \tag{39}$$

$V_{imp}$ is parameterized as:

$$V_{imp} = \frac{2u_*^2}{U_{top}} \eta_{imp} \left(St - \frac{u_*}{U_{top}} Re_*^{-1/2}\right), \tag{40}$$

where $U_{top}$ is the mean horizontal wind speed on top of the canopy, $\eta_{imp}$ is the particle collection efficiency due to impaction, and $St$ is the Stokes number. Kouznetsov and Sofiev (2012) used Eq. (41) to parameterize $\frac{u_*}{U_{top}}$ as:

$$\frac{u_*}{U_{top}} = min\left[(C_s + C_R LAI/2)^2, \left(\frac{u_*}{U_{top}}\right)_{max}\right], \tag{41}$$

where $C_s = 0.003$, $C_R = 0.3$, and $\left(\frac{u_*}{U_{top}}\right)_{max} = 0.3$ are constants.

The Stokes number $St$ is expressed as:

$$St = \frac{\tau_p u_*}{a}, \tag{42}$$

where $\tau_p$ is the particle relaxation time calculated as $\tau_p = V_g/g$.

The expression for $\eta_{imp}$ is given as:

$$\eta_{imp} = exp\left\{\frac{-0.1}{St_e - 0.15} - \frac{1}{\sqrt{St_e - 0.15}}\right\} \qquad if\ St_e > 0.15, \tag{43}$$

$$\eta_{imp} = 0 \qquad\qquad if\ St_e \leq 0.15, \tag{44}$$





where $St_e$ is the effective Stokes number calculated as:

$$St_e = St - Re_c^{-\frac{1}{2}}, \tag{45}$$

where $Re_c$ is the critical Reynolds number calculated as:

$$Re_c = \left(\frac{U_{top}}{u_*}\right)^2 Re_*. \tag{46}$$

The term $V_g$ in Eq. (36) is parameterized using Eq. (4).

Note that in the *KS12* scheme, the parameterization of $V_d$ over smooth surfaces requires solving the universal velocity profiles (either numerically or analytically) described by Kouznetsov and Sofiev (2012). We exclude the details of the solution procedure in this paper. We used the analytical solutions of the velocity profile obtained from the author of the *KS12* scheme through personal communication.

**2.4. The *ZH14* scheme**

Zhang and He (2014) developed an empirical resistance-based parameterization for dry particle deposition by modifying the *Z01* scheme. The overall structure of the *ZH14* scheme is similar to that of the *Z01* scheme (i.e., $V_d$ is calculated using Eq. (3)). In the *ZH14* scheme, the parameterizations of $R_a$ and $R_g$ are similar to those of the *Z01* scheme. However, in *ZH14* scheme, parametrizations for the surface resistance term $R_s$ were modified for three bulk particle sizes (i.e., PM₂.₅, PM₂.₅₋₁₀, and PM₁₀₊).

Recalling, $R_s = 1/V_{ds}$, the parameterizations of $V_{ds}$ are given below.

For particle sizes less than or equal to 2.5 µm (PM₂.₅), $V_{ds}$ is expressed as:

$$V_{ds(PM2.5)} = a_1 u_*, \tag{47}$$

where $a_1$ is an empirical constant derived by regression analysis. Values of $a_1$ are given by Zhang and He (2014) for five groups of 26 LUCs.

For particle sizes between 2.5 and 10 µm (PM₂.₅₋₁₀), $V_{ds}$ is expressed as:

$$V_{ds(PM2.5-10)} = (b_1 u_* + b_2 u_*^2 + b_3 u_*^3)e^{k1\left(\frac{LAI}{LAI_{max}}-1\right)}, \tag{48}$$

where $b_1$, $b_2$, and $b_3$ are LUC dependent constants, $LAI_{max}$ is the maximum leaf area index for a given LUC, and $k1$ is a constant, which is a function of $u_*$, and expressed as:

$$k1 = c_1 u_* + c_2 u_*^2 + c_3 u_*^3, \tag{49}$$

where $c_1$, $c_2$, and $c_3$ are the LUC dependent constants.

For particle sizes larger than 10 µm (PM₁₀₊), $V_{ds}$ is expressed as:

$$V_{ds(10+)} = (d_1 u_* + d_2 u_*^2 + d_3 u_*^3)e^{k2\left(\frac{LAI}{LAI_{max}}-1\right)}, \tag{50}$$

where $d_1$, $d_2$, and $d_3$ are the LUC dependent constants, and $LAI_{max}$ is the maximum leaf area index for a given LUC. The parameter $k2$ is a constant, which is a function of $u_*$, and is expressed as:

$$k2 = f_1 u_* + f_2 u_*^2 + f_3 u_*^3, \tag{51}$$

where $f_1$, $f_2$, and $f_3$ are the LUC dependent constants.

**2.5. The *ZS14* scheme**

Zhang and Sao (2014) used an analytical solution of the steady-state flux equation to derive an expression to compute dry deposition velocity $V_d$ as:

$$V_d = \left(R_g + \frac{R_s - R_g}{\exp\left(\frac{R_a}{R_g}\right)}\right)^{-1}, \tag{52}$$

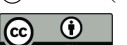



For neutral atmospheric stability conditions, the parameterizations of $R_a$ for rough and smooth surfaces are given in Eqs. (53), and (54), respectively:

$$R_a = \frac{Sc_T}{\kappa u_*} ln\left(\frac{z-d}{h_c-d}\right), \qquad (rough\ surfaces) \tag{53}$$

$$R_a = \frac{B_1 Sc_T}{\kappa u_*} ln\left(\frac{z}{z_0}\right), \qquad (smooth\ surfaces) \tag{54}$$

where $B_1$ is an empirical constant (0.45), and $Sc_T$ is the turbulent Schmidt number expressed as:

$$Sc_T = \left(1 + \frac{\alpha^2 V_g^2}{u_*^2}\right), \tag{55}$$

where $\alpha$ is a dimensionless coefficient taken as 1.

The gravitational resistance term $R_g$ is calculated as $R_g = 1/V_g$. The parameterization of the surface resistance term $R_s$ is given by Zhang and Sao (2014) as follows:

$$R_s = \left\{ R V_{dm} \left[ \frac{E}{C_d} \frac{\tau_c}{\tau} + \left(1 + \frac{\tau_c}{\tau}\right) Sc^{-1} + 10^{\frac{-3}{T_{p,\delta}^+}} \right] + V_{g,w} \right\}^{-1}, \tag{56}$$

where $R = \exp(-b\sqrt{St})$ with $b$ is an empirical constant, $E$ is the total collection efficiency, $C_d$ is the drag partition coefficient, $Sc$ is the Schmidt number, $T_{p,\delta}^+$ is the dimensionless particle relaxation time near the surface, and $V_{g,w}$ is the gravitational settling velocity of particle after humidity correction. $\frac{\tau_c}{\tau}$ is the ratio of the drag on the roof of the roughness element ($\tau_c$) to the total shear stress ($\tau$) and is calculated as:

$$\frac{\tau_c}{\tau} = \frac{\beta \lambda_e}{1 + \beta \lambda_e}, \tag{57}$$

where $\beta$ is the ratio of the pressure-drag coefficient to friction-drag coefficient, and $\lambda_e$ is the effective frontal area index. The parameter $\lambda_e$ is a function of frontal area index or roughness density ($\lambda$), and plane area index ($\eta$). The expression of $\lambda_e$ is

$$\lambda_e = \frac{\lambda}{(1-\eta)^{c_2}} exp\left(-\frac{c_1 \lambda}{(1-\eta)^{c_2}}\right), \tag{58}$$

where $c_1 = 6$ and $c_2 = 0.1$.

Eq. (57) is used to compute $T_{p,\delta}^+$ as:

$$T_{p,\delta}^+ = \frac{T_{p,\delta} u_*^2}{\nu}, \tag{59}$$

where $T_{p,\delta}$ is the particle relaxation time near the surface ($T_{p,\delta} = V_g/g$).

$V_{dm}$ is calculated using two separate expressions for rough and smooth surfaces, as expressed in Eqs. (60) and (61), respectively:

$$V_{dm} = \frac{u_*}{u_a h_c} \qquad (for\ rough\ surfaces), \tag{60}$$

where $u_a$ is the horizontal air speed and $h_c$ is the height of the roughness element.

$$V_{dm} = B_2 u_* \quad (for\ smooth\ surfaces), \tag{61}$$

where $B_2$ is an empirical constant taken as 3.

In Eq. (56), the total collection efficiency ($E$) comprised of collection efficiencies by Brownian diffusion ($E_B$), impaction ($E_{IM}$), and interception ($E_{IN}$). The parameterizations for each of these three terms are given below:

$$E_B = C_B Sc^{-2/3} Re^{n_B - 1}, \tag{62}$$

where $C_B$ and $n_B$ are empirical parameters function of flow regimes, and are given by Zhang and Sao (2014).

$$E_{IM} = \left(\frac{St}{0.6 + St}\right)^2, \tag{63}$$

where $St$ is the Stokes number and is expressed as $St = \tau_p u_* / d_c$. Here, $d_c$ is the diameter of the surface collection element. Values of $d_c$ are given by Zhang and Sao (2014) for various surfaces.



$E_{IN} = A_{in} u_* 10^{-St} \frac{2 d_{p,w}}{d_c}$,                                                                                          (64)

where $A_{in}$ is a surface dependent micro-roughness characteristic element, and $d_{p,w}$ is the wet diameter of the particle.

### 3.    Methods

#### 3.1.    An evaluation of the dry deposition parameterizations

To assess the accuracy of the five parameterizations, the modeled dry deposition velocities were compared with field
measurements from both rough and smooth surfaces. The measurement studies conducted on various natural surfaces were
collected from the literature. More specifically, the studies cited in the review article on particle flux measurements by Pryor et
al. (2008) were collected to acquire the meta-data on particle deposition. The availability of the measured and/or reported
parameters (e.g., particle size and density, air temperature, relative humidity, horizontal wind speed, friction velocity, atmospheric
stability parameter, canopy height, roughness height, zero-plane displacement height, and leaf area index) from these measurement
studies was thoroughly investigated and compiled. It was found that the many (ca. 50%) of the studies cited by Pryor et al. (2008)
did not report most of the aforementioned parameters necessary to run the parameterizations to perform a valid comparison between
the model output and measurements. To reduce uncertainty, those studies were excluded from the parametrization accuracy
evaluation. In addition, a literature search was performed in Web of Science® to find measurement studies published after 2008,
and those studies were thoroughly assessed to determine the availability of required input parameters to run the dry deposition
models. Finally, 29 measurement studies covering five land use categories (LUCs) were selected to evaluate the accuracy of the
five parameterizations. The five LUCs include grass, deciduous, and coniferous forests (rough surfaces), and natural water and
ice/snow (smooth surfaces). Table 1 summarizes information related to sampling location, latitude, longitude, elevation above
mean sea level (AMSL), sampling periods, and particle sizes reported in the measurement studies. The global spatial distribution
of these measurement studies is shown in Figure 1 according to the five LUCs.

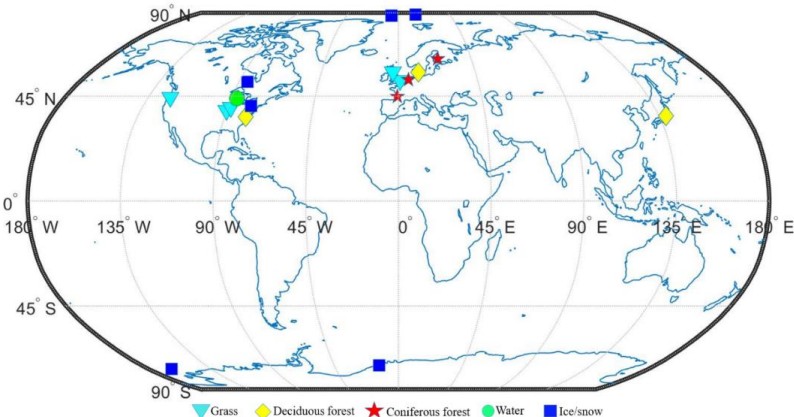


Figure 1. Global distribution of dry particle deposition measurement locations (listed in Table 1) used to evaluate
the *Z01*, *PZ10*, *KS12*, *ZH14*, and *ZS14* parameterizations. Note that for multiple measurement campaigns conducted in
one location, only one data point is shown. Two wind tunnel studies on water surfaces are not shown.

Measurements conducted over grass by Wesely et al. (1977), Neumann and den Hartog (1985), Allen et al. (1991), Nemitz
et al. (2002), and Vong et al. (2004) were used to evaluate the performance of the five parameterizations. For coniferous forest,





modeled deposition velocities were compared with measurements from Lamaud et al. (1994), Wyzers and Duyzer (1996), Gallagher et al. (1997), Ruijgrok et al. (1997), Buzorius et al. (2000), Rannik et al. (2000), Gaman et al. (2004), Pryor et al. (2007), and Grönholm et al. (2009). Experiments conducted over deciduous forest are limited, and only three studies (Wesely et al., 1983; Pryor, 2006; Matsuda et al., 2010) were used in the present paper.

To evaluate the performance of the parameterizations over water surfaces, studies by Möller and Schumann (1970), Sehmel et al. (1974), Zufall et al. (1998) and Caffrey et al. (1998) were used. We note that the studies by Moller and Schumann, and Sehmel et al. were conducted in the wind tunnels, and thus the observed deposition may not necessarily reflect deposition under natural conditions. Particle deposition measurements on ice/snow pack were collected from eight studies: Ibrahim (1983); Duan et al. (1987); Nilsson and Rannik (2001); Gronlund et al. (2002); Contini et al. (2010); Held et al. (2011a); Held et al. (2011b); and Donateo and Contini (2014). The parametrizations were fed using reported values of particle properties (diameter and density), meteorological conditions (stability parameter, temperature, wind speed, etc.), and surface properties (canopy height, roughness length, leaf area index, etc.). However, reasonable values of the missing parameters were assumed when needed.

In the present study, the accuracy of the dry deposition parameterizations was evaluated using the normalized mean bias factor ($B_{NMBF}$). The $B_{NMBF}$ provides a statistically robust and unbiased symmetric measure of the factor by which the modeled dry deposition velocities differ from the measured ones, and the sense of that factor (i.e., the positive and negative values imply the oveprediction and underprediction by models, respectively). The interpretation of the $B_{NMBF}$ is simple, and it avoids any inflation that may be caused by low values of measured quantities (Yu et al., 2006).

To quantify the disagreement between the modeled and observed quantities, the normalized mean bias factors were calculated for the pairs of modeled ($V_{d(modeled),i}$) and measured dry deposition velocities ($V_{d(measured),i}$), respectively. In this study, the expressions for computing $B_{NMBF}$ used in two different forms, which are:

For the $V_{d(modeled),i} > V_{d(measured),i}$ case (i.e., overestimation):

$$B_{NMBF} = \frac{\sum V_{d(modeled),i}}{\sum V_{d(measured),i}} - 1 = \frac{\sum V_{d(modeled),i} - \sum V_{d(measured),i}}{\sum V_{d(measured),i}}$$

$$= \sum \left[ \frac{V_{d(measured),i}}{\sum V_{d(measured),i}} \frac{(V_{d(modeled),i} - V_{d(measured),i})}{V_{d(measured),i}} \right] \tag{65}$$

For the $V_{d(modeled),i} < V_{d(measured),i}$ case (i.e., underestimation):

$$B_{NMBF} = 1 - \frac{\sum V_{d(measured),i}}{\sum V_{d(modeled),i}} = \frac{\sum V_{d(measured),i} - \sum V_{d(modeled),i}}{\sum V_{d(modeled),i}}$$

$$= \sum \left[ \frac{V_{d(modeled),i}}{\sum V_{d(modeled),i}} \frac{(V_{d(modeled),i} - V_{d(measured),i})}{V_{d(modeled),i}} \right] \tag{66}$$

The step-wise derivation of the Eqs. (65-66) and their application on training air quality datasets are illustrated by Yu et al. (2006).

### 3.2. Uncertainty analysis of the dry deposition parameterizations

To quantify the influence of imprecision in the model input parameter values on the modeled velocities, a classical Monte Carlo uncertainty analysis was applied. The Monte Carlo techniques have been widely used to evaluate the propagated uncertainty in the modeled outputs in many geophysical models (e.g., Alcamo and Bartnickj, 1987; Derwent and Hov, 1988; Chen et al., 1997;



Tatang et al., 1997; Hanna et al., 1998, 2001; Bergin et al., 1999; Bergin and Milford, 2000; Beekman et al., 2003; Mallet and Sportisse, 2006). Monte Carlo uncertainty evaluation techniques are relatively straightforward and flexible means for incorporating probabilistic values in the modeled dry deposition velocities. Indeed, the techniques are less reliant on assumptions about distributions of the input parameters (Hanna et al., 2002).

In this study, we define uncertainty in the parameterizations as the inability to confidently specify single-valued quantities because of the imprecision in the model input parameters. A classical Monte Carlo uncertainty method was applied to assess the overall uncertainty of a dry particle deposition parameterization with regard to the uncertainties in the following input parameters: $RH$, $h$, $z_0$, $d$, $LAI$, $U$, $u_*$, and $L_O$. The uncertainty estimates for those input parameters were obtained from the literature and are presented in Table 2. Using the uncertainty ranges for each of these parameters, uniform probability distribution functions were assigned since information on their actual distributions are lacking.

The Monte Carlo simulations were performed using R statistical software (version 3.2.4). Each simulation was run by randomly drawing 100 samples from the assigned uniform probability density function (PDF). The simulations were repeated 10,000 times. Frequency distributions or the PDFs of the modeled dry deposition velocity are the basic results of the Monte Carlo simulations. These PDFs were approximated assuming normal distributions, and then the 5th, 50th, and 95th percentile dry deposition velocities were computed. We use the range of the central 90% (the difference between 95[th] and 5[th] percentiles) of the PDFs as a convenient measure of uncertainty in the modeled deposition velocity. These steps were repeated for all five parameterizations using seven different particle sizes: 0.005, 0.05, 0.5, 1.0, 1.5, 2.0, and 2.5 µm on the five selected five LUCs (i.e., grass, deciduous and coniferous forests, water, and ice/snow). These particle sizes were selected to represent four distinct particle modes: nucleation (<0.01 µm), Aitken (0.01-0.1 µm), accumulation (0.1-1.0 µm), and coarse (>1.0 µm), respectively.

### 3.3. Sensitivity analysis

In this study, the Sobol' sensitivity method (Sobol' 1990) was applied to identify the most influential input parameter or the set of parameters of a dry deposition parameterization, and to characterize the relative contribution of the parameters to the overall variability in the modeled $V_d$. As opposed to the local sensitivity analysis (e.g., OAT approach), the Sobol' method is a global sensitivity approach, in which a set of input parameters of a model can be varied simultaneously over their entire parameter value space to identify their relative contributions to the overall model output variance. The Sobol' method has been applied in environmental modeling applications (Tang et al., 2007; Pappenberger et al., 2008; van Werkhoven et al., 2008; Yang, 2011), but has not yet been applied in dry particle deposition modeling research. Given that in most of the dry deposition parameterizations, model inputs can span a wide range within their physical realms, the application of a global sensitivity analysis used in this study should be viewed as a critical step toward the understanding of different sub-physical processes of particle deposition.

In the Sobol' method, the variance contributions to the total output variance of individual parameters and parameter interactions can be determined. These contributions are characterized by the ratio of the partial variance ($V_i$) to the total variance ($V$) as expressed in Eq. 67. This ratio is commonly termed as Sobol' first order index ($S_i$) (Saltelli et al., 2010; Nossent et al., 2011). The first order indices represent the fractions of the unconditional model output variance. In this study, Sobol' first order sensitivity indices were calculated as:

$$S_i = \frac{V_i}{V} = \frac{V_{X_i}(E_{X_{\sim i}}(V_d|X_i))}{V(V_d)}, \tag{67}$$

where $X_i$ is the $i$-th input parameter and $\boldsymbol{X}_{\sim i}$ denotes the matrix of all input parameters but $X_i$. The meaning of the inner expectation operator is that the mean of $V_d$ is taken over all possible values of $X_{\sim i}$ while keeping $X_i$ fixed. The outer variance is taken over all possible values of $X_i$. The variance $V(V_d)$ in the denominator is the total (unconditional) variance.



The numerator in Eq. (67) can be interpreted as follows: $V_{X_i}(E_{\mathbf{X}_{\sim i}}(V_d|X_i))$ is the expected reduction in variance that would be obtained if $X_i$ could be fixed. In regard to the variability of the model input parameters in dry deposition schemes, $S_i$ provides a means to quantity the effect of parameter $X_i$ by itself. A higher order ($S_{ij}$) or total order ($S_{Ti}$) can be computed when the total effect of a parameter, inclusive of all its interaction with other model input parameters, are of interest. In this paper, we confine the sensitivity analysis to Sobol' first order indices only.

For each of the five parameterizations evaluated here, four to nine input parameters were selected for determining the first order Sobol' sensitivity indices. An exception to applying the Sobol' method was made for the *KS12* parameterization while evaluating the parameter sensitivity for smooth surfaces. Due to the complex nature of *KS12* smooth surface parameterization, it was not computationally feasible to apply the Sobol' method. Instead, the OAT approach was applied for water and ice/snow surfaces. Note that the total number of input parameters that go into each model varies between parameterizations, and LUC types. For each parameterization, five particle sizes ($d_p$ = 0.001, 0.01, 0.1, 1.0, and 10 μm) were assessed for Sobol' analysis. The sensitivity of each parameterization was tested for the following three sets of input parameters for five LUCs: (i) particle properties, (ii) aerodynamic parameters, and (iii) surface characteristics of particle deposition. First, the sensitivity of particle deposition to particle properties (aerodynamic diameter and density) was tested. Sensitivity indices were calculated for the particle size range of 0.001 μm to 10 μm. Second, the sensitivity of the schemes was tested for aerodynamic parameters (friction velocity, wind speed, and stability condition) for different particle sizes one-at-a-time. Third, the sensitivity of the schemes to surface characteristics was tested. Surface characteristics include $h$, $z_0$, $d$, and $LAI$. The sensitivity ranges for the parameter values used for Sobol' analysis are reported in Table 3.

The Sobol 2007 package in R statistical software package (version 3.2.4) was used to perform the Sobol' sensitivity analysis. In the Sobol' method, the Monte Carlo simulations were performed by drawing samples from the assigned parameter value distribution. In this study, all the selected parameters were approximated using uniform PDFs. To assert uncertainty in the simulations, bootstrapping (Efron and Tibshirani, 1993) with re-sampling was used to achieve 95% confidence intervals on the Sobol' first order indices. For a fixed particle size, the simulations were run 100,000 times and samples were bootstrapped 1,000 times. To identify the most important parameters in each of the five dry deposition models with respect to particle size and LUC, a parameter ranking (e.g., from most to least influential) was conducted.

The results section is organized in the following manner. First, the accuracy of five dry particle deposition parameterizations (i.e., *Z01*, *PZ10*, *KS12*, *ZH14*, and *ZS14*) are compared with measured dry deposition velocities obtained from five LUCs. Second, the uncertainties in modeled dry deposition velocities due to the imprecision in the model input parameter values quantified using Monte Carlo simulation techniques are presented. Third, the sensitivity analysis results for modeled dry deposition velocities by the five parameterizations are presented.

## 4. Results

### 4.1. Evaluation of the dry particle deposition parameterizations

Field measurements conducted on five LUCs: grass, coniferous forest, deciduous forest, water surfaces, and ice/snow were used to evaluate the agreement between measured and modeled dry deposition velocity ($V_d$). The parameterizations were run using reported values of the meteorological (e.g., $U$, $u_*$, $T$, $RH$, and $L_O$) and canopy (e.g., $h$, $z_0$, $d$, and $LAI$) parameters, and particle properties (e.g., $d_p$ and $\rho$) from the measurement studies. Reasonable parameter values were assumed for any missing or unreported parameters. Normalized mean bias factors ($B_{NMBF}$) were used as an indicator of the agreement between measured and modeled $V_d$. $B_{NMBF}$ is a signed quantity-its magnitude indicates the factor by which the modeled and observed $V_d$ differ from each other, and its sign provides an indicator as to whether the modeled $V_d$ is greater or less than the measured $V_d$. It is to be noted that uncertainties



in the measured dry deposition velocities were not considered while evaluating the performance of the five parameterizations in terms of accuracy.

### 4.1.1. Evaluation of dry particle deposition to grass

Five measurement studies conducted on grass (Wesely et al., 1977; Allen et al., 1991; Neumann and den Hartog, 1985; Nemitz et al., 2002; and Vong et al., 2004) were used to evaluate the accuracy of the parameterizations. In those studies, reported values of meteorological parameters, canopy properties, and particle size vary widely. For example, the $u_*$ varies from 0.05 to 0.70 m/s, wind speed ($U$) varies from 0.67 to 6.20 m/s, particle size ($d_p$) varies from 0.05 to 2.28 µm, and $LAI$ varies from 2 to 4 m²/m². The parameterizations were fed with reported values from each of the studies to reduce any uncertainty in the accuracy comparison, however, for any missing parameter value(s), the assumed input parameter values typically fell within the aforementioned ranges.

Table 4 summarizes the $B_{NMBF}$ for modeled $V_d$ computed against five measurement studies on grass. The $B_{NMBF}$ is interpreted as follows: for example, if $B_{NMBF}$ is positive, the parameterization overestimates the measured $V_d$ by a factor of $B_{NMBF}+1$. If $B_{NMBF}$ is negative, the model underestimates the measured $V_d$ by a factor of 1- $B_{NMBF}$. For the case using the observations from Allen et al. (1991), the $B_{NMBF}$ values of -17.61, -18.12, -0.55, and -5.13 indicate that the *Z01*, *KS12*, *ZH14*, and *ZS14* parameterizations underestimated the measured $V_d$ by factors of 18.61, 19.12, 1.55, and 6.13, respectively, whereas, the

$B_{NMBF}$ value of +15.96 indicates that the *PZ10* parameterization overestimated the observations by a factor of 16.96.

These results provide means for a relative comparison of the parameterizations' accuracy. For instance, the $B_{NMBF}$ values corresponding to the Allen et al. study suggest that the *ZH14* parameterization is the most accurate and the *KS12* parameterization is least accurate. Similar comparison between the modeled and observed $V_d$ can be made using the $B_{NMBF}$ values for the remaining four studies in Table 4. Nonetheless, it is evident that none of the parameterizations performed best in terms of accuracy for all of

the five studies since the $B_{NMBF}$ values show high variability both in terms of the magnitude and direction of the bias (i.e., positive or negative) when assessed against all the five studies listed in Table 4.

The characteristics of a parameterization (e.g., *Z01*) to simultaneously over-predict (i.e., the positive $B_{NMBF}$ for Neumann and den Hartog, and Nemitz et al.) and under-predict (i.e., the negative $B_{NMBF}$ for Allen et al. 1991, Wesely et al. 1977, and Vong et al., 2004) the measurements could be misleading, resulting in erroneous judgement of the performance of the parameterizations.

To address this limitation, an ensemble approach was taken, in which $B_{NMBF}$ was calculated for each of the parameterizations using all the observations reported in the five studies. The results from this ensemble analysis indicate that, except for the *Z01* parameterization, the other four parameterizations underestimated the measured $V_d$ by factors ranging from 1.54 to 10.37. In contrast, the *Z01* parameterization overestimated the observation by a factor of 6.45 (Table 4). Overall, these results indicate that the *ZH14* parameterization provided the best agreement between the measured and modeled $V_d$ of the five parameterizations.

### 4.1.2. Evaluation of dry particle deposition to coniferous forest

Nine studies conducted on coniferous forest (Lamaud et al., 1994; Wyer and Duyzers, 1996; Gallagher et al., 1997; Ruijgrok et al., 1997; Rannik et al., 2000; Buzorious et al., 2000; Gaman et al., 2004; Pryor et al., 2007; and Grönholm et al., 2009) were used to evaluate the accuracy of the parameterizations. In these studies, the largest variations (ranges are given in the parentheses) were associated with $u_*$ (0.06-1.30 m/s), $U$ (0.60-6.19 m/s), LAI (6-10 m²/m²), and $d_p$ (0.01-0.60 µm). For any missing

parameter value(s), the assumed input parameter values typically fell within the aforementioned ranges.

Comparison of the computed $B_{NMBF}$ values for coniferous forest (Table 5) shows that the majority of the simulations performed using the five parameterizations underestimated the measured $V_d$. For example, the *PZ10* parameterization



underestimated observed $V_d$ by factors ranged from 1.51 to 27.98 ($B_{NMBF}$ values varied from -0.51 to -26.98) for eight of the nine studies on coniferous forest. Table 5 also illustrates that both the magnitude and sign of the $B_{NMBF}$ values varied widely when the accuracy of the five parameterizations was evaluated against only one study (e.g., Pryor et al., 2007). Of the $B_{NMBF}$ values associated with the Rannik et al. (2000) study, the *Z01* and *KS12* parameterizations overestimated the measured $V_d$ by factors of 4.16 and 1.51, respectively, whereas the *PZ10*, *ZH14*, and *ZS14* parameterizations underestimated the measured $V_d$ by factors of 3.54, 2.13, and 19.75, respectively. The bias factors for the *Z01* parameterization for the following studies: Lamaud et al. (1994), Gallagher et al. (1997), Buzorious et al. (2000), and Gaman et al. (2004), were +0.77, -1.74, +0.75, and -0.90, respectively. Comparing these values with the corresponding $B_{NMBF}$ values of the other four parameterizations, it can be deduced that the *Z01* parameterization is the most accurate against those observations reported in these four studies. However, the accuracy of the *Z01* parameterization is not the best for the other five studies, as can be seen from Table 5.

An ensemble approach similar to the one described in the previous section was used to determine the most and the least accurate parameterizations. From this analysis, the bias factors for the *Z01*, *PZ10*, *KS12*, *ZH14*, and *ZS14* parameterizations are -2.35, -3.93, -1.75, -2.31, and -3.67, respectively, suggesting that the *KS12* is the most accurate parameterization (i.e., under-predicted the observations by a factor of 2.75), and the *PZ10* is the least accurate parameterization (i.e., under-predicted the observations by a factor of 4.93) for coniferous forest. It can be noted that the performance of the *Z01* and *ZH14* parameterizations are nearly identical, while the *ZH14* is the second most accurate (i.e., under-predicted the observations by a factor of 3.31).

### 4.1.3. Evaluation of dry particle deposition to deciduous forest

A similar comparison between measured and modeled $V_d$ was performed using three studies (Wesely et al., 1983; Pryor, 2006; and Matsuda et al., 2012) for deciduous forest. In these studies, the largest variations (ranges are given in the parentheses) were associated with $u_*$ (0.12-1.13 m/s), $U$ (1.20-6.00 m/s), LAI (0.20-10 m$^2$/m$^2$), and $d_p$ (0.05-2.50 μm). For any missing parameter value(s), the assumed input parameter values typically fell within the aforementioned ranges.

Computed $B_{NMBF}$ values for deciduous forest are presented in Table 6. For the Wesely et al. (1983) study, comparison of the $B_{NMBF}$ values between the parameterizations show that the performance of the *ZS14* parameterization was the most accurate (i.e., $B_{NMBF}$ = -2.28; under-predicted the observations by a factor of 3.28). The $B_{NMBF}$ values associated with the *PZ10* parameterization showed strong variation between the studies (e.g., two orders of magnitude discrepancy between the Wesely et al. (1983) and Pryor (2006) or Matsuda et al. (2012) studies).

Evidently, none of the parameterizations performed consistently better for all the three studies. Overall, the results from the ensemble approach show that all the parameterizations overestimated the observations reported in three studies. Considering the $B_{NMBF}$ values obtained by this approach, it is apparent that the *ZH14* is the most accurate parameterization (i.e., $B_{NMBF}$ = -3.75, underestimated the observed $V_d$ by a factor of 4.75), and the *ZS14* is the least accurate of the five parameterizations (i.e., $B_{NMBF}$ = -10.93, underestimated the observed $V_d$ by a factor of 11.93) for deciduous forest.

### 4.1.4. Evaluation of dry particle deposition to water surfaces

Only a limited number of measurement studies on size-segregated dry particle deposition over natural water surfaces are available in the literature. In this research, four studies (Möller and Schumann, 1970; Sehmel et al., 1974; Zuffal et al., 1998; and Caffery et al., 1998) conducted over water surfaces were used to evaluate the parameterizations' accuracy. From these studies, the reported values of the parameters that show the largest variations (ranges are given in the parentheses) are $u_*$ (0.11-0.40 m/s) and $d_p$ (0.03 to 48 μm).



Table 7 shows that the *PZ10* parameterization performed best for two studies (i.e., Möller and Schumann, 1970; and Caffery et al., 1998), in which $B_{NMBF}$ values were -1.65 and +0.35, respectively. Comparison of the $B_{NMBF}$ values between the *Z01* and *ZH14* parameterizations reveal that the accuracy of the two parameterizations varied widely among the studies (e.g., $B_{NMBF}$ ranged from -0.144 to +18.87 and -0.33 to +10.28, respectively). Nevertheless, none of the five parameterizations was able to reproduce the measured $V_d$ satisfactorily for all the four studies. Comparison of the $B_{NMBF}$ values obtained by the ensemble

approach showed that the *ZH14* parameterization is the most accurate, which underestimated the measured $V_d$ by a factor of 1.25 (i.e., $B_{NMBF}$ = -0.25), and the *PZ10* is the least accurate parameterization (i.e., $B_{NMBF}$ = -0.89).

### 4.1.5.     Evaluation of dry particle deposition to snow and ice surfaces

        Two studies over snow (Ibrahim, 1983; and Duan et al., 1987), and six studies over ice surfaces (Nilsson and Rannik,

2001; Gronlund et al., 2002; Contini et al., 2010; Held et al., 2011a; Held et al., 2011b; and Donateo and Contini, 2014) were used to evaluate the accuracy of the four parameterizations (*Z01*, *PZ10*, *KS12*, and *ZH14*) for smooth surfaces. The *ZS14* parameterization was not included here because it does not allow prediction of deposition over ice/snow surfaces. The $B_{NMFB}$ values for the parameterizations are presented in Table 8.

        Of the four parameterizations, agreement between the modeled and measured $V_d$ is not satisfactory for the *PZ10* and *KS12*

parameterizations because they significantly underestimated the measured $V_d$ (e.g., the bias factors from ensemble approach are -53.03 and -21.80, respectively). In contrast, the *Z01* and *ZH14* parameterizations predicted the measured $V_d$ with reasonable accuracy (e.g., the bias factors from ensemble approach were +1.98 and +0.26, respectively). Table 8 also shows that the *ZH14* parameterization performed best for six of the eight measurements in which the $B_{NMBF}$ varied between -0.74 to 3.98. Overall, for the nine studies combined (i.e., ensemble measurements), the *ZH14* parameterization is the most accurate (overestimated the

measured $V_d$ by a factor of 1.26), and the *PZ10* is the least accurate parameterization (underestimated the measured $V_d$ by a factor of 54.03).

        To summarize, the results from the ensemble evaluation of the parameterizations is graphically shown in Figs. 2(A-B) for the five LUCs. The horizontal dotted-dashed line in the plots indicates 100% agreement between modeled and measured $V_d$, whereas any dispersion from this line either above (i.e., over-estimation) or below (i.e., under-estimation) indicates the degree of the model's

accuracy.

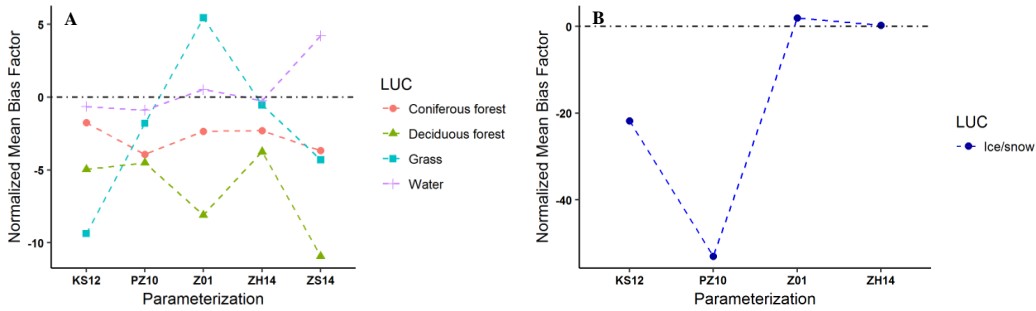

Figure 2. Ensemble averaged, normalized mean bias factors for the five parameterizations: a) three rough surfaces and water, b) Ice/snow.

**4.2. Uncertainty analysis results from the Monte Carlo simulations**

        The overall uncertainty in the modeled $V_d$ due to imprecision in the model inputs was assessed by performing a set of Monte Carlo simulations on the five dry particle deposition parameterizations.  Uncertainties (in terms of imprecision) in the





following model input parameters: *RH*, *U*, $u_*$, $L_O$, *h*, $z_0$, *d*, and *LAI* were approximated using uniform distributions. Note that not all of the five parameterizations require an identical number of input parameters. For example, Monte Carlo simulations performed
on rough surfaces (i.e., grass, coniferous, and deciduous forests) for the *Z01*, *PZ10*, *KS12*, *ZH14*, and *ZS14* parameterizations, imprecision in four (*RH*, *L*, $u_*$, and $z_0$), eight (*RH*, *L*, $u_*$, *U*, $z_0$, *h*, *d*, and *LAI*), four (*RH*, $u_*$, *U*, and *LAI*), four (*RH*, *L*, $u_*$, and $z_0$), and two (*RH*, $u_*$) input parameters, respectively, were assessed to evaluate the overall uncertainty in modeled dry deposition velocities.

The results from the Monte Carlo simulations are summarized in Table 9 and are presented and discussed in two steps.
First, the uncertainty estimates that are shown in Table 9 for five parameterizations on five LUCs are used to elucidate the models' precision, which is one of the indicators of overall performance of the parametrization (Sections 3.2.1-3.2.5). Second, the size-dependent uncertainty ranges (i.e., the difference between the 95[th] and 5[th] percentiles) was divided by the 50[th] percentile $V_d$, which can be treated as a normalized measure of uncertainty. This approach was taken to make reasonable comparison between different particle sizes for different parameterizations (Section 3.2.6). Note that the *ZS14* parameterization does not treat different vegetative
covers separately; therefore, inter-comparison of the Monte Carlo simulation results is confined to the first four parameterizations listed in Table 9.

### 4.2.1. Uncertainties in the modeled $V_d$ for grass

The uncertainties in simulated $V_d$ (i.e., differences between 95[th] and 5[th] percentiles of distribution) for the given range of $d_p$ (i.e., 0.005-2.5 µm) on grass varied widely (Table 9). In the *Z01* parameterization, the estimated uncertainty for nucleation mode
particles (0.0038 m s[-1] for $d_p$ = 0.005 µm) was larger than that of coarse mode particles (0.0001 m s[-1] for $d_p$ > 1.0 µm). Overall, in the *Z01* parameterization, the trend was that as the particle size increased from 0.005 to 2.5 µm, uncertainties in modeled $V_d$ decreased considerably. In the *PZ10* parameterization, the range of uncertainty for the simulated particle sizes is narrower as compared to those of the *Z01* parameterization. Although not consistent, a decreasing trend in uncertainties can be seen for all the particle sizes in the *PZ10* parameterization. Of the simulated particle sizes, the uncertainty for $d_p$ = 0.005 µm is the largest (0.0016
m s[-1]) in the *KS12* parameterization. As particle size increased from 0.005 to 2.5 µm, significant decrease in uncertainties is observed. For $d_p$ = 0.05 to 1.5 µm, the 5[th] and 95[th] percentile $V_d$ were nearly identical (Table 9), suggesting that the *KS12* parameterization is the most precise of five parameterizations specifically for those particle sizes. From Table 9, it can be deduced that the uncertainties associated with the *ZH14* parameterization, which is an improved and simplified version of the *Z01* parameterization, were fairly constant (ca. 00003 m s[-1]) for the seven particle sizes simulated here for grass.

### 4.2.2. Uncertainties in the modeled $V_d$ for coniferous forest

For nucleation mode particles (i.e., $d_p$ = 0.005 µm), the largest uncertainty (0.0036 m s[-1], median $V_d$ = 0.0180 m s[-1]) was associated with the *Z01* parameterization (Table 9). Overall, the uncertainties in the *Z01* parameterization showed a decreasing trend as the particle size increased from 0.005 to 2.5 µm. We note that in the *PZ10* parameterization, the relative magnitude of the uncertainties associated with 0.005, 1.0, 1.5, 2.0, and 2.5 µm particles were of the same order (i.e., varied between 0.0010 to 0.0031
m s[-1]). In comparison, uncertainties in modeled $V_d$ for 0.05 and 0.5 µm particles were smaller by factors of ca. 10. In the *KS12* parameterization, the largest uncertainty was found for the nucleation mode particles (i.e., 0.0027 m s[-1]; median $V_d$ = 0.0299 m s[-1]), and the uncertainties in modeled $V_d$ decreased substantially as $d_p$ increased. The uncertainties in modeled $V_d$ in the *ZH14* parameterization were constant (0.0002 m s[-1]) for all seven particle sizes indicating the model's ability to reproduce dry deposition velocities with high precision.

### 4.2.3. Uncertainties in the modeled $V_d$ for deciduous forest

A similar comparison of the uncertainties in modeled $V_d$ can be made for deciduous forest. It is seen from Table 9 that, for all the parameterizations except for *ZH14*, the largest uncertainties were associated with nucleation mode particles. That is, *Z01*





and *KS12* parameterizations showed substantially greater uncertainties for $d_p$ = 0.005 µm (0.0030 and 0.0027 m s$^{-1}$, respectively) as compared to the Aitken or coarse mode particles, for which the relative magnitude of the uncertainties were smaller by factors of ca. 13-30. In the *KS12* parameterization, the identical values of the 5$^{th}$ and 95$^{th}$ percentile $V_d$ resulted in uncertainty values of zero for each simulated particle size of 0.5 to 2.0 µm, which indicates that it is the most precise of all four parameterizations. In addition, the uncertainties in the modeled $V_d$ in the *ZH14* parameterization were constant (0.0004 m s$^{-1}$) for all seven particle sizes.

### 4.2.4. Uncertainties in the modeled $V_d$ for water surfaces

For water surfaces, the uncertainties in modeled $V_d$ varied largely for the *Z01* parameterization (Table 9). That is, the largest uncertainty (0.0021 m s$^{-1}$) was associated with $d_p$ = 0.005 µm (median $V_d$ = 0.0099 m s$^{-1}$), and as $d_p$ increased to 2.5 µm, the uncertainty decreased to 0.0001 m s$^{-1}$ (for 2.5 µm particles, median $V_d$ = 0.0009 m s$^{-1}$). Relatively narrower ranges in the uncertainties in modeled $V_d$ for the *PZ10* and *KS12*, and constant uncertainties in the *ZH14* parameterizations with regard to changes in particle size suggest their higher precision as compared to the *Z01* parameterization under similar model input parameter uncertainties. Overall, as compared to the simulated uncertainties in the modeled $V_d$ by the *Z01*, *PZ10*, *KS12*, and *ZH14* parameterizations, uncertainties in the *ZS14* parameterization are larger for $d_p$ = 0.05 to 2.5 µm.

### 4.2.5. Uncertainties in the modeled $V_d$ for ice/snow surfaces

Comparison between the simulated uncertainties in modeled $V_d$ revealed that the uncertainties vary significantly for the *Z01* and *KS12* parameterizations as $d_p$ changes. For example, uncertainties estimated from Table 9 for these two parametrizations decreased from 0.0023 to 0.0003 m s$^{-1}$ and 0.0027 to 0.0008 m s$^{-1}$, respectively, as particle size increased from 0.005 to 2.5 µm. Note that the median $V_d$ by the *PZ10* parameterization is an order of magnitude lower than that of other three parameterizations, which results in close to zero uncertainties for all seven particle sizes. Also revealed in Table 9, the uncertainties in the *ZH14* parameterization are constant (0.0002 m s$^{-1}$) with regard to changes in the particle size.

### 4.2.6. Normalized uncertainties in the modeled $V_d$

An extended analysis of the results presented in the previous sections are summarized here. The normalized uncertainties presented in the Table 10 can be interpreted as follows: any value that is closer to zero indicates higher model precision (i.e., less uncertainty). As shown in Table 10, the normalized uncertainties for grass and $d_p$ = 0.005 µm associated with the *Z01*, *PZ10*, *KS12*, *ZH14*, and *ZS14* parameterizations are 0.20, 0.11, 0.09, 0.20, and 0.20, respectively. These results suggest that *KS12* is the least uncertain (i.e., most precise) parameterization for nucleation mode particles, whereas, the *Z01*, *ZH14*, and *ZS14* are the most uncertain (i.e., least precise) parameterizations. Similar comparisons can be made for other particle sizes, as well as between the different LUCs. For example, the uncertainties associated with $d_p$ = 0.05 µm is greater for the *PZ10* parameterization for deciduous forest as compared to grass (0.20 > 0.13).

Comparison of the normalized uncertainties in modeled $V_d$ over smooth surfaces (i.e., water and ice/snow) also reveals interesting findings. For example, for $d_p$ = 0.5 µm, the normalized uncertainties over water surfaces for the *Z01*, *PZ10*, *KS12*, and *ZH14* parameterizations are 0.20, 0.00, 0.50, and 0.17, respectively. These results suggest that the *PZ10* parameterization is the least uncertain (i.e., most precise), whereas, the *KS12* is the most uncertain (i.e., least precise) parameterization for accumulation mode particles. Over ice/snow surfaces, with $d_p$ = 0.005 µm, both the *Z01* and *ZH14* parameterizations have large uncertainties (normalized uncertainties are 0.18 and 0.17). In contrast, *PZ10* is the most precise parameterization with close to zero normalized uncertainty value.

The normalized uncertainties presented in Table 10 also reveal interesting findings about the relative magnitude of imprecision for a given particle size on various LUCs by one parameterization. For example, with $d_p$ = 0.005 µm, the range in normalized uncertainties varies from 0.18-0.20 and 0.09-0.20 for all the five LUCs for the *Z01* and *KS12* parameterizations, respectively.



Figs. 3(A-E) show the relative comparison between uncertainties in modeled $V_d$ by five parameterization for seven particle sizes across five LUCs. For LUC grass, Fig. 3A shows that in the uncertainties in the *Z01* and *ZH14* parameterizations show nearly

identical trends, which are relatively narrow. That is, the uncertainties for particle sizes from 0.005 to 2.5 µm varied from 12-22% and 11-20% in the *Z01* and *ZH14* parameterizations, respectively. In contrast, uncertainties in the *PZ10* and *KS12* parameterizations exhibit large dispersion (i.e., uncertainty ranges from ~0-13% in the *PZ10*, and ~0-14% in the *KS12* parameterizations). The largest uncertainties in the simulated $V_d$ are associated with the *ZS14* parameterization, in which the range of uncertainty varied from ~0-34% for the seven particle sizes. We note that the minimum $V_d$ produced by the *KS12* parameterization is at $d_p$ = 0.5 µm for grass,

coniferous and deciduous forest, and ice/snow surfaces, which can be confirmed from the Fig. 3(A-C and E). In addition, Fig. 3(D-E) show that the position of this minimum $V_d$ in the *PZ10* parameterization ranged from $d_p$ = 0.5-1.0 µm for water and ice/snow surfaces.

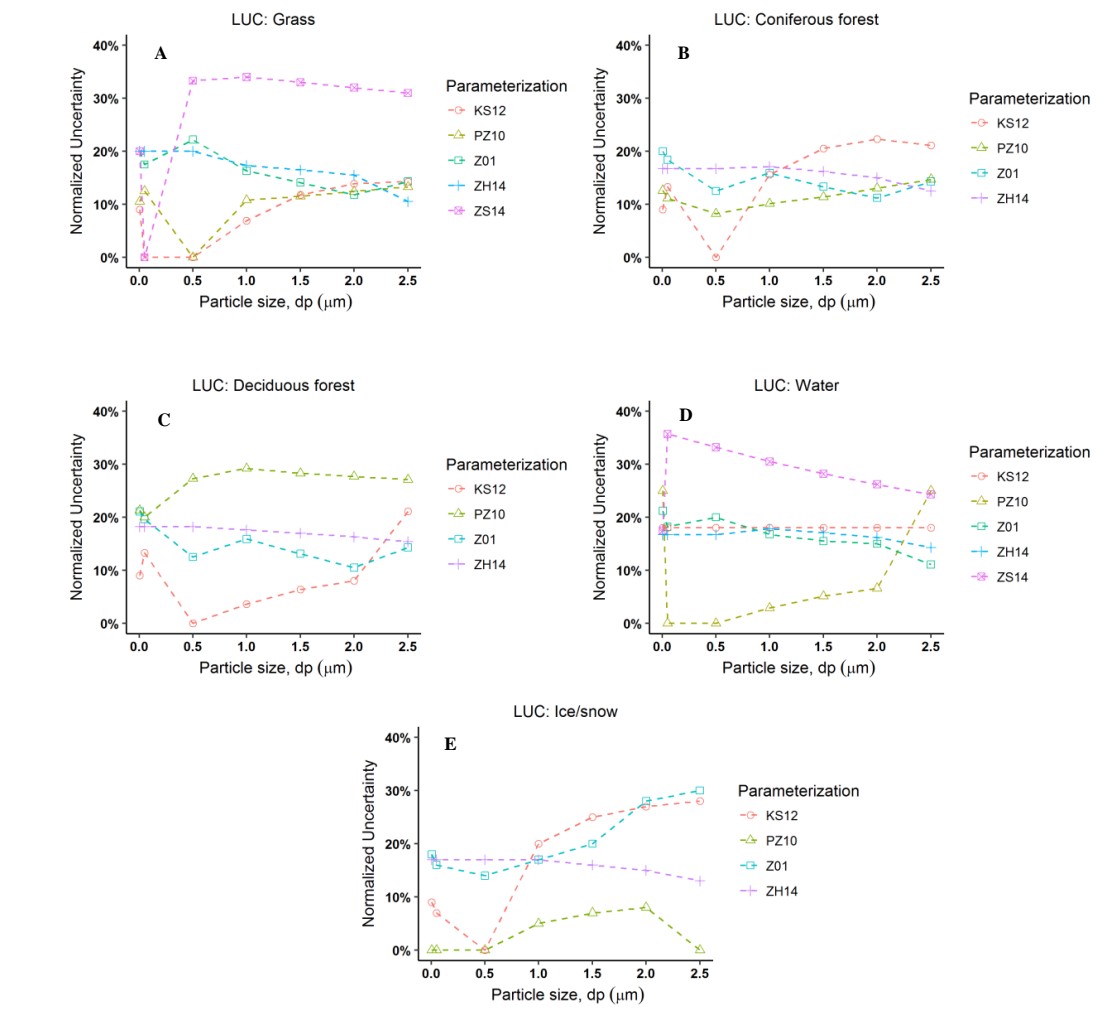

Figure 3. Comparison of the simulated uncertainties in the modeled dry deposition velocities as a function of particle size in five parameterizations for five LUCs.

Similar comparison can be made to evaluate the relative magnitude of the uncertainties in the modeled $V_d$ predicted by the parameterizations over other LUCs from Figs. 3(B-E). In Fig. 3B, uncertainties in modeled $V_d$ for coniferous forest ranged





from 11-20%, 8-15%, ~0-21%, and 13-17%, in the *Z01*, *PZ10*, *KS12*, and *ZH14* parameterizations, respectively. In Fig. 3C, uncertainties in modeled $V_d$ for deciduous forest ranged from 10-21%, 20-29%, ~0-21%, and 15-18% in the *Z01*, *PZ10*, *KS12*, and *ZH14* parameterizations, respectively. In Fig. 3D, uncertainties in modeled $V_d$ for water surfaces ranged from 11-21%, ~0-

25%, 18%, 14-18%, and 18-36% in the *Z01*, *PZ10*, *KS12*, *ZH14*, and *ZS14* parameterizations, respectively. In Fig. 3E, uncertainties in modeled $V_d$ for ice/snow surfaces ranged from 14-30%, ~0-8%, ~0-28%, and 13-17% in the *Z01*, *PZ10*, *KS12*, and *ZH14* parameterizations, respectively.

**4.3. Sensitivity analysis results: Sobol' first order sensitivity index**

For Sobol' first order sensitivity analysis, five particle sizes (i.e., $d_p$ = 0.001, 0.01, 0.1, 1.0, and 10 µm) were selected. A

sample size (*n*) of 100,000 was used for model evaluations for each of the five particle sizes. To assess the confidence intervals for the first order Sobol' sensitivity index, bootstrapping resampling was used. In the bootstrapping method, the *n* samples used for the sensitivity simulations were sampled 1,000 times with replacement. In the following sections, the results from the Sobol' sensitivity analysis, and evolution of the parameter rankings are presented.

The Sobol' sensitivity analysis performed here is used to achieve a ranking of the model input parameters. The ranking

of the parameters from most to least sensitive of the five particle sizes for the five parameterizations is shown in Table 11. Tables S1-S5 show the first order Sobol' indices of the various input parameters used in five dry particle deposition parameterizations for five LUCs. In these tables, particle size-dependent first order Sobol' index ($S_i$) for different model input parameters are presented with 95% confidence intervals (*CI*) obtained by bootstrap sampling. For example, the results of the first order Sobol' indices for the *Z01* parameterization on five LUCs are presented in Table S1. It is important to note that the number of parameters tested for

Sobol' analysis varied between different LUCs, mainly because the number of parameters required for modeling $V_d$ for one LUC may be more or less as compared to another LUC.

As shown in Table S1, for the *Z01* parameterization on grass, the importance of the most influential parameters on the modeled dry deposition velocities for five particles sizes can be compared using the corresponding $S_i$ values of the model input parameters (e.g., $i = RH, \rho, L_O, u_*$, etc.). For example, with $d_p$ = 0.001 µm, it can be clearly seen that the $u_*$ is by far the single most

sensitive parameter with an $S_i$ value of 0.918, which indicates that 91.8% of the variation in the modeled $V_d$ can be attributed to variations in $u_*$. The other parameters that have significant effect on the modeled $V_d$ are $z_0$ and $L_O$. These two parameters have $S_i$ values of 0.044 and 0.009, respectively. As compared to the first order Sobol' value of $u_*$, these values are significantly smaller; however, the lower limits of the corresponding 95% C.I. intervals for $z_0$ and $L_O$ are greater than zero, indicating that they have a significant effect on the modeled velocities. The $S_i$ values for the other two parameters, $RH$ and particle $\rho$, were approximately

zero for $d_p$ = 0.001 µm (Table S1), and indicate that these variables have no influence on the modeled $V_d$.

Comparison between the first order Sobol' indices for different particle sizes for grass shows strong variations for certain input parameters, which reveals interesting findings about the relative importance (from the most to the least) of the model input parameters to the modeled dry $V_d$. For example, as seen from Table S1, as $d_p$ increases from 0.001 to 10 µm, $S_i$ values of $u_*$ decrease from 0.918 to 0.245, which indicates that deposition of coarse particles is not strongly influenced by variations in friction velocity.

From Table S1 it is also seen that parameters that influence particle properties (i.e., $RH$ and $\rho_P$) have higher $S_i$ values for the coarse particles as compared to the fine or accumulation mode particles. Similar comparisons between size-dependent behavior of parameter sensitivity for other rough surfaces (i.e., coniferous and deciduous forests) can be made using the $S_i$ values reported in Table S1.





The results of the first order Sobol' indices for the *Z01* parameterization on two smooth surfaces: water and ice/snow are also presented in Table S1. Over liquid water surfaces, variation in $u_*$ values has the largest influence modeled $V_d$ for $d_p = 0.001$ to 10.0 µm. As is seen from Table S1, the $S_i$ values of $u_*$ can alone explain 98.3-99.5% of the variations in modeled $V_d$ for particle sizes up to fine mode (i.e., 0.001-0.01 µm). For coarse mode particles (e.g., $d_p = 10$ µm), $u_*$ is also the most influential parameter, contributing ca. 56% of the total variation in modeled $V_d$, while relative humidity is the second most influential/sensitive parameter with an $S_i$ value of 0.393. The influence of $u_*$ also tends to dominate the modeled $V_d$ over ice/snow surfaces. This theory can be confirmed by comparing the size-dependent $S_i$ values of $u_*$ shown in Table S1, which suggest that $u_*$ is the single most sensitive parameter ($S_i = 0.978$) for $d_p = 1.0$ µm. As the particle size increased to 10 µm, *RH* and $u_*$ can explain 92.7% of the total variation in modeled $V_d$ in the *Z01* parameterization.

The results of the first order Sobol' indices for the *PZ10* parameterization on five LUCs are presented in Table S2. The size-dependent $S_i$ values on coniferous forest can be compared here to elucidate the contribution of different input parameters on the modeled $V_d$. It can be noted that, on rough surfaces, the *PZ10* parameterization was tested for the most number of input parameters (i.e., nine) among the five parameterizations. Some canopy properties such as *h*, *d*, *LAI*, and meteorological properties such as *U* were tested for their influence on modeled $V_d$ in addition to those parameters that were tested for the rough surfaces in the *Z01* parameterization. As seen from Table S2, for coniferous forest, for $d_p = 0.001$ µm, $u_*$ and $L_O$ are the two most influential parameters ($S_i$ values of 0.492 and 0.462, respectively). Although *LAI* is not the most influential parameter for the range of $d_p$ tested here, its influence on the overall variability in the modeled $V_d$ increase from 0.5 to 31.3% as particle size increases from 0.001 to 0.1 µm. Similarly, wind speed tends to show an increasing influence as $d_p$ increases from 0.001 to 1.0 µm (overall contribution of *U* in the variability in $V_d$ shows an increase from 0.1% to 27.7%). For coarse particles (i.e., $d_p = 10$ um), $u_*$ and $L_O$ are the two most influential parameters with $S_i$ values of 0.372 and 0.350, respectively. Together with *RH*, the three parameters can explain 92% of the variation in the modeled $V_d$. Results from the first order Sobol' indices for the other LUCs for the *PZ10* parameterization presented in Table S2 can be explained in a manner similar to that used to explain the contribution of the most sensitive parameters to the modeled dry deposition velocities. For the water surface, $u_*$ is the most influential parameter for $d_p = 0.001$ µm as 99.4% of the total variance on the modeled $V_d$ is attributed to its variability. Indeed, for particle sizes up to 0.1 µm, the $u_*$ itself is most sensitive parameter. As seen from Table 11, *RH* becomes the most influential model parameter for $d_p = 1.0$ and 10.0 µm, which alone contribute to 69.5% and 95.6% of the total variabilities in the modeled $V_d$, respectively.

Table S3 shows the first order Sobol' indices for the *KS12* parameterization on five LUCs. For brevity, the results of the first order sensitivity indices for deciduous forest are discussed herein. It is seen that $u_*$ is the single most influential parameter for $d_p = 0.001$ to 0.1 µm (e.g., total contribution on the modeled $V_d$ attributable to $u_*$ ranges from 94.4 to 96.7%). For $d_p = 1.0$ and 10 µm, *RH* is the most influential parameter with $S_i$ values of 0.629 and 0.934, respectively.

Table S4 shows the first order Sobol' indices for the *ZH14* parameterization on five LUCs. The results show a strong influence of $u_*$ on the modeled $V_d$. As shown in Table S4, the $S_i$ values alone can explain nearly 100% of the variation in the modeled $V_d$ for $d_p = 0.001$ to 1.0 µm. For large particles (e.g., $d_p = 10$ µm), *RH* is the most influential parameter, however, the contributions of other parameters as listed in Table S4 vary with regard to changes in LUCs.

## 5. Discussion

The accuracy of the parameterizations should be interpreted within the context of the field measurements used in this study assuming that they were accurate. In addition, the inter-comparison of the parameterizations' accuracy is subject to uncertainties with regard to the assumed values of missing meteorological parameters, particle properties, or surface features.



Evidently, the normalized mean bias factors obtained using the ensemble approach is a useful measure to inter-compare the parameterizations' performance against a sub-set of field measurements for a given LUC. Extending the comparison of the normalized mean bias factors across the five LUCs for the five parameterizations investigated in this study provides a relative

assessment of their accuracy. However, the *ZH14* parameterization is most accurate for all parameterizations except coniferous forest, where it is a close second to the *KS12* parameterization.

For rough surfaces, our results suggest that *ZH14* is the most accurate parameterization for grass and deciduous forest, and it is the second most accurate parameterization for coniferous forest. In contrast, *KS12*, *PZ10*, and *ZS14* are the least accurate parameterizations for grass, coniferous, and deciduous forests, respectively. It is interesting that in most cases the models under-

predicted the measured dry deposition velocities (negative bias factors in Tables 4-8). Indeed, for grass, except for the *Z01* parameterization, the other four parameterizations under-predicted the measured $V_d$ by factors of 1.54 to 10.37 ($B_{NMBF}$ varied from -0.54 to -9.37). With regard to deciduous and coniferous forests, all of the five models (from the most to the least accurate: *ZH14*, *PZ10*, *KS12*, *Z01*, and *ZS14; KS12*, *ZH14*, *Z01*, *ZS14*, and *PZ10*) under-predicted the measured $V_d$ by factors of 4.75 to 11.93, and 2.75 to 4.93, respectively.

A direct quantitative comparison of the accuracy of the five parameterizations with those reported in other studies is impossible because the metric used in the present study ($B_{NMBF}$) is not commonly used to evaluate the accuracy of the dry deposition models. However, qualitatively, our findings regarding the *PZ10* performance for coniferous forests are in accordance with those reported by Petroff and Zhang (2010). They reported that the *PZ10* parameterization under-predicted the measured deposition velocities for the following subset of observations that we also used for coniferous forest: Lamaud et al. (1994), Gallagher et al.

(1997), Buzorious et al. (2000), Gaman et al. (2004), and over-predicted for Grönholm et al. (2009).

The accuracy results over smooth surfaces suggest that, for the water surface, the best agreement between the measured and modeled $V_d$ was found for the *ZH14* parameterization. Overall, the accuracy ranking from best to worst is as follows: *ZH14*, *Z01*, *KS12*, *PZ10*, and *ZS14*. Over ice/snow surface, the results suggest that the *ZH14* is the most accurate parameterization, and *PZ10* is the least accurate. Qualitatively, this finding is consistent with Petroff and Zhang (2010), who reported that their model

significantly underestimated the measured deposition velocities over ice/snow surface for the following studies: Ibrahim (1983), Duan et al. (1987), Nilsson and Rannik (2001), and Contini et al. (2010), which were also used in the present study. We also note that the *Z01* parameterization overestimated the measured $V_d$ from the aforementioned studies. This finding is consistent with Petrofff and Zhang (2010), as they compared their model with *Z01* over the ice/snow surface. One possible explanation for a large discrepancy between modeled and measured $V_d$ by *PZ10* is an incorrect magnitude of the drift velocity applied, corresponding to

phoretic effects on ice and snow.

Collectively for both rough and smooth surfaces, it is found that the *ZH14* scheme is the most accurate for these LUCs: grass, deciduous forest, water, and ice/snow surfaces. *KS12* performed slightly better for coniferous forest only. The performance of the *PZ10* scheme could be viewed as moderate. This finding is interesting considering that the *ZH14* is the simplest resistance-based scheme of the five parameterizations. We emphasize that *Z01* and *ZH14* parameterizations share similar structural features, but simplifications of the particle collection processes by constant values by *ZH14* (see Eqs. (47-51)) could produce better

agreement. In addition, we note that the *KS12* parametrization is exclusively based on wind tunnel measurements and its performance over forest canopies is not satisfactory, as reported by the model developers Kouznetsov and Sofiev (2012). However, we find that *KS12* performed the best over coniferous forests with the nine studies used in this research. However, Kouznetsov and Sofiev (2012) did not use the same subset of studies to evaluate the model performance as we used.

Given the complex nature and incomplete knowledge of the dry particle deposition process, it is of importance to account for the uncertainties in the modeled deposition velocities in atmospheric transport models (Petroff and Zhang, 2010; Zhang et al.,



2012). Although there have been many dry deposition models developed over the years, the information on the model output uncertainties is meager and not up-to-date. To assert uncertainty on the modeled dry deposition velocities, Gould and Davidson (1992) adopted a step-wise uncertainty test of Slinn's (1982) model. However, in reality, the model parameters are subject to
simultaneous variability, and a OAT test cannot adequately propagate the error to the overall model outputs. This limitation was partially overcome by Ruijgrok (1992), who performed a probabilistic uncertainty test of Slinn's model.

The Monte Carlo uncertainty analysis performed in this study assumes that in the five parameterizations all the major physical processes (e.g., turbulent diffusion, Brownian diffusion, impaction, interception, and gravitational settling) of dry particle deposition are accounted for satisfactorily. Thus, the uncertainty analysis conforms to the uncertainties in the model input variables
and their overall contribution to the propagated uncertainties in the modeled dry deposition velocities. Additional uncertainties in the modeled deposition velocities could arise from inadequate model formulation. However, addressing this issue of a model's structural uncertainty was outside the scope of this paper.

The values of the eight model parameters, covering four meteorological ($U$, $u_*$, $L_O$, and $RH$) and four canopy morphological ($z_0$, $d$, $h$, and $LAI$) properties, used in the Monte Carlo simulations were assumed to be uniformly distributed because
their true distributions were unknown. It is emphasized that these parameters are not all necessarily independent; $z_0$ and $d$ are functions of the surface characteristics (Zhang and Sao, 2014; Shao and Yang, 2005, 2008). Considering these underlying assumptions, the uncertainties in modeled $V_d$ reported in this paper should be viewed as the effect of the chosen parameter PDFs on the output uncertainty. The uncertainty bounds (i.e., the central 90% values) reported in the Table 9 could be treated as a metric of the quality of the modeled outputs. The normalized uncertainties reported in this study are a useful indicator to assess the overall
performance of a model for four particle modes (seven particle sizes) across five LUCs.

We applied Sobol' sensitivity analysis to identify the most influential parameter(s) of the five parameterizations. Parameter rankings achieved using the Sobol' first order indices for different models provide a robust evaluation of the models' sensitivity by varying a set of input parameters within their plausible ranges. It is emphasized that a local sensitivity analysis such as OAT could lead to incomplete or misleading inference of the parameter sensitivity on the model's output because assumptions
of model linearity are not always justified for dry deposition parameterizations due to their complex formulations.

The Sobol' sensitivity rankings presented in Table 11 can be used for inter-comparison between models' parameter sensitivity. Over rough surfaces, for nucleation size particles (e.g., $d_p = 0.001$ µm), $u_*$ is the most sensitive parameter for $Z01$, $PZ10$, $KS12$, and $ZH14$ parameterizations. As particle size increases from 0.001 µm to 1.0 µm, except for the $PZ10$ scheme and for 1.0 µm for grass in $KS12$ scheme, $u_*$ remains the most influential parameter. This finding is in accordance with previous studies
(Zhang et al., 2001; Zhang and He, 2014) that show that dry particle deposition velocities for atmospheric particles are greatly influenced by friction velocity. We note that in the $PZ10$ scheme, $LAI$ and $L_O$ are the two most commonly-found sensitive parameters for $d_p = 0.001$ to 1.0 µm for rough surfaces. As seen from the parameter rankings (Table 11), for $d_p = 10$ µm in the $Z01$, $PZ10$, $KS10$, $ZH14$ schemes, $RH$ is the most influential factor. We postulate that with particle growth, high humidity may have a significant effect on coarse mode particles, and as a result, other model input parameters become less sensitive. The parameter
ranking of the $PZ10$ scheme for deciduous forest shows that $L_O$ is the most influential parameter. Similarly, for coniferous forest, $L_O$ is found to be one of the most sensitive parameters for most particle sizes. One possible reason for this finding could be the interdependency of the particle mixing length parameter and $L_O$ in the $PZ10$ scheme. Indeed, the mixing length indirectly relates to particle collection efficiencies in the $PZ10$ parameterization (see Eqs. 19, (26-27)). The rankings of the $Z01$ and $ZH14$ parameters are nearly identical for rough and smooth surfaces. This finding is not surprising given that these two parameterizations were
developed using similar assumptions.





In general, dry particle deposition parameterizations developed for different particle size ranges and surfaces vary widely in terms of their complexity in model structure. The complexity in their numerical formulations often depends on the purpose (e.g., operational or research) of the model development (Petroff et al., 2008a). Comparing two previously developed one-dimensional aerosol deposition models for broadleaf and coniferous canopies (see details in Petroff et al., 2008b; Petroff et al., 2009) with the

*PZ10* parameterization, Petroff and Zhang (2010) argued that the mathematical formulations in those models are too complex and require numerous input parameters for implementation in aerosol transport models. Following this hypothesis, we attempt to qualitatively evaluate the relative complexity of the five dry particle deposition parameterizations tested in this study for incorporation into atmospheric transport models.

Of the five parameterizations, we note that the model structure of the *PZ10* is relatively more complex than those of the

*Z01*, *ZH14*, and *ZS14* parameterizations. The complexity of the *KS12* parameterization tends to be different by a large degree between rough (i.e., vegetative canopies and snow) and smooth (i.e., water) surfaces. The *ZS14* formulation (Eqs. (52-65)) is of comparable complexity to the rough surface formulation in the *KS12* parameterization (Eqs. (36-46)), and these parameterizations can be viewed as moderately complex. The formulation of the *Z01* parameterization can be viewed as moderately complex as well. In this parameterization, three processes (Brownian diffusion, interception, and impaction) were parameterized using Eqs. (9-15)

to describe the particle deposition at the collection surface. We claim that the *KS12* parameterization for smooth surfaces is the most complex of the five models. This is mainly because it requires solving the dimensionless dry deposition velocity profiles over smooth surfaces using an analytical approach, which can be complex and computationally-expensive.

A direct qualitative comparison of the relative complexities of the major process terms in the *PZ10* and *Z01* parameterizations is possible because both of these parameterizations are resistance-based (i.e., expressions of $V_d$ in Eqs. 3 and 7

are of similar forms). It is evident from Eqs. (20-32) that the formulations in the *PZ10* parametrization to compute the three surface collection process terms are relatively complex as compared to those in the *Z01* parameterization. In the *ZH14* parameterization (a resistance-based scheme as well), these process terms are not explicitly parameterized. Presumably, by incorporating a large number of LUC dependent constants to compute surface deposition velocity using Eqs. (47-51), simplifications were made possible to the *ZH14* parameterization. The use of fitting parameters to account for poorly understood dry deposition processes in

parameterizations is not uncommon. Due to the complex nature and inadequate understanding of the particle collection processes to leaf surfaces, suggestions were made to treat particle deposition on vegetative surfaces in a simplified manner using empirically derived fitting parameters (Petroff et al., 2008a). Consequently, Petroff and Zhang (2010) also introduced a large number of artificial parameters to account for characteristic length and orientation of the canopy obstacle, and different LUCs to parameterize the particle collection efficiencies (e.g., due to Brownian diffusion, interception, turbulent and inertial impaction). Based on these

considerations and those in the previous paragraph, we claim that the *ZH14* is the simplest of the five parameterizations.

## 6. Conclusions

In terms of overall performance for incorporation in atmospheric transport models, we suggest that parameterization accuracy and uncertainty should be considered jointly, while sensitivity of the model input parameters should be treated

separately for each dry particle deposition parameterization. The paper presents a comprehensive evaluation of the performance of five parameterizations in terms of their accuracy, model output uncertainty, and parameter sensitivity. Based on the results, it is evident that the *ZH14* parameterization is the most accurate for four of the five LUCs (grass, deciduous forest, water, and ice/snow surfaces) and second most accurate for the fifth LUC (coniferous forest). Of the five parametrizations, the uncertainty range for the *ZH14* (11-20%) has the lowest upper bound across the five LUCs for particle size ranging from 0.005-2.5 µm. In

terms of the lower bound of the uncertainty range, the *ZH14* is second to the *Z01* (10-30%) parameterization. We demonstrated





that the Sobol' sensitivity analysis can be successfully applied to dry deposition models to rank the input parameters by taking into account the complex interactions between them.

The large dispersion in the parametrizations' accuracy may indicate that despite considerable efforts in developing sophisticated process-based dry deposition models, there remain major gaps in our understanding of the dry deposition process. Another possible explanation for the large dispersion may be that it is significantly caused by measurement uncertainties, which were not addressed in this paper. However, inter-variability in modeled deposition velocities is not uncommon, as pointed out by Ruijgrok et al. (1995) in an inter-comparison study of several earlier dry deposition models. We emphasize that the accuracy results presented in this paper should be discussed in terms of the locations in which the parameterization accuracy has been evaluated against measurements for the five LUCs (Table 1; Figure 1).

The results from the uncertainty analysis using the Monte Carlo simulations on the size-segregated particles should be of interest to atmospheric transport modelers as well as to the scientific community interested in quantifying the uncertainty bounds in the atmospheric deposition fluxes of pollutants to ecosystems using concentration data from monitoring stations. This is because until now, uncertainties in modeled $V_d$ for size-segregated particles for a suite of currently-available dry particle deposition parameterizations has been unavailable. We stress that future work on probabilistic uncertainty analysis should focus 870 on quantifying uncertainties for additional LUCs than those covered in this study. One of the major limitations of our uncertainty analysis approach is the assumption of uniform distribution of all imprecise model input parameters. To address this limitation, accurate information on the input parameter PDFs is needed.

        With the help of field observations, and improved theoretical knowledge of dry deposition, the Sobol' parameter rankings could be used to fine-tune dry deposition models to better account for processes that are currently lacking or poorly 875 parameterized. Future work should focus on estimating higher order (i.e., second order and total order) Sobol' indices. Such indices would be useful for model developers interested in understanding the joint influence of multiple input parameters on the modeled deposition velocities.

        Based on the qualitative evaluation of relative complexity of the five parameterizations, we suggest that the model structure of the *ZH14* parameterization is the least complex. After reviewing over 100 air quality models, Kouznetsov and Sofiev (2012) reported that resistance-based approaches are extensively implemented in most of those models. Thus, in practice, it may 880 be preferable to use a relatively simple parameterization over a complex (and potentially computationally expensive) one, if the accuracy and uncertainty of the model justify it. Based on these criteria (i.e., accuracy, uncertainty, and complexity), we propose that, of the five parameterizations we tested, the *ZH14* parameterization is currently superior for incorporation into atmospheric transport models.

**7. Code availability**

The codes supporting this study can be found online at: https://osf.io/a6q97/

**Notes:** The authors declare no competing financial interest.

**8. Acknowledgements**

This project was funded by the National Science Foundation's (NSF's) Coupled-Natural-Human System Program through Award No. 1313755.





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


**Table 1. Measurement studies used to evaluate the five parameterizations.**

| Land use category | No. | Title | Authors | Location | Latitude | Longitude | Elevation AMSL (m) | Sampling dates | Particle size (μm) |
|---|---|---|---|---|---|---|---|---|---|
| Grass | 1 | Dry deposition of fine aerosol to a grass surface | Allen et al. (1991) | U.K. | 51.88° N | 0.94° E | 28 | June, 1988-June, 1989 | 0.48 |
| | 2 | An eddy-correlation measurement of particulate deposition from the atmosphere | Wesely et al. (1977) | U.S. | 38.84° N | 90.06° W | 225 | February-March, 1976 | 0.075 |
| | 3 | Eddy correlation measurements of atmospheric fluxes of ozone, sulphur, and particulates during the Champaign intercomparison study | Neumann and den Hartog (1985) | U.S. | 40.11° N | 88.27° W | 225 | June, 1982 | 0.10-2.28 |
| | 4 | Micrometeorological measurements of particle deposition velocities to moorland vegetation | Nemitz et al. (2002) | U.K. | 55.79° N | 3.23° W | 109 | May-October, 1999 | 0.12-0.45 |
| | 5 | Eddy correlation measurements of aerosol deposition to grass | Vong et al. (2004) | U.S. | 44.46° N | 123.11° W | 81 | May-June, 2000 | 0.52 |
| | 6 | The Landes experiment: Biosphere-atmosphere exchanges of ozone and aerosol particles above a pine forest | Lamaud et al. (1994) | France | 44.84° N | 0.58° W | 58 | June, 1992 | 0.04 |
| | 7 | Micrometeorological measurement of the dry deposition flux of sulphate and nitrate aerosols to coniferous forest | Wyers and Duyzers (1996) | The Netherlands | 52.27° N | 5.71° E | 26 | April-December, 1993 | 0.6 |
| | 8 | Atmospheric particles and their interactions with natural surfaces | Gallagher et al. (1997) | The Netherlands | 52.27° N | 5.71° E | 26 | June-July, 1993 | 0.1-0.50 |
| Coniferous forest | 9 | The dry deposition of particles to a forest canopy: a comparison of model and experimental results | Ruijgrok et al. (1997) | The Netherlands | 52.27° N | 5.71° E | 26 | June-July, 1993 | 0.35-0.60 |
| | 10 | Deposition velocities of nucleation mode particles into a Scots pine forest | Rannik et al. (2000) | Finland | 61.85° N | 24.28° E | 181 | September, 2000 | 0.015-0.35 |
| | 11 | Vertical aerosol fluxes measured by the eddy covariance method and deposition of nucleation mode particles above a Scots pine forest in southern Finland | Buzorious et al. (2000) | Finland | 61.85° N | 24.28° E | 181 | March-May, 1997 | 0.015 |
| | 12 | Relaxed eddy accumulation system for size-resolved aerosol particle flux measurements | Gaman et al. (2004) | Finland | 61.85° N | 24.28° E | 181 | September-October, 2001 | 0.05 |
| | 13 | Analyses of flux methods and functional dependencies | Pryor et al. (2007) | Finland | 61.85° N | 24.28° E | 181 | January-December, 2004 | 0.01-0.07 |
| | 14 | Aerosol particle dry deposition to canopy and forest floor measured by two-layer eddy covariance system | Grönholm et al. (2009) | Finland | 61.85° N | 24.28° E | 181 | March, 2003 | 0.01-0.05 |
| Deciduous forest | 15 | Fluxes of gases and particles above a deciduous forest in wintertime | Wesely et al. (1983) | U.S. | 35.98° N | 78.91° W | 77 | January-February, 1981 | 0.48 |
| | 16 | Size-resolved particle deposition velocities of sub-100nm diameter particles over a forest | Pryor (2006) | Denmark | 55.48° N | 11.64° E | 40 | May-June, 2004 | 0.025-0.065 |
| | 17 | Deposition velocity of PM2.5 sulfate in the summer above a deciduous forest in central Japan | Matsuda et al. (2012) | Japan | 36.40° N | 138.58° E | 1380 | July, 2009 | 2.5 |
| Water | 18 | Mechanisms of transport from the atmosphere to the Earth's surface | Möller and Schumann (1970) | (Wind tunnel) | - | - | - | - | 0.03-1.13 |
| | 19 | Particle deposition rates on a water surface as a function of particle diameter and air velocity | Sehmel et al.(1974) | (Wind tunnel) | - | - | - | - | 0.25-29 |
| | 20 | Airborne concentrations and dry deposition fluxes of particulate species to surrogate surfaces deployed in southern Lake Michigan | Zufall et al. (1998) | U.S. | 44.00° N | 87.00° W | - | July, 1994; January, 1995 | 0.4-24.0 |
| | 21 | Determination of size-dependent dry particle deposition velocities with multiple intrinsic elemental tracers | Caffrey et al. (1998) | U.S. | 44.00° N | 87.00° W | - | July, 1994 | 0.05-48.0 |
| | 22 | Aerosol dry deposition measured with eddy-covariance technique at Wasa and Aboa, Dronning Maud Land, Antarctica | Gronlund et al. (2002) | Antarctica | 73.00° S | 13.41° W | 584 | January, 2000 | 0.02-0.2 |
| | 23 | Deposition velocity of ultrafine particles measured with the Eddy-Correlation Method over the Nansen Ice Sheet (Antarctica) | Contini et al. (2010) | Antarctica | 74.88° S | 163.17° W | 84.7 | December, 2006 | 0.06-0.070 |
| | 24 | On the potential contribution of open lead particle emissions to the central Arctic aerosol concentration | Held et al. (2011a) | Arctic Ocean | 87.00° N | 6.00° W | - | August-September, 2008 | 0.045 |
| | 25 | Near-surface profiles of aerosol number concentration and temperature over the Arctic Ocean | Held et al. (2011b) | Arctic Ocean | 87.00° N | 6.00° W | - | August-September, 2008 | 0.045 |
| Ice/snow | 26 | Correlation of dry deposition velocity and friction velocity over different surfaces for PM2.5 and particle number concentrations | Donateo and Contini (2014) | Antarctica | 74.88° S | 163.17° W | - | December, 2006 | 0.015, 0.13 |
| | 27 | An experimental and theoretical investigation of the dry deposition of particles to snow, pine trees and artificial collectors | Ibrahim (1983) | Canada | 51.25° N | 85.32° W | 450 | February, 1980; March, 1981 | 0.7,0.8, 7.0 |
| | 28 | Eddy correlation measurements of the dry deposition of particles in wintertime | Duan et al. (1987) | U.S. | 40.70° N | 77.96° W | 177 | December, 1985 | 0.22,0.73 |
| | 29 | Turbulent aerosol fluxes over the Arctic Ocean 1. Dry deposition over sea and pack ice | Nilsson and Rannik (2001) | Arctic Ocean | 88.00° N | 15.00° E | - | August, 1999 | 0.02,0.05 |





**Table 2. Parameter values and associated uncertainties used in Monte Carlo simulation.**

| Parameter | Base value (assumed) | | Uncertainty range | Reference |
|---|---|---|---|---|
| **Relative humidity, *RH* (%)** | 80 | (all LUCs) | ±5% | Heinonen (2002) |
| **Wind speed, *U* (m/s)** | 4 | (all LUCs) | ±3% | Högström and Smedman (2004) |
| **Friction velocity, $u_*$ (m/s)** | 0.3 | (all LUCs) | ±10% | Andreas (1992) |
| **Monin-Obukhov length, $L_O$ (m)** | 50 | (all LUCs) | ±10% | Weidinger et al. (2000) |
| **Roughness length, $z_0$ (m)** | 0.04 | (Grass) | ±25% | Su et al. (2001) |
| | 1.2 | (Coniferous forest) | | |
| | 1.5 | (Deciduous forest) | | |
| | 0.001 | (Ice/snow) | | |
| **Canopy height, *h* (m)** | 0.5 | (Grass) | ±5% | Larjavaara and Muller-Landau (2013) |
| | 15 | (Coniferous forest) | | |
| | 25 | (Deciduous forest) | | |
| **Zero-plane displacement height, *d* (m)** | 0.3 | (Grass) | ±25% | Su et al. (2001) |
| | 7 | (Coniferous forest) | | |
| | 16 | (Deciduous forest) | | |
| **Leaf area index (one-sided), *LAI* (m²/m²)** | 4 | (Grass) | ±5% | Richardson et al. (2011) |
| | 10 | (Coniferous forest) | | |
| | 10 | (Deciduous forest) | | |

**Table 3. Input parameter ranges for the Sobol' sensitivity analysis.**

| Parameter | Range | | Reference |
|---|---|---|---|
| Relative humidity, *RH* (%) | 10-100 | (all LUCs) | Assumed |
| Wind speed, *U* (m/s) | 1-5 | (all LUCs) | Studies # 1-29 |
| Friction velocity, $u_*$ (m/s) | 0.1-0.5 | (all LUCs) | Studies # 1-29 |
| Monin-Obukhov length, $L_O$ (m) | 10-100 | (all LUCs) | Studies # 1-29 |
| Roughness length, $z_0$ (m) | 0.02-.0.10 | (Grass) | Studies # 1-5 |
| | 0.9-3.0 | (Coniferous forest) | Studies # 6-14 |
| | 0.5-1.5 | (Deciduous forest) | Studies # 15-17 |
| | 0.00002-0.0066 | (Ice/snow) | Studies # 22-29 |
| Canopy height, *h* (m) | 0.15-0.77 | (Grass) | Studies # 1-5 |
| | 14-20 | (Coniferous forest) | Studies # 6-14 |
| | 20-25 | (Deciduous forest) | Studies # 15-17 |
| Zero-plane displacement height, *d* (m) | 0.10-0.49 | (Grass) | Studies # 1-5 |
| | 7-12 | (Coniferous forest) | Studies # 6-14 |
| | 8-16 | (Deciduous forest) | Studies # 15-17 |
| Leaf area index (one-sided), *LAI* (m²/m²) | 1-4 | (Grass) | Studies # 1-5 |
| | 0.2-10 | (Coniferous forest) | Studies # 6-14 |
| | 0.2-10 | (Deciduous forest) | Studies # 15-17 |

[1]Studies are listed in Table 1.

**Table 4. Results of the normalized mean bias factors for grass (bold-faced value indicates the most accurate parameterization).**

| Study | Dry particle deposition parameterization | | | | |
|---|---|---|---|---|---|
| | *Z01* | *PZ10* | *KS12* | *ZH14* | *ZS14* |
| Allen et al. (1991) | -17.61 | 15.96 | -18.12 | -0.55 | -5.13 |
| Wesely et al. (1977) | -2.78 | -28.78 | -7.56 | -10.62 | -102.92 |
| Neumann and den Hartog (1985) | 0.96 | -0.12 | -0.50 | 4.79 | 0.56 |
| Nemitz et al. (2002) | 5.15 | 1.12 | -3.82 | 2.17 | -0.10 |
| Vong et al. (2004) | -4.55 | -4.55 | -25.71 | -2.12 | -4.03 |
| *Five studies* | 5.45 | -1.80 | -9.37 | **-0.54** | -4.30 |




**Table 5. Results of the normalized mean bias factors for coniferous forest (bold-faced value indicates the most accurate parameterization).**

| | Dry particle deposition parameterization | | | | |
|---|---|---|---|---|---|
| Study | Z01 | PZ10 | KS12 | ZH14 | ZS14 |
| Lamaud et al. (1994) | 0.77 | -12.75 | -1.91 | -2.14 | -16.71 |
| Wyers and Duyzers (1996) | -25.98 | -26.98 | -81.39 | -13.57 | -4.51 |
| Gallagher et al. (1997) | -1.74 | -6.34 | -19.83 | -1.90 | -2.39 |
| Ruijgrok et al. (1997) | -5.70 | -0.51 | -0.93 | -2.58 | -0.48 |
| Rannik et al. (2000) | 3.16 | -2.54 | 0.51 | -1.13 | -18.75 |
| Buzorious et al. (2000) | 0.75 | -6.65 | -2.91 | -4.53 | -67.41 |
| Gaman et al. (2004) | -0.90 | -13.00 | -6.12 | -1.84 | -17.45 |
| Pryor et al. (2007) | 0.69 | -5.37 | -0.26 | -0.84 | -12.22 |
| Grönholm et al. (2009) | 0.95 | 0.13 | 1.55 | 1.72 | -1.90 |
| *Nine studies* | -2.35 | -3.93 | **-1.75** | -2.31 | -3.67 |

**Table 6. Results of the normalized mean bias factors for deciduous forest (bold-faced values indicate the most accurate parameterization).**

| | Dry particle deposition parameterization | | | | |
|---|---|---|---|---|---|
| Study | Z01 | PZ10 | KS12 | ZH14 | ZS14 |
| Wesely et al. (1983) | -9.25 | -130.30 | -34.58 | -5.27 | -2.28 |
| Pryor (2006) | 1.55 | -2.42 | -2.42 | -0.90 | -13.62 |
| Matsuda et al. (2012) | -5.19 | -1.34 | -1.91 | -2.37 | -0.15 |
| *Three studies* | -8.11 | -4.51 | -4.96 | **-3.75** | -10.93 |

**Table 7. Results of the normalized mean bias factors for water surfaces (bold-faced value indicates the most accurate parameterization).**

| | Dry particle deposition parameterization | | | | |
|---|---|---|---|---|---|
| Study | Z01 | PZ10 | KS12 | ZH14 | ZS14 |
| Möller and Schumann (1970) | 18.87 | -1.65 | -2.51 | 10.28 | 106.00 |
| Sehmel et al. (1974) | 0.44 | 0.45 | -0.59 | 1.51 | 3.65 |
| Zufall et al. (1998) | -0.144 | -0.39 | -0.47 | -0.33 | 5.14 |
| Caffrey et al. (1998) | 0.75 | 0.35 | -0.85 | 0.70 | 3.61 |
| *Four studies* | 0.52 | -0.89 | -0.64 | **-0.25** | 4.22 |

**Table 8. Results of the normalized mean bias factors for ice/snow surfaces (bold-faced value indicates the most accurate parameterization).**

| | Dry particle deposition parameterization | | | |
|---|---|---|---|---|
| Study | Z01 | PZ10 | KS12 | ZH14 |
| Gronlund et al. (2002) | -1.22 | -271.73 | -105.92 | -2.58 |
| Contini et al. (2010) | 5.68 | -57.22 | -24.96 | 0.62 |
| Held et al. (2011a) | 2.96 | -38.66 | -15.58 | 0.67 |
| Held et al. (2011b) | 2.78 | -42.93 | -16.71 | 0.52 |
| Donateo and Contini (2014) | 1.62 | -35.26 | -12.57 | -0.32 |
| Ibrahim (1983) | 4.14 | -6.72 | -7.72 | 3.98 |
| Duan et al. (1987) | 0.22 | -12.09 | -15.49 | 0.42 |
| Nilsson and Rannik (2001) | 1.69 | -37.78 | -13.46 | -0.74 |
| *Eight studies* | 1.98 | -53.03 | -21.80 | **0.26** |





**Table 9.** Median (50th percentile) values of the simulated dry deposition velocities (m s$^{-1}$) with 5th and 95th percentiles of distribution indicating the range of uncertainty.

| Land use category | Particle size (μm) | Z01 5th | Z01 50th | Z01 95th | PZ10 5th | PZ10 50th | PZ10 95th | KS12 5th | KS12 50th | KS12 95th | ZH14 5th | ZH14 50th | ZH14 95th | ZS14* 5th | ZS14* 50th | ZS14* 95th |
|---|---|---|---|---|---|---|---|---|---|---|---|---|---|---|---|---|
| Grass | 0.005 | $1.72\times10^{-2}$ | $1.9\times10^{-2}$ | $2.1\times10^{-2}$ | $9.0\times10^{-3}$ | $9.5\times10^{-3}$ | $1.0\times10^{-2}$ | $1.7\times10^{-2}$ | $1.77\times10^{-2}$ | $1.86\times10^{-2}$ | $1.4\times10^{-3}$ | $1.5\times10^{-3}$ | $1.7\times10^{-3}$ | $4.0\times10^{-4}$ | $5.0\times10^{-4}$ | $5.0\times10^{-4}$ |
| | 0.05 | $5.2\times10^{-3}$ | $5.7\times10^{-3}$ | $6.2\times10^{-3}$ | $8.0\times10^{-4}$ | $8.0\times10^{-4}$ | $9.0\times10^{-4}$ | $9.0\times10^{-4}$ | $9.0\times10^{-4}$ | $9.0\times10^{-4}$ | $1.4\times10^{-3}$ | $1.5\times10^{-3}$ | $1.7\times10^{-3}$ | $1.0\times10^{-4}$ | $1.0\times10^{-4}$ | $1.0\times10^{-4}$ |
| | 0.5 | $9.0\times10^{-4}$ | $9.0\times10^{-4}$ | $1.1\times10^{-3}$ | $9.0\times10^{-4}$ | $9.0\times10^{-4}$ | $9.0\times10^{-4}$ | $1.0\times10^{-4}$ | $1.0\times10^{-4}$ | $1.0\times10^{-4}$ | $1.4\times10^{-3}$ | $1.5\times10^{-3}$ | $1.7\times10^{-3}$ | $7.0\times10^{-4}$ | $9.0\times10^{-4}$ | $1.0\times10^{-3}$ |
| | 1.0 | $6.15\times10^{-4}$ | $6.7\times10^{-4}$ | $7.25\times10^{-4}$ | $1.7\times10^{-3}$ | $1.8\times10^{-3}$ | $1.9\times10^{-3}$ | $1.45\times10^{-4}$ | $1.49\times10^{-4}$ | $1.55\times10^{-4}$ | $1.45\times10^{-3}$ | $1.58\times10^{-3}$ | $1.72\times10^{-3}$ | $1.47\times10^{-3}$ | $1.75\times10^{-3}$ | $2.06\times10^{-3}$ |
| | 1.5 | $5.72\times10^{-4}$ | $6.16\times10^{-4}$ | $6.59\times10^{-4}$ | $2.71\times10^{-3}$ | $2.87\times10^{-3}$ | $3.04\times10^{-3}$ | $2.59\times10^{-4}$ | $2.74\times10^{-4}$ | $2.91\times10^{-4}$ | $1.63\times10^{-3}$ | $1.66\times10^{-3}$ | $1.8\times10^{-3}$ | $2.20\times10^{-3}$ | $2.61\times10^{-3}$ | $3.06\times10^{-3}$ |
| | 2.0 | $6.16\times10^{-4}$ | $6.55\times10^{-4}$ | $6.93\times10^{-4}$ | $3.97\times10^{-3}$ | $4.23\times10^{-3}$ | $4.5\times10^{-3}$ | $4.3\times10^{-4}$ | $4.6\times10^{-4}$ | $4.94\times10^{-4}$ | $1.8\times10^{-3}$ | $1.77\times10^{-3}$ | $1.9\times10^{-3}$ | $2.91\times10^{-3}$ | $3.44\times10^{-3}$ | $4.10\times10^{-3}$ |
| | 2.5 | $7.0\times10^{-4}$ | $7.0\times10^{-4}$ | $8.0\times10^{-4}$ | $5.6\times10^{-3}$ | $6.0\times10^{-3}$ | $6.4\times10^{-3}$ | $7.0\times10^{-4}$ | $7.0\times10^{-4}$ | $8.0\times10^{-4}$ | $1.8\times10^{-3}$ | $1.9\times10^{-3}$ | $2.0\times10^{-3}$ | $3.6\times10^{-3}$ | $4.2\times10^{-3}$ | $4.9\times10^{-3}$ |
| Coniferous Forest | 0.005 | $1.62\times10^{-2}$ | $1.8\times10^{-2}$ | $1.98\times10^{-2}$ | $2.31\times10^{-2}$ | $2.46\times10^{-2}$ | $2.62\times10^{-2}$ | $2.87\times10^{-2}$ | $2.99\times10^{-2}$ | $3.14\times10^{-2}$ | $1.1\times10^{-3}$ | $1.2\times10^{-3}$ | $1.3\times10^{-3}$ | | | |
| | 0.05 | $4.4\times10^{-3}$ | $4.9\times10^{-3}$ | $5.3\times10^{-3}$ | $2.6\times10^{-3}$ | $2.7\times10^{-3}$ | $2.9\times10^{-3}$ | $1.4\times10^{-3}$ | $1.5\times10^{-3}$ | $1.6\times10^{-3}$ | $1.1\times10^{-3}$ | $1.2\times10^{-3}$ | $1.3\times10^{-3}$ | | | |
| | 0.5 | $7.0\times10^{-4}$ | $8.0\times10^{-4}$ | $8.0\times10^{-4}$ | $4.7\times10^{-3}$ | $4.9\times10^{-3}$ | $5.1\times10^{-3}$ | $2.0\times10^{-4}$ | $2.0\times10^{-4}$ | $2.0\times10^{-4}$ | $1.1\times10^{-3}$ | $1.2\times10^{-3}$ | $1.3\times10^{-3}$ | | | |
| | 1.0 | $4.87\times10^{-4}$ | $5.29\times10^{-3}$ | $5.71\times10^{-3}$ | $9.28\times10^{-3}$ | $9.76\times10^{-3}$ | $1.03\times10^{-2}$ | $3.24\times10^{-4}$ | $3.5\times10^{-4}$ | $3.79\times10^{-4}$ | $1.16\times10^{-3}$ | $1.27\times10^{-3}$ | $1.38\times10^{-3}$ | | | |
| | 1.5 | $4.7\times10^{-3}$ | $5.03\times10^{-4}$ | $5.73\times10^{-4}$ | $1.48\times10^{-2}$ | $1.57\times10^{-2}$ | $1.66\times10^{-2}$ | $6.27\times10^{-4}$ | $6.96\times10^{-4}$ | $7.7\times10^{-4}$ | $1.23\times10^{-3}$ | $1.34\times10^{-3}$ | $1.45\times10^{-3}$ | | | |
| | 2.0 | $5.29\times10^{-4}$ | $5.6\times10^{-4}$ | $5.92\times10^{-4}$ | $2.11\times10^{-2}$ | $2.26\times10^{-2}$ | $2.41\times10^{-2}$ | $1.07\times10^{-3}$ | $1.20\times10^{-3}$ | $1.34\times10^{-3}$ | $1.34\times10^{-3}$ | $1.45\times10^{-3}$ | $1.56\times10^{-3}$ | | | |
| | 2.5 | $6.0\times10^{-4}$ | $7.0\times10^{-4}$ | $7.0\times10^{-4}$ | $2.72\times10^{-2}$ | $2.93\times10^{-2}$ | $3.15\times10^{-2}$ | $1.7\times10^{-3}$ | $1.9\times10^{-3}$ | $2.1\times10^{-3}$ | $1.5\times10^{-3}$ | $1.6\times10^{-3}$ | $1.7\times10^{-3}$ | | | |
| Deciduous Forest | 0.005 | $1.28\times10^{-2}$ | $1.42\times10^{-2}$ | $1.58\times10^{-2}$ | $9.7\times10^{-3}$ | $1.08\times10^{-2}$ | $1.2\times10^{-2}$ | $2.87\times10^{-2}$ | $3.0\times10^{-2}$ | $3.14\times10^{-2}$ | $2.2\times10^{-3}$ | $2.2\times10^{-3}$ | $2.6\times10^{-3}$ | | | |
| | 0.05 | $4.10\times10^{-3}$ | $4.6\times10^{-3}$ | $5.0\times10^{-3}$ | $9.0\times10^{-4}$ | $1.0\times10^{-3}$ | $1.1\times10^{-3}$ | $1.4\times10^{-3}$ | $1.5\times10^{-3}$ | $1.6\times10^{-3}$ | $2.2\times10^{-3}$ | $2.2\times10^{-3}$ | $2.6\times10^{-3}$ | | | |
| | 0.5 | $7.0\times10^{-4}$ | $8.0\times10^{-4}$ | $8.0\times10^{-4}$ | $9.0\times10^{-4}$ | $1.1\times10^{-3}$ | $1.2\times10^{-3}$ | $2.0\times10^{-4}$ | $2.0\times10^{-4}$ | $2.0\times10^{-4}$ | $2.2\times10^{-3}$ | $2.2\times10^{-3}$ | $2.6\times10^{-3}$ | | | |
| | 1.0 | $4.84\times10^{-4}$ | $5.25\times10^{-4}$ | $5.67\times10^{-4}$ | $1.68\times10^{-3}$ | $1.96\times10^{-3}$ | $2.25\times10^{-3}$ | $9.02\times10^{-5}$ | $9.16\times10^{-5}$ | $9.35\times10^{-5}$ | $2.25\times10^{-3}$ | $2.46\times10^{-3}$ | $2.68\times10^{-3}$ | | | |
| | 1.5 | $4.65\times10^{-4}$ | $4.98\times10^{-4}$ | $5.31\times10^{-4}$ | $2.46\times10^{-3}$ | $2.86\times10^{-3}$ | $3.28\times10^{-3}$ | $1.66\times10^{-4}$ | $1.71\times10^{-4}$ | $1.77\times10^{-4}$ | $2.33\times10^{-3}$ | $2.54\times10^{-3}$ | $2.76\times10^{-3}$ | | | |
| | 2.0 | $5.2\times10^{-4}$ | $5.49\times10^{-4}$ | $5.78\times10^{-4}$ | $3.29\times10^{-3}$ | $3.81\times10^{-3}$ | $4.35\times10^{-3}$ | $2.77\times10^{-4}$ | $2.87\times10^{-4}$ | $3.0\times10^{-4}$ | $2.43\times10^{-3}$ | $2.65\times10^{-3}$ | $2.86\times10^{-3}$ | | | |
| | 2.5 | $6.0\times10^{-4}$ | $7.0\times10^{-4}$ | $7.0\times10^{-4}$ | $4.2\times10^{-3}$ | $4.8\times10^{-3}$ | $5.5\times10^{-3}$ | $1.7\times10^{-3}$ | $1.9\times10^{-3}$ | $2.1\times10^{-3}$ | $2.6\times10^{-3}$ | $2.6\times10^{-3}$ | $3.0\times10^{-3}$ | | | |
| Water | 0.005 | $8.9\times10^{-3}$ | $9.9\times10^{-3}$ | $1.1\times10^{-2}$ | $4.0\times10^{-4}$ | $4.0\times10^{-4}$ | $5.0\times10^{-4}$ | $4.66\times10^{-4}$ | $5.12\times10^{-4}$ | $5.58\times10^{-4}$ | $1.6\times10^{-3}$ | $1.8\times10^{-3}$ | $1.9\times10^{-3}$ | $1.04\times10^{-2}$ | $1.14\times10^{-2}$ | $1.24\times10^{-2}$ |
| | 0.05 | $4.0\times10^{-3}$ | $4.4\times10^{-3}$ | $4.8\times10^{-3}$ | $1.0\times10^{-4}$ | $1.0\times10^{-4}$ | $1.0\times10^{-4}$ | $5.06\times10^{-5}$ | $6.06\times10^{-5}$ | $6.06\times10^{-5}$ | $1.6\times10^{-3}$ | $1.8\times10^{-3}$ | $1.9\times10^{-3}$ | $2.3\times10^{-3}$ | $2.8\times10^{-3}$ | $3.3\times10^{-3}$ |
| | 0.5 | $9.0\times10^{-4}$ | $1.0\times10^{-3}$ | $1.1\times10^{-3}$ | $1.0\times10^{-4}$ | $1.0\times10^{-4}$ | $1.0\times10^{-4}$ | $1.49\times10^{-5}$ | $1.64\times10^{-5}$ | $1.79\times10^{-5}$ | $1.6\times10^{-3}$ | $1.8\times10^{-3}$ | $1.9\times10^{-3}$ | $2.19\times10^{-2}$ | $2.62\times10^{-2}$ | $3.06\times10^{-2}$ |
| | 1.0 | $6.82\times10^{-4}$ | $7.44\times10^{-4}$ | $8.07\times10^{-4}$ | $1.11\times10^{-4}$ | $1.13\times10^{-4}$ | $1.15\times10^{-4}$ | $5.0\times10^{-5}$ | $5.5\times10^{-5}$ | $5.99\times10^{-5}$ | $1.66\times10^{-3}$ | $1.82\times10^{-3}$ | $1.98\times10^{-3}$ | $4.32\times10^{-2}$ | $5.08\times10^{-2}$ | $5.87\times10^{-2}$ |
| | 1.5 | $6.39\times10^{-4}$ | $6.92\times10^{-4}$ | $7.47\times10^{-4}$ | $1.85\times10^{-4}$ | $1.89\times10^{-4}$ | $1.94\times10^{-4}$ | $1.08\times10^{-4}$ | $1.19\times10^{-4}$ | $1.3\times10^{-4}$ | $1.73\times10^{-3}$ | $1.9\times10^{-3}$ | $2.06\times10^{-3}$ | $6.26\times10^{-2}$ | $7.29\times10^{-2}$ | $8.31\times10^{-2}$ |
| | 2.0 | $6.88\times10^{-4}$ | $7.42\times10^{-4}$ | $8.0\times10^{-4}$ | $2.87\times10^{-4}$ | $2.95\times10^{-4}$ | $3.06\times10^{-4}$ | $1.90\times10^{-4}$ | $2.09\times10^{-4}$ | $2.27\times10^{-4}$ | $1.84\times10^{-3}$ | $2.0\times10^{-3}$ | $2.17\times10^{-3}$ | $8.02\times10^{-2}$ | $9.25\times10^{-2}$ | $1.04\times10^{-1}$ |
| | 2.5 | $8.0\times10^{-4}$ | $9.0\times10^{-4}$ | $9.0\times10^{-4}$ | $4.0\times10^{-4}$ | $4.0\times10^{-4}$ | $5.0\times10^{-4}$ | $2.96\times10^{-4}$ | $3.25\times10^{-4}$ | $3.55\times10^{-4}$ | $2.0\times10^{-3}$ | $2.1\times10^{-3}$ | $2.3\times10^{-3}$ | $9.63\times10^{-2}$ | $1.1\times10^{-1}$ | $1.23\times10^{-1}$ |
| Ice/snow | 0.005 | $1.15\times10^{-2}$ | $1.26\times10^{-2}$ | $1.38\times10^{-2}$ | $4.0\times10^{-4}$ | $4.0\times10^{-4}$ | $4.0\times10^{-4}$ | $2.8\times10^{-2}$ | $2.93\times10^{-2}$ | $3.07\times10^{-2}$ | $1.1\times10^{-3}$ | $1.2\times10^{-3}$ | $1.3\times10^{-3}$ | | | |
| | 0.05 | $3.4\times10^{-3}$ | $3.7\times10^{-3}$ | $4.0\times10^{-3}$ | $2.0\times10^{-5}$ | $2.0\times10^{-5}$ | $2.0\times10^{-5}$ | $1.4\times10^{-3}$ | $1.5\times10^{-3}$ | $1.5\times10^{-3}$ | $1.1\times10^{-3}$ | $1.2\times10^{-3}$ | $1.3\times10^{-2}$ | | | |
| | 0.5 | $6.0\times10^{-4}$ | $7.0\times10^{-4}$ | $7.0\times10^{-4}$ | $2.0\times10^{-5}$ | $2.0\times10^{-5}$ | $2.0\times10^{-5}$ | $2.0\times10^{-4}$ | $2.0\times10^{-4}$ | $2.0\times10^{-4}$ | $1.1\times10^{-3}$ | $1.2\times10^{-3}$ | $1.3\times10^{-2}$ | | | |
| | 1.0 | $4.41\times10^{-4}$ | $4.81\times10^{-4}$ | $5.23\times10^{-4}$ | $6.71\times10^{-5}$ | $6.86\times10^{-5}$ | $7.06\times10^{-5}$ | $4.38\times10^{-4}$ | $4.85\times10^{-4}$ | $5.37\times10^{-4}$ | $1.15\times10^{-3}$ | $1.26\times10^{-3}$ | $1.37\times10^{-3}$ | | | |
| | 1.5 | $4.67\times10^{-4}$ | $5.14\times10^{-4}$ | $5.69\times10^{-4}$ | $1.45\times10^{-4}$ | $1.49\times10^{-4}$ | $1.55\times10^{-4}$ | $9.14\times10^{-4}$ | $1.04\times10^{-3}$ | $1.17\times10^{-3}$ | $1.24\times10^{-3}$ | $1.34\times10^{-3}$ | $1.45\times10^{-3}$ | | | |
| | 2.0 | $6.0\times10^{-4}$ | $6.84\times10^{-4}$ | $7.92\times10^{-4}$ | $2.54\times10^{-4}$ | $2.63\times10^{-4}$ | $2.75\times10^{-4}$ | $1.61\times10^{-3}$ | $1.84\times10^{-3}$ | $2.1\times10^{-3}$ | $1.35\times10^{-3}$ | $1.46\times10^{-3}$ | $1.57\times10^{-3}$ | | | |
| | 2.5 | $9.0\times10^{-4}$ | $1.0\times10^{-3}$ | $1.2\times10^{-3}$ | $4.0\times10^{-4}$ | $4.0\times10^{-4}$ | $4.0\times10^{-4}$ | $2.5\times10^{-3}$ | $2.9\times10^{-3}$ | $3.3\times10^{-3}$ | $1.5\times10^{-3}$ | $1.6\times10^{-3}$ | $1.7\times10^{-3}$ | | | |

*The parameterization does not categorize rough or vegetative surfaces into different LUCs. In this analysis, grass is used to represent a rough surface.



**Table 10. Normalized uncertainties in modeled dry deposition velocities.**

| Land use category | Particle size, $d_p$ (µm) | Dry particle deposition parameterization | | | | |
|---|---|---|---|---|---|---|
| | | *Z01* | *PZ10* | *KS12* | *ZH14* | *ZS14* |
| Grass | 0.005 | 0.20 | 0.11 | 0.09 | 0.20 | 0.20 |
| | 0.05 | 0.18 | 0.13 | 0.00 | 0.20 | 0.00 |
| | 0.5 | 0.22 | 0.00 | 0.00 | 0.20 | 0.33 |
| | 1.0 | 0.16 | 0.11 | 0.07 | 0.17 | 0.34 |
| | 1.5 | 0.14 | 0.11 | 0.12 | 0.17 | 0.33 |
| | 2.0 | 0.12 | 0.12 | 0.14 | 0.16 | 0.32 |
| | 2.5 | 0.14 | 0.13 | 0.14 | 0.11 | 0.31 |
| Coniferous forest | 0.005 | 0.20 | 0.13 | 0.09 | 0.17 | |
| | 0.05 | 0.18 | 0.11 | 0.13 | 0.17 | |
| | 0.5 | 0.13 | 0.08 | 0.00 | 0.17 | |
| | 1.0 | 0.16 | 0.10 | 0.16 | 0.17 | |
| | 1.5 | 0.13 | 0.11 | 0.20 | 0.16 | |
| | 2.0 | 0.11 | 0.13 | 0.22 | 0.15 | |
| | 2.5 | 0.14 | 0.15 | 0.21 | 0.13 | |
| Deciduous forest | 0.005 | 0.21 | 0.21 | 0.09 | 0.18 | |
| | 0.05 | 0.20 | 0.20 | 0.13 | 0.18 | |
| | 0.5 | 0.13 | 0.27 | 0.00 | 0.18 | |
| | 1.0 | 0.16 | 0.29 | 0.04 | 0.18 | |
| | 1.5 | 0.13 | 0.28 | 0.06 | 0.17 | |
| | 2.0 | 0.10 | 0.28 | 0.08 | 0.16 | |
| | 2.5 | 0.14 | 0.27 | 0.21 | 0.15 | |
| Water | 0.005 | 0.21 | 0.25 | 0.18 | 0.17 | 0.18 |
| | 0.05 | 0.18 | 0.00 | 0.18 | 0.17 | 0.36 |
| | 0.5 | 0.20 | 0.00 | 0.18 | 0.17 | 0.33 |
| | 1.0 | 0.17 | 0.03 | 0.18 | 0.18 | 0.31 |
| | 1.5 | 0.15 | 0.05 | 0.18 | 0.17 | 0.28 |
| | 2.0 | 0.15 | 0.07 | 0.18 | 0.16 | 0.26 |
| | 2.5 | 0.11 | 0.25 | 0.18 | 0.14 | 0.24 |
| Ice/snow | 0.005 | 0.18 | 0.00 | 0.09 | 0.17 | |
| | 0.05 | 0.16 | 0.00 | 0.07 | 0.17 | |
| | 0.5 | 0.14 | 0.00 | 0.00 | 0.17 | |
| | 1.0 | 0.17 | 0.05 | 0.20 | 0.17 | |
| | 1.5 | 0.20 | 0.07 | 0.25 | 0.16 | |
| | 2.0 | 0.28 | 0.08 | 0.27 | 0.15 | |
| | 2.5 | 0.30 | 0.00 | 0.28 | 0.13 | |

Note: a normalized uncertainty value of zero indicates that the 95[th] and 5[th] percentile $V_d$ are of equal magnitude.

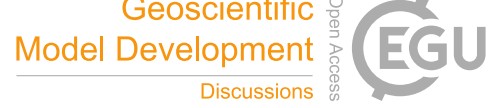


**Table 11. Ranking of first order Sobol' sensitivity indices for the five dry particle deposition parameterizations.**

| Land use category | $d_p$ (µm) | Z01 | PZ10 | KS12 | ZH14 | ZS14 |
|---|---|---|---|---|---|---|
| Grass | 0.001 | $u_*$, $z_0$, $L_O$, (RH, $\rho_P$) | $u_*$, LAI, $z_0$, U, $L_O$, (h, d, RH, $\rho_P$) | $u_*$, (RH, $\rho_P$, LAI) | $u_*$, ($z_0$, $L_O$, RH, $\rho_P$) | U, $u_*$, (RH, $\rho_P$) |
| | 0.01 | $u_*$, $z_0$, $L_O$, (RH, $\rho_P$) | LAI, U, $u_*$, ($L_O$, h), ($z_0$, d, RH, $\rho_P$) | $u_*$, (RH, $\rho_P$, LAI) | $u_*$, ($z0$, $L_O$, RH, $\rho P$) | U, $u_*$, (RH, $\rho_P$) |
| | 0.1 | $u_*$, ($z_0$, $L_O$, RH, $\rho_P$) | LAI, U, d, ($u_*$, $L_O$, h, $z_0$, RH, $\rho_P$) | $u_*$, (RH, $\rho_P$, LAI) | $u_*$, ($z_0$, $L_O$, RH, $\rho_P$) | $u_*$, U, (RH, $\rho_P$) |
| | 1.0 | $u_*$, $\rho_P$, RH, ($z_0$, $L_O$) | U, LAI, RH, $u_*$, $L_O$, (h, $z_0$, d, $\rho_P$) | RH, $u_*$, $\rho_P$, LAI | $u_*$, ($z_0$, $L_O$, RH, $\rho_P$) | $u_*$, U, RH, $\rho_P$ |
| | 10 | RH, $u_*$, $\rho_P$, ($z_0$, $L_O$) | RH, $u_*$, U, LAI, ($\rho_P$, $z_0$), $L_O$, (h, d) | RH, $u_*$, LAI, $\rho_P$ | RH, $\rho_P$, ($u_*$, $L_O$, $z_0$) | $u_*$, U, RH, $\rho_P$ |
| Coniferous Forest | 0.001 | $u_*$, $L_O$, $z_0$, (RH, $\rho_P$) | $u_*$, $L_O$, $z_0$, LAI, h, (d, U, RH, $\rho_P$) | $u_*$, (RH, $\rho_P$, LAI) | $u_*$, $L_O$, ($z_0$, RH, $\rho_P$) | |
| | 0.01 | $u_*$, $L_O$, $z_0$, (RH, $\rho_P$) | $L_O$, LAI, $u_*$, U, $z_0$, h, (d, RH, $\rho_P$) | $u_*$, (RH, $\rho_P$, LAI) | $u_*$, $L_O$, ($z_0$, RH, $\rho_P$) | |
| | 0.1 | $u_*$, $L_O$, ($z_0$, RH, $\rho_P$) | $L_O$, LAI, U, $u_*$, (d, $z_0$, h, RH, $\rho_P$) | $u_*$, (RH, $\rho_P$, LAI) | $u_*$, ($L_O$, $z_0$, RH, $\rho_P$) | |
| | 1.0 | $u_*$, $\rho_P$, RH, ($L_O$, $z_0$) | $L_O$, U, LAI, $u_*$, RH, ($\rho_P$, d, $z_0$, h) | $u_*$, RH, $\rho_P$, LAI | $u_*$, $L_O$, ($z_0$, RH, $\rho_P$) | |
| | 10 | RH, $u_*$, $\rho_P$, $L_O$, $z_0$ | $u_*$, $L_O$, RH, $z_0$, ($\rho_P$, U), (LAI, d, h) | RH, $u_*$, LAI, $\rho_P$ | RH, $u_*$, $L_O$, $z_0$, $\rho_P$ | |
| Deciduous Forest | 0.001 | $u_*$, $z_0$, $L_O$, (RH, $\rho_P$) | $L_O$, $u_*$, LAI, $z_0$, U, (h, d, RH, $\rho_P$) | $u_*$, (RH, $\rho_P$, LAI) | $u_*$, $L_O$, ($z_0$, RH, $\rho_P$) | |
| | 0.01 | $u_*$, $z_0$, $L_O$, (RH, $\rho_P$) | $L_O$, LAI, $u_*$, U, ($z_0$, d), (h, RH, $\rho_P$) | $u_*$, (RH, $\rho_P$, LAI) | $u_*$, $L_O$, ($z_0$, RH, $\rho_P$) | |
| | 0.1 | $u_*$, $z_0$, ($L_O$, RH, $\rho_P$) | $L_O$, LAI, U, $u_*$, d, ($z_0$, h, RH, $\rho_P$) | $u_*$, (RH, $\rho_P$, LAI) | $u_*$, $L_O$, ($z_0$, RH, $\rho_P$) | |
| | 1.0 | $u_*$, $\rho_P$, ($L_O$, $z_0$, RH) | $L_O$, LAI, U, $u_*$, RH, ($z_0$, d), (h, $\rho_P$) | RH, $\rho_P$, $u_*$, LAI | $u_*$, $L_O$, ($z_0$, RH, $\rho_P$) | |
| | 10 | RH, $u_*$, $\rho_P$, $z_0$, $L_O$ | $L_O$, RH, $u_*$, LAI, $\rho_P$, U, $z_0$, (d, h) | RH, $\rho_P$, $u_*$, LAI | RH, $u_*$, LAI, $L_O$, $\rho_P$, $z_0$ | |
| Water | 0.001 | $u_*$, ($L_O$, RH, $\rho_P$) | $u_*$, ($L_O$, RH, $\rho_P$) | $u_*$, $L_O$, ($\rho_P$, RH) | $u_*$, ($L_O$, RH, $\rho_P$) | |
| | 0.01 | $u_*$, ($L_O$, RH, $\rho_P$) | $u_*$, ($L_O$, RH, $\rho_P$) | $u_*$, $L_O$, ($\rho_P$, RH) | $u_*$, ($L_O$, RH, $\rho_P$) | |
| | 0.1 | $u_*$, ($L_O$, RH, $\rho_P$) | $u_*$, $\rho_P$, ($L_O$, RH) | $u_*$, $L_O$, ($\rho_P$, RH) | $u_*$, ($L_O$, RH, $\rho_P$) | |
| | 1.0 | $u_*$, $\rho_P$, (RH, $L_O$) | RH, $\rho_P$, ($u_*$, $L_O$) | $\rho_P$, $u_*$, (RH, L) | $u_*$, ($L_O$, RH, $\rho_P$) | |
| | 10 | $u_*$, RH, ($\rho_P$, $L_O$) | RH, $\rho_P$, ($u_*$, $L_O$) | $\rho_P$, $u_*$, $L_O$, RH | $u_*$, RH, ($L_O$, $\rho_P$) | |
| Ice/snow | 0.001 | $u_*$, $L_O$, ($z_0$, RH, $\rho_P$) | $u_*$, ($L_O$, $z_0$, RH, $\rho_P$) | $u_*$, RH, $\rho_P$ | $u_*$, ($L_O$, $z_0$, RH, $\rho_P$) | |
| | 0.01 | $u_*$, $z_0$, $L_O$, (RH, $\rho_P$) | $u_*$, ($L_O$, $z_0$, RH, $\rho_P$) | $u_*$, RH, $\rho_P$ | $u_*$, ($L_O$, $z_0$, RH, $\rho_P$) | |
| | 0.1 | $u_*$, ($L_O$, $z_0$, RH, $\rho_P$) | $u_*$, ($L_O$, $z_0$, RH, $\rho_P$) | $u_*$, RH, $\rho_P$ | $u_*$, ($L_O$, $z_0$, RH, $\rho_P$) | |
| | 1.0 | $u_*$, ($L_O$, $z_0$, RH, $\rho_P$) | RH, $\rho_P$, ($z_0$, $L_O$, $z_0$) | $u_*$, RH, $\rho_P$ | $u_*$, ($L_O$, $z_0$, RH, $\rho_P$) | |
| | 10 | $u_*$, RH, ($\rho_P$, $L_O$, $z_0$) | RH, $\rho_P$, ($z_0$, $L_O$, $z_0$) | RH, $u_*$, $\rho_P$ | $u_*$, RH, ($z_0$, $L_O$, $\rho_P$) | |

Note: the ranking of the parameters is from most sensitive to least sensitive. Parameters within the parentheses have identical Sobol' first order indices.