# Peer review of "Evaluation of five dry particle deposition parameterizations for incorporation into atmospheric transport models"

_Geoscientific Model Development, 2017_

## Short Comment (SC1) · 28 Jun 2017

Tanvir,

Thanks for your manuscript. As you are probably aware GMD has the mission to make the program code and reference data being the basis of published manuscripts accessible to the reader. This will make it easier for readers to apply and extend the presented model(s) for their work which is also in the interest of the author. It is also vital to guarantee reproducibility of the shown results which is evidently a key objective of scientific work. Providing persistent access to the software code is obviously a prerequisite to achieve this.

[Figure]

I would like to suggest the following changes to your manuscript for a revised submission:

- upload your program code and input files (if applicable) as a supplement-

- clarify the license under which the reader can use the supplementary program code, see also https://en.wikipedia.org/wiki/Creative_Commons_license (this should be done in the section "Code availability" in your manuscript but it is also recommended to add to the header of any script)

- State in the section "Code availability" :
  - a persistent access to the code (preferable as supplement)
  - license of use
  - the R versions required to run the scripts
  - the R version used to create the results shown in the paper

Many thanks and all the best

Lutz Gross GMD executive editor

---

## Referee Comment (RC1) · A. Oskouie (Referee) · 1 Jul 2017

The work presented here has special significance for researchers in terms of narrowing down their options based on parametrization made in this work. The authors have extensively discussed various variables involved in dry deposition process that looked like a textbook format. The authors should make every effort to shorten the text and eliminate some of the fundamental discussions as related to dry deposition along with the equations. There are some grammatical issues that needs to be addressed; however, it is minimal. The authors should avoid referencing to dry particle deposition, instead they should just mention , dry deposition'. Ambient particle density due to heterogeneity of particulate matter can not be determined and used in these equations properly, so the authors should indicate such uncertainties. The authors should reference to Noll and Oskouie's pioneering work in the field of dry deposition to enrich their work with significant studies made in this field. The uncertainties for ambient particulate density is addressed in Oskouie's work with unique calibration curves developed for determination of density of the ambient particles using supersonic TOF device which is used as the only calibration curves available for such characterization.

Overall this work should be considered for publication with minor revisions.

---

## Referee Comment (RC2) · B. Hicks (Referee) · 3 Aug 2017

This is a well written paper, with conclusions that are important and with which I agree. The introduction gives one of the best overviews of the development of dry deposition formulations that I have come across. The text introduces a new (to me) method for ordering the sensitivity of a model output to the various input properties, and concludes (no surprise) that u* is a key property. Since u* is a measure of the level of turbulence affecting the eddy transport of all atmospheric properties, this is an expected result. However, it leads us further down a path that seems (to me) to have no foreseeable end, since u* is among the most difficult of all atmospheric properties to quantify, both

instrumentally and even more so by using available models or even tabulations.

The following observations arose during my reading of the mss. 1. The first sentence of the introduction could be omitted. 2. Line 33. Suggest omit "accurate." 3. Line 36. "particles" – plural. 4. Line 41. The time scale as stated is that which relates to gravitational settling. It is not appropriate for small particles, for which the relevant time scale needs to take into account the depth of the layer in which the particles are mixed and the various resistances associated with their deposition. In any case, I do not see the need to refer to the time scale here.

5. Line 49. Terrain complexity is a central factor that none of the existing formulations address. 6. Line 57. " . . . there remain . . . " singular. 7. Line 66. There is a recent paper in JGR (lead author – Hicks) that might shed some light on the way in which data have or have not agreed with model predictions. 8. Lines 83-85. My immediate reaction is to wonder why the benefits of multi-variable partial correlation are not mentioned here. This attributes the variance among the contributing factors so that an ordering becomes obvious. The implication of the text is that a single correlation analysis is not appropriate because of the various covariances that could contribute. To my mind, this is precisely what a multiple regression is intended to resolve. 9. Line 96 and onwards. This is the best description of what the models actually assume that I have come across. Congratulations to whoever it was who did the grunt work. Well done! The detail elevates my own fear that the modelers are constructing simulations that reflect articles of faith rather than of evidence. I appreciate that it is comparative examinations like that of the present text that might well shed some informed light on the benefits of model complexity. However, I have yet to see a reason not to start with something simple and add complexities as observations then warranted. 10. I recommend using readily apparent subheadings for each of the model descriptions. 11. Section 3 – Methods. I note that the selection of data requires that each dataset contains measurements of all of the variables whose relevance is IMAGINED by the modelers. I consider this to be rather limiting. Surely, the selection of variables mea-

sured in field studies is a measure of what the experimentalists thought to be important factors. Examination of this might constitute a Delphic approach to determining what is important. I note, also, that requiring measurements of such things as leaf area index (LAI) assumes an application to situations in which a vegetated canopy of some specific kind dominates. For example, it has long been known that the hairs on leaves can play a role. How are these considered in the context of a LAI? 12. The definitions and roles of such properties as the zero plane displacement and the roughness length remain subjects of sometimes heated discussion in meteorological circles. Using these concepts rather cavalierly in dry deposition models seems a poorly justified extrapolation of a very poor basic understanding. I would welcome a more kindly-phrased reflection of this theme somewhere in the present text. 13. Line 345. The wind tunnel studies CANNOT be considered in the same context as open-water deposition. One cannot get 2-m white-capped waves in a wind tunnel. 14. Line 359. " . . . were used . . ." 15. Equations (63) and (64) are no more than expressions that quantify how much the averages differ. Why go into all of the mathematics when the description is so simple. And why not simply call it the amount by which the ratio of the averages differs from unity. (Defining BNMBF seems rather extreme. In practice, the distributions involved are likely to be log-normal or close to it, and hence the arithmetic average is an incorrect concept.) 16. At this point, I encountered Tables 1 and 2. Some comments . . . - What time of day are he numbers meant to represent? - What do the uncertainty ranges mean? If they are meant to refer to long-term ensemble averages base on some other variable, then I would agree with some of them. - The humidity assumption of 80% seems very high. Unless nighttime conditions are assumed. - However, I note that nighttime is indeed assumed: the value of L is positive. A value of 50 m is commonly VERY stable. - I could argue with all of the other "base values," all of which represent the environment that some investigator saw out his window. For example, grass 50 cm tall seems very unlikely. At that stage, it should have been harvested, grazed or mowed. - The use of a one-sided LAI worries me a little. Particles will deposit to both sides of leaves. Why use the one-sided value? - In Table 2, I hope

that the separate listing of ranges for u and u* does not mean that these are allowed to vary independently. The friction velocity is usually from 4% of wind speed (over water) to 20% of the wind speed (over a forest), with the variation of this proportionality being well known. It changes from day to night of course, but it seems that the analysis presented here is only for nighttime. - The variation of L from 10 to 100 m is very constraining, since once again the analysis represents conditions that would often be considered uncommonly stable. - The roughness length quoted as the lower limit for snow/ice is 0.02 mm. As far as I know, this is less that aerodynamically smooth, and is therefore unlikely (some would say impossible). Please be careful. - In Table 3, I note that all of the field programs are for daytime conditions, yet the Base Values tabulated indicate an assumption of stable (nighttime) conditions. Something must be wrong. A suitable explanation would be that the sign for L is incorrect.

Bruce B. Hicks 22 July 2017

---

## Author Comment (AC1) · 30 Aug 2017

**Short Comments Received from GMD Executive Editor**

We thank Lutz Gross for the comments. We have revised the manuscript following his comments. Our responses are compiled below (==*SC: short comment*==, *AR: author's response*==):

==SC:== I would like to suggest the following changes to your manuscript for a revised submission:
- upload your program code and input files (if applicable) as a supplement
- clarify the license under which the reader can use the supplementary program code, see also https://en.wikipedia.org/wiki/Creative_Commons_license (this should be done in the section "Code availability" in your manuscript but it is also recommended to add to the header of any script)
- State in the section "Code availability" :
  - a persistent access to the code (preferable as supplement)
  - license of use
  - the R versions required to run the scripts
  - the R version used to create the results shown in the paper

==AR:== We have included the codes in the supplement. We have reflected this change in the 'code availability' by including the license under which the codes can be used, the *R* version(s) that was used and is required to run the scripts.

The **code availability** section in the revised manuscript now reads:

"The *R* scripts used in this study are available in the supplemental material and can be found online at: https://osf.io/a6q97/. The *R* version 3.2.4 was used to create the scripts. A similar or higher version of *R* is required to run the scripts. The codes can be used, distributed, and reproduced for all non-commercial use under the terms of the Creative Commons Attribution-Noncommercial 3.0 Unported License (http://creativecommons.org/licenses/by-nc/3.0/), provided the original work is properly cited."

---

## Author Comment (AC2) · 30 Aug 2017

**Comments Received from Reviewer # 1**

We thank Ali Oskouie for carefully reviewing our manuscript and providing valuable suggestions. We have revised the manuscript following his comments. Our responses are compiled in the supplement, where we have addressed each comment individually and detailed the respective changes to the manuscript (*RC: reviewer's comment*, *AR: author's response*).

**RC**: The work presented here has special significance for researchers in terms of narrowing down their options based on parametrization made in this work. The authors have extensively discussed various variables involved in dry deposition process that looked like a textbook format. The authors should make every effort to shorten the text and eliminate some of the fundamental discussions as related to dry deposition along with the equations. There are some grammatical issues that needs to be addressed; however, it is minimal.

**AR**: We have omitted the first sentence of the introduction section and former Eq. (1), which was used to express the timescale for particle deposition. However, we think that it is necessary to keep the equations from five dry deposition parameterizations evaluated here in their original forms in the manuscript because they help guide the discussion of the relative complexity of the model parameterizations, which has been addressed in section 5 (lines 837-866) of the manuscript. We believe we have corrected the grammatical issues mentioned by the reviewer.

**RC:** The authors should avoid referencing to dry particle deposition, instead they should just mention 'dry deposition'.

**AR:** In the revised manuscript, after its first use in line 38 as 'dry particle deposition', we referred to it as 'dry deposition' in following texts when it appeared.

**RC:** Ambient particle density due to heterogeneity of particulate matter cannot be determined and used in these equations properly, so the authors should indicate such uncertainties.

**AR:** We understand your concern here. We addressed this issue by adding the following sentences (lines 394-397) in the revised manuscript.

"It is noted that a constant dry particle density of 1500 kg/m$^3$ (Petroff and Zhang, 2010) was used in all Monte Carlo simulations. Because of the inhomogeneous nature of ambient particles, accurate quantification of particle density is challenging. In their work, Oskouie et al. (2003) developed methods using a time-of-flight instrument to minimize the effect of uncertainties in density estimation to particle size characterization."

**RC:** The authors should reference to Noll and Oskouie's pioneering work in the field of dry deposition to enrich their work with significant studies made in this field.

**AR:** Thank you for this reference. We have included the following sentence (lines 51-53) in the revised manuscript.

"Significant advances in understanding the governing mechanisms of dry deposition were made through use of experimental deposition data on walls of vertical pipes to develop size-resolved parameterizations for atmospheric particle deposition on ground surface (Muyshondt et al., 1998; Noll et al., 2001; Feng et al., 2008)."

**RC:** The uncertainties for ambient particulate density is addressed in Oskouie's work with unique calibration curves developed for determination of density of the ambient particles using supersonic TOF device which is used as the only calibration curves available for such characterization.

**AR:** Thanks for pointing out to the references.

**References**

Feng, J.: A size-resolved model and a four-mode parameterization of dry deposition of atmospheric aerosols, Journal of Geophysical Research: Atmospheres, 113, 2008.

Muyshondt, A., Anand, N., and McFarland, A. R.: Turbulent deposition of aerosol particles in large transport tubes, Aerosol Science and Technology, 24, 107-116, 1996.

Noll, K. E., Jackson, M. M., and Oskouie, A. K.: Development of an atmospheric particle dry deposition model, Aerosol Science & Technology, 35, 627-636, 2001.

Oskouie, A. K., Noll, K. E., and Wang, H.-C.: Minimizing the effect of density in determination of particle aerodynamic diameter using a time of flight instrument, Journal of aerosol science, 34, 501-506, 2003.

Petroff, A. and Zhang, L.: Development and validation of a size-resolved particle dry deposition scheme for application in aerosol transport models, Geoscientific Model Development, 3, 753-769, 2010.

---

## Author Comment (AC3) · 30 Aug 2017

**Comments Received from Reviewer #2**

We thank Bruce Hicks for carefully reviewing our manuscript and providing valuable suggestions. We have revised the manuscript following his comments. Our responses are compiled in the supplement, where we have addressed each comment individually and detailed the respective changes to the manuscript (*RC: reviewer's comment, AR: author's response*).

**RC:** This is a well written paper, with conclusions that are important and with which I agree. The introduction gives one of the best overviews of the development of dry deposition formulations that I have come across. The text introduces a new (to me) method for ordering the sensitivity of a model output to the various input properties, and concludes (no surprise) that u* is a key property. Since u* is a measure of the level of turbulence affecting the eddy transport of all atmospheric properties, this is an expected result. However, it leads us further down a path that seems (to me) to have no foreseeable end, since u* is among the most difficult of all atmospheric properties to quantify, both instrumentally and even more so by using available models or even tabulations. The following observations arose during my reading of the mss.

**AR:** We understand and agree with your concern regarding difficulties in measuring friction velocity ($u_*$). We revised the opening paragraph of the conclusion section to reflect on your comment as follows (lines 878-883):

"One could argue that, if the different models exhibited greatest sensitivities to different parameters, and those parameters were more uncertain, the models exhibiting greatest sensitivity to the least certain parameters would be the most uncertain. In this way, sensitivity plays a potential role in determining which model is better. However, because our results showed that all models were most sensitive to $u_*$, or, at large size, *RH*, sensitivity does not end up playing a role in assigning which model is best. We also note that accurate measurement of $u_*$ is extremely challenging (Andreas, 1992; Weber, 1999), and there exists ambiguity in its definition in boundary-layer meteorology (Weber, 1999)."

**RC:** The first sentence of the introduction could be omitted.

**AR:** We have omitted the first sentence.

**RC:** Line 33. Suggest omit "accurate

**AR:** We omitted the word as suggested (line 31) in the revised manuscript.

**RC:** Line 36. "particles" – plural.

**AR:** We have corrected the grammatical error (line 34) in the revised manuscript.

**RC:** Line 41. The time scale as stated is that which relates to gravitational settling. It is not appropriate for small particles, for which the relevant time scale needs to take into account the

depth of the layer in which the particles are mixed and the various resistances associated with their deposition. In any case, I do not see the need to refer to the time scale here.

**AR:** We agree with this comment. The equation (previously denoted as Eq. 1) stating the time scale for dry deposition is now omitted. The subsequent equations are re-numbered.

**RC**: Line 49. Terrain complexity is a central factor that none of the existing formulations Address.

**AR:** We agree that the issue of terrain complexity is not addressed in the five parameterizations we evaluated. To address this limitation, we have included the following sentences (lines 58-62):
"In all these models, the conventional resistance-based approach does not consider surface inhomogeneity or terrain complexity (i.e., deposition over flat terrain is assumed). However, Hicks (2008) argued about the importance of considering terrain complexity in dry deposition models because the assumption of surface homogeneity in existing deposition models limits the accuracy of pollutant load estimation in sensitive ecosystems that are located in complex terrain (e.g., on mountaintops or hills)."

**RC:** Line 57. " . . . there remain . . . " singular

**AR:** We have corrected the grammatical error (line 63) in the revised manuscript.

**RC:** Line 66. There is a recent paper in JGR (lead author – Hicks) that might shed some light on the way in which data have or have not agreed with model predictions.

**AR:** Thanks for pointing out to this reference. We have included the following sentences (lines 70-72):
"In a recent study, Hicks et al. (2016) compared five deposition models with measurements conducted over forests. They found that for particle sizes less than ca. 0.2 µm, the modeled deposition velocities agreed fairly well with measured velocities. The largest discrepancy was observed for particle sizes of 0.3 to ca. 5.0 µm."

**RC:** Lines 83-85. My immediate reaction is to wonder why the benefits of multi-variable partial correlation are not mentioned here. This attributes the variance among the contributing factors so that an ordering becomes obvious. The implication of the text is that a single correlation analysis is not appropriate because of the various covariances that could contribute. To my mind, this is precisely what a multiple regression is intended to resolve.

**AR:** For linear models, multiple variable partial correlation coefficients (PCC) provides the strength of linear relationship between the output and the parameter being tested for its sensitivity while removing the effects from all other input parameters (Brevault et al., 2013). Because the dry deposition models are non-linear, sensitivity analysis using PCC cannot provide accurate ranking of model input parameters in terms of their sensitivity to model outputs.

Rather, in non-linear models, a variance-based global sensitivity test approach such as Sobol' method for sensitivity analysis is more appropriate to achieve accurate parameter ranking and quantify the relative contributions of the input parameters to model outputs.

**RC:** Line 96 and onwards. This is the best description of what the models actually assume that I have come across. Congratulations to whoever it was who did the grunt work. Well done! The detail elevates my own fear that the modelers are constructing simulations that reflect articles of faith rather than of evidence. I appreciate that it is comparative examinations like that of the present text that might well shed some informed light on the benefits of model complexity. However, I have yet to see a reason not to start with something simple and add complexities as observations then warranted.

**AR:** Thanks for this comment.

**RC:** I recommend using readily apparent subheadings for each of the model descriptions.

**AR:** We have re-labeled the subheadings (sections 2.1 to 2.4) for each of the models in the revised manuscript.

**RC:** Section 3 – Methods. I note that the selection of data requires that each dataset contains measurements of all of the variables whose relevance is IMAGINED by the modelers. I consider this to be rather limiting. Surely, the selection of variables measured in field studies is a measure of what the experimentalists thought to be important factors. Examination of this might constitute a Delphic approach to determining what is important. I note, also, that requiring measurements of such things as leaf area index (LAI) assumes an application to situations in which a vegetated canopy of some specific kind dominates. For example, it has long been known that the hairs on leaves can play a role. How are these considered in the context of a LAI?

**AR:** Thank you for this comment. The presence of leaf hairs or trichromes is not considered in the context of LAI in any of the dry deposition models we evaluated in this study. We came across several measurement studies (Rauret et al. 1994; Howsam et al. 2000; Meyers et al. 2006; Burkhardt, 2010) that suggested enhanced particle deposition in the presence of trichomes. In fact, trichomes along with other microroughness elements such as epicuticular wax could increase fine particle deposition rates. We believe that the complexity associated with interactions between particle deposition and leaf microroughness elements poses a major challenge in dry deposition models to account for such interactions in a mechanistic fashion. That said, in our work, we stated that we did not consider the structural uncertainty (that which may arise from inadequate formulation of deposition processes) of the dry deposition models while evaluating their performance (lines 807-808).

**RC:** The definitions and roles of such properties as the zero plane displacement and the roughness length remain subjects of sometime heated discussion in meteorological circles. Using these concepts rather cavalierly in dry deposition models seems a poorly justified

extrapolation of a very poor basic understanding. I would welcome a more kindly-phrased reflection of this theme somewhere in the present text.

**AR:** To address your comment, we have included the following discussion (lines 796-807) in the revised manuscript:

"Additional uncertainties in the modeled deposition velocities could arise from inadequate model formulation and/or inappropriate use of certain micrometeorological parameters. For example, in dry deposition models (such as *PZ10*), $d$ and $z_0$ are often calculated as a fraction of canopy height ($h$), and are often taken as $d \approx 2h/3$ and $z_0 \approx 0.1h$. These expressions are valid for dense canopies (Katul et al., 2010). If the leaf area density is highly skewed or shows a bimodal distribution, such approximations cannot be used (Katul et al., 2010). In addition, the parameter values of $d$ and $z_0$ are subject to large uncertainty and are often very difficult to measure in urban areas (Cherin et al., 2015). Therefore, caution must be taken while using constant $d$ and $z_0$ values from lookup tables. Also, current deposition models do not explicitly treat terrain complexity in their formulations. Hicks (2008) argued that conventional use of $d$ and $z_0$ for non-flat terrain such as mountains is not appropriate in that context. In addition, experimentally derived values of $d$ and $z_0$ often represent local characteristics. Thus, it poses a challenge to scale those up in a model grid cell (Schaudt and Dickinson, 2000) of atmospheric transport models. Using remote sensing, robust scaling of these parameter values is achieved, which could be used to acquire representative values in a model grid cell (Tian et al., 2011)."

**RC:** Line 345. The wind tunnel studies CANNOT be considered in the same context as open water deposition. One cannot get 2-m white-capped waves in a wind tunnel.

**AR:** We agree that the conditions are likely to be different between a wind tunnel and the field in the context of particle deposition over water surfaces. However, in dry deposition models (e.g., Petroff and Zhang, 2010, Zhang et al., 2001), the roughness length ($z_0$) for a water surface is calculated as a function of friction velocity and/or wind speed. The empirical expressions commonly used to estimate $z_0$ for water surfaces were developed based on laboratory or wind-wave tunnel experiments (Uneo and Deushi, 2003). For example, in the Petroff and Zhang, 2010 (*PZ10*) model, the $z_0$ was parameterized using an expression based on a wind-profile experiment conducted on a laboratory reservoir by Charnock (1955). In Charnock's expression, the roughness length does not depend on the wave state parameters such spectral peak frequency, wave steepness, and wave age (Uneo and Deushi, 2003). In our work, we used the formulations given by Keller and Keller (1992) to calculate roughness length for water surfaces, which were developed based on experiments conducted on a wave-tank. In their formulation, Keller and Keller (1992) also did not relate $z_0$ to wave state parameters (e.g., spectral peak frequency). They compared the wave-tank derived $z_0$ values with those obtained through open ocean momentum flux measurements conducted by Large and Pond (1981) and found good agreement. Given these reasons, we have included the wind-tunnel measurements in our study.

We changed the following sentence as "We note that the studies by Moller and Schumann, and Sehmel et al. were conducted in the wind tunnels, and thus the observed deposition does not (replaced *may not* with *does not*) necessarily reflect deposition under natural conditions" in the revised manuscript (line 361).

RC: Line 359. " . . . were used . . ."

AR: We have made the suggested correction (line 366) in the revised manuscript.

RC: Equations (63) and (64) are no more than expressions that quantify how much the averages differ. Why go into all of the mathematics when the description is so simple. And why not simply call it the amount by which the ratio of the averages differs from unity. (Defining BNMBF seems rather extreme. In practice, the distributions involved are likely to be log-normal or close to it, and hence the arithmetic average is an incorrect concept.)

AR: We have expressed these equations (now re-numbered as Eqs. 64-65) in their basic (simplified) forms. We have included a sentence to clarify the meaning of $B_{NMFB}$ (lines 370-371) as suggested.

We agree that the distribution of deposition velocity ($V_d$) for a wide range of particle sizes (e.g., 0.01 -100 μm) should exhibit a log-normal distribution (e.g., Figure 7 in Hicks et al., 2016). However, in most of the measurement studies used to evaluate model performance in our study, particle sizes ($d_p$) were confined to from 0.01 to 0.80 μm, while the number of data points for $d_p > 1.0$ μm (column 10 in Table 1) were minimal. Therefore, we think using $B_{NMFB}$ is a reasonable approach to quantify the accuracy of the models.

RC: At this point, I encountered Tables 1 and 2. Some comments . . .

RC: What time of day are the numbers meant to represent?

AR: We are aware that the base values of certain parameters (e.g., relative humidity (*RH*), and Monin-Obukhov length ($L_O$)) could exhibit more diel variability than some other parameters listed in Table 2. To provide a basis for relative comparison, the base parameter values listed in Table 2 were used to perform uncertainty analysis, with the objective of demonstrating the overall uncertainties in modeled deposition velocities in the five parameterizations across five LUCs. These base values used for uncertainty analysis should be viewed as reasonable, arbitrary choices within the range of the reported values of those parameters in the measurement studies listed in Table 1. Here, we did not explicitly differentiate between daytime vs. nighttime conditions.

RC: What do the uncertainty ranges mean? If they are meant to refer to long-term ensemble averages base on some other variable, then I would agree with some of them.

AR: Each uncertainty range denotes the degree of imprecision (in percentage) associated with each of corresponding parameters listed in Table 2. To avoid confusion, we have referred to it

as 'uncertainty' in Table 2 in the revised manuscript. The uncertainty values (in percentage) were obtained from a literature search and the references are listed in column 3 of Table 2.

RC: The humidity assumption of 80% seems very high. Unless nighttime conditions are assumed.

AR: In several measurement studies (e.g., Neumann and den Hartog, 1985; Ruijgrok et al., 1997; Vong et al., 2004; Grönholm et al., 2009), particle deposition was measured at both daytime and nighttime. The threshold for very high humidity ($RH$) was reported as >95% (e.g., Ruijgrok et al., 1997) in the context of their measurement study. In addition, the ranges of $RH$ were reported as 35-100% and 49-90% by Grönholm et al. (2009) and Vong et al. (2004), respectively. The choice of $RH$ = 80% was arbitrary, but was within the reported range of $RH$ by the measurement studies listed in Table 1. In addition, at high $RH$ (e.g., $RH$ close to 90%), atmospheric particles are likely to grow (the phenomena commonly known as hygroscopic growth) to twice their initial dry diameter (Vong et al., 2004). The hygroscopic growth was accounted for in the models according to Eq. (15).

RC: However, I note that nighttime is indeed assumed: the value of L is positive. A value of 50 m is commonly VERY stable.

AR: We understand your concern regarding the choice of Monin-Obukhov length ($L_O$) value as 50 m for the uncertainty test. In this study, we denoted uncertainties in the modeled $V_d$ in normalized terms (i.e., the difference between the 95$^{th}$ and 5$^{th}$ percentiles $V_d$ was divided by the 50$^{th}$ percentile $V_d$) rather than in absolute terms (i.e., the difference between the 95$^{th}$ and 5$^{th}$ percentiles $V_d$ or the central 90% values of $V_d$). Therefore, the choice of $L_O$ has minimal effect on the normalized uncertainties reported in Table 10.

RC: I could argue with all of the other "base values," all of which represent the environment that some investigator saw out his window. For example, grass 50 cm tall seems very unlikely. At that stage, it should have been harvested, grazed or mowed.

AR: The measurement studies used for model evaluation on grass reported typical values of grass height as 30-60 cm (e.g., Neumann and den Hartog, 1985; Vong et al., 2004). We agree that grass height of 50 cm could be higher than average height, but it was not uncommon in particle deposition measurement studies. For example, an article by Wesely et al. (1985) (not used in study) reported sulfur deposition on grass with height ranging from 40-50 cm.

RC: The use of a one-sided LAI worries me a little. Particles will deposit to both sides of leaves. Why use the one-sided value?

AR: We used one-sided LAI because atmospheric transport models (e.g., GEOS-Chem) regularly use MODIS derived LAI values as inputs to the dry deposition models. According to MODIS LAI database (https://modis.gsfc.nasa.gov/data/dataprod/mod15.php), "LAI is defined as one-sided green leaf area per unit ground area in broadleaf canopies and as half the total needle surface

area per unit ground area in coniferous canopies." In addition, reported values of LAI in the measurement studies (listed in Table 1) over vegetative surfaces often do not differentiate between two vs. one-sided LAI. Thus, to avoid ambiguity, we used one-sided LAI.

RC: In Table 2, I hope that the separate listing of ranges for u and u* does not mean that these are allowed to vary independently. The friction velocity is usually from 4% of wind speed (over water) to 20% of the wind speed (over a forest), with the variation of this proportionality being well known. It changes from day to night of course, but it seems that the analysis presented here is only for nighttime.

AR: The separate uncertainty ranges (now, for clarity, denoted as 'uncertainty' in Table 2, as compared to 'uncertainty range' in the previous manuscript) for wind speed ($U$) and friction velocity ($u_*$) denote respective imprecisions associated with those two parameters. In the uncertainty analysis presented in this paper, we used these two parameters independent of each other by allowing them to vary only within their specified imprecision/uncertainty ranges. The objective of the uncertainty analysis was to quantify the effects of imprecision in measured values of the input parameters (e.g., $U$, $u_*$, $RH$, $L_O$, etc.) to the modeled $V_d$. Thus, we did not explicitly distinguish between daytime and nighttime periods.

RC: The variation of L from 10 to 100 m is very constraining, since once again the analysis represents conditions that would often be considered uncommonly stable.

AR: The analysis presented here assumed near-neutral conditions. The range (10 to 100 m) for $L_O$ (from stable side) was consistent with reported range for neutral conditions ($-200 < L_O < 200$) by Pryor et al. (2007) (study #13 in Table 1). In addition, we did not find any change in parameter rankings reported in Table 11 by increasing the upper bound value of $L_O$ (in the assumed range of 10-100 m) by factors of 10 to 20.

RC: The roughness length quoted as the lower limit for snow/ice is 0.02 mm. As far as I know, this is less that aerodynamically smooth, and is therefore unlikely (some would say impossible). Please be careful.

AR: The lower limit of the roughness length ($z_0$) is consistent with the reported $z_0$ values in the measurement studies of particle deposition on ice/snow (e.g., Fig. 2a in Held et al., 2001a (study#24 in Table 1)). In addition, the article by Tjernström (2005) reported the lower bound $z_0$ value as 0.01 mm from Arctic Ocean Experiment of 2001.

RC: In Table 3, I note that all of the field programs are for daytime conditions, yet the Base Values tabulated indicate an assumption of stable (nighttime) conditions. Something must be wrong. A suitable explanation would be that the sign for L is incorrect.

AR: In several measurement studies listed in Table 1 (e.g., Neumann and den Hartog, 1985; Ruijgrok et al., 1997; Vong et al., 2004; Grönholm et al., 2009), particle deposition was

measured under both daytime and nighttime conditions. The analysis presented here assumed stable conditions, thus the sign for $L_O$ is correct.

**References**

Brevault, L., Balesdent, M., Bérend, N., and Le Riche, R.: Comparison of different global sensitivity analysis methods for aerospace vehicle optimal design, 2013.

Burkhardt, J.: Hygroscopic particles on leaves: nutrients or desiccants?, Ecological Monographs, 80, 369-399, 2010.

Charnock, H.: Wind stress on a water surface, Quarterly Journal of the Royal Meteorological Society, 81, 639-640, 1955.

Cherin, N., Roustan, Y., Musson-Genon, L., and Seigneur, C.: Modelling atmospheric dry deposition in urban areas using an urban canopy approach, Geoscientific Model Development, 8, 893-910, 2015.

Grönholm, T., Launiainen, S., Ahlm, L., Mårtensson, E., Kulmala, M., Vesala, T., and Nilsson, E.: Aerosol particle dry deposition to canopy and forest floor measured by two-layer eddy covariance system, Journal of Geophysical Research: Atmospheres, 114, 2009.

Hicks, B. B.: On estimating dry deposition rates in complex terrain, Journal of Applied Meteorology and Climatology, 47, 1651-1658, 2008.

Hicks, B. B., Saylor, R. D., and Baker, B. D.: Dry deposition of particles to canopies—A look back and the road forward, Journal of Geophysical Research: Atmospheres, 121, 2016.

Howsam, M., Jones, K. C., and Ineson, P.: PAHs associated with the leaves of three deciduous tree species. I—concentrations and profiles, Environmental pollution, 108, 413-424, 2000.

Katul, G., Grönholm, T., Launiainen, S., and Vesala, T.: Predicting the dry deposition of aerosol-sized particles using layer-resolved canopy and pipe flow analogy models: Role of turbophoresis, Journal of Geophysical Research: Atmospheres, 115, 2010.

Keller, M. R., Keller, W. C., and Plant, W. J.: A wave tank study of the dependence of X band cross sections on wind speed and water temperature, Journal of Geophysical Research: Oceans, 97, 5771-5792, 1992.

Large, W. and Pond, S.: Open ocean momentum flux measurements in moderate to strong winds, Journal of physical oceanography, 11, 324-336, 1981.

Meyers, T., Luke, W., and Meisinger, J.: Fluxes of ammonia and sulfate over maize using relaxed eddy accumulation, Agricultural and forest meteorology, 136, 203-213, 2006.

Neumann, H. and Den Hartog, G.: Eddy correlation measurements of atmospheric fluxes of ozone, sulphur, and particulates during the Champaign intercomparison study, Journal of Geophysical Research: Atmospheres, 90, 2097-2110, 1985.

Petroff, A. and Zhang, L.: Development and validation of a size-resolved particle dry deposition scheme for application in aerosol transport models, Geoscientific Model Development, 3, 753-769, 2010.

Pryor, S., Larsen, S. E., Sørensen, L. L., Barthelmie, R. J., Grönholm, T., Kulmala, M., Launiainen, S., Rannik, Ü., and Vesala, T.: Particle fluxes over forests: Analyses of flux methods and functional dependencies, Journal of Geophysical Research: Atmospheres, 112, 2007.

Rauret, G., Llauradó, M., Tent, J., Rigol, A., Alegre, L., and Utrillas, M.: Deposition on holm oak leaf surfaces of accidentally released radionuclides, Science of the total environment, 157, 7-16, 1994.

Ruijgrok, W., Tieben, H., and Eisinga, P.: The dry deposition of particles to a forest canopy: a comparison of model and experimental results, Atmospheric Environment, 31, 399-415, 1997.

Schaudt, K. and Dickinson, R.: An approach to deriving roughness length and zero-plane displacement height from satellite data, prototyped with BOREAS data, Agricultural and Forest Meteorology, 104, 143-155, 2000.

Tian, X., Li, Z., Van der Tol, C., Su, Z., Li, X., He, Q., Bao, Y., Chen, E., and Li, L.: Estimating zero-plane displacement height and aerodynamic roughness length using synthesis of LiDAR and SPOT-5 data, Remote sensing of environment, 115, 2330-2341, 2011.

Tjernström, M.: The summer Arctic boundary layer during the Arctic Ocean Experiment 2001 (AOE-2001), Boundary-Layer Meteorology, 117, 5-36, 2005.

Ueno, K. and Deushi, M.: A new empirical formula for the aerodynamic roughness of water surface waves, Journal of oceanography, 59, 819-831, 2003.

Vong, R. J., Vickers, D., and Covert, D. S.: Eddy correlation measurements of aerosol deposition to short grass, Tellus B, 56, 105-117, 2004.

Weber, R. O.: Remarks on the definition and estimation of friction velocity, Boundary-Layer Meteorology, 93, 197-209, 1999.

Wesely, M., Cook, D., Hart, R., and Speer, R.: Measurements and parameterization of particulate sulfur dry deposition over grass, Journal of Geophysical Research: Atmospheres, 90, 2131-2143, 1985.

---

## Author Comment (AC4) · 30 Aug 2017

[revised manuscript text omitted]
 | 0.000 | 0.000 | 0.001 | 0.027 | 0.022 | 0.032 | 0.000 | 0.000 | 0.000 | 0.000 | 0.000 | 0.000 | 0.000 | 0.000 | 3.180 |

**Codes for evaluation of model accuracy using Zhang et al. (2001) parameterization**

```
**Accuracy Evaluation: Grass**
**Dry deposition parameterization by Zhang et al. (2001)**
attach(Allen_etal_1991)                              # Use separate text file to feed V1, V2,… for different studies
C1 = 0.2789
C1 = 0.2789
C2 = 3.115
C3 = 5.145*10^-11
RH = 90/100
dp_i = 0.48
dp_d = dp_i*10^-6
rd = dp_d/2
r_w = {(C1*rd^C2)/(C3*rd^C4-log10(RH))+rd^3}^(1/3)
dp = r_w*2
**Correction factor, C**
k_B = 1.38*10^-23
Temp_1 = V1
Temp = 273.15+V1
P = 101325
d_air = 3.72*10^-10
lambda = (k_B*Temp)/(sqrt(2)*3.1416*P*d_air^2)
C = 1+(2*lambda/dp)*(1.257+0.4*exp(-0.55*dp/lambda))
dyn.vis = ((5*10^-8)*Temp)+4*10^-6
rho = 2000
Vg = (rho*(dp)^2*9.81*C)/(18*dyn.vis)
**Compute aerodynamic resistance Ra:**
z = 2
L = V4
x = z/L
**Compute shi_H (stability function)**
shi_H.1 = 2*log(0.5*{1+(1-16*x)^0.5})
shi_H.2 = -5*x
shi_H =ifelse(x <= 0, shi_H.1 , shi_H.2)
zR = 3.5
z0 = 0.01
u_star = V2                                          #Use from input text file
k_c = 0.41
Ra = (log(zR/z0)-shi_H)/(k_c*u_star)
**Compute surface resistance Rs:**
e_0 = 3
R1 = 1
**Compute E_B (collection efficiency from Brownian diffusion)**
kin.vis = ((9*10^-8)*Temp)+10^-5
gamma = 0.54
D = (C*k_B*Temp)/(3*3.1416*dyn.vis*dp)
Sc = (kin.vis/D)
E_B = Sc^(-gamma)
**Compute E_IM (collection efficiency from impaction)**
alpha = 1.2
beta =  2
A = 2/1000
St = (Vg*u_star)/(9.81*A)
E_IM = {St/(alpha+St)}^beta
**Compute E_IN (collection efficiency from interception)**
E_IN = 0.5*(dp/A)^2
Rs = 1/{(e_0*u_star)*(E_B+E_IM+E_IN)*R1}
**Compute Dry deposition velocity**

**Accuracy Evaluation: Coniferous forest**
**Dry deposition parameterization by Zhang et al. (2001)**
attach(Rannik_etal_2000)                             # Use separate text file to feed V1, V2,… for different studies
C1 = 0.2789
C2 = 3.115
C3 = 5.145*10^-11
C4 = -1.399
RH = 90/100
dp_i = V1
dp_d = dp_i*10^-6
rd = dp_d/2
r_w = {(C1*rd^C2)/(C3*rd^C4-log10(RH))+rd^3}^(1/3)
dp = r_w*2
**Correction factor, C**
k_B = 1.38*10^-23
```

```
Temp = 273.15+25
P = 101325
d_air = 3.72*10^-10
lambda = (k_B*Temp)/(sqrt(2)*3.1416*P*d_air^2)
C = 1+(2*lambda/dp)*(1.257+0.4*exp(-0.55*dp/lambda))
dyn.vis = 1.891*10^-5
rho = 1500
Vg = (rho*(dp)^2*9.81*C)/(18*dyn.vis)
**Compute aerodynamic resistance Ra:**
z = 23.7
L = 200
x = z/L
shi_H2 = -5*x
zR = 26
z0 = 1.2
u_star = V2
k_c = 0.41
Ra = (log(zR/z0)-shi_H2)/(k_c*u_star)
e_0 = 3
R1 =1
kin.vis = 1.683*10^-5
gamma = 0.56
D = (C*k_B*Temp)/(3*3.1416*dyn.vis*dp)
Sc = (kin.vis/D)
E_B = Sc^(-gamma)
**Compute E_IM (collection efficiency from impaction)**
alpha = 1.0
beta =  2
A = 5/1000
St = (Vg*u_star)/(9.81*A)
E_IM = {St/(alpha+St)}^beta
**Compute E_IN (collection efficiency from interception)**
E_IN = 0.5*(dp/A)^2
Rs = 1/{(e_0*u_star)*(E_B+E_IM+E_IN)*R1}
**Compute Dry deposition velocity**
Vd =  Vg+(1/(Ra+Rs));Vd   #unit: m/s

**Accuracy Evaluation: Deciduous forest**
**Dry deposition parameterization by Zhang et al. (2001)**
 attach(Wesely_etal_1983)                        # Use separate text file to feed V1, V2,… for different studies
C1 = 0.2789
C2 = 3.115
C3 = 5.145*10^-11
C4 = -1.399
RH = 95/100
dp_i = 0.4
dp_d = dp_i*10^-6
rd = dp_d/2
r_w = {(C1*rd^C2)/(C3*rd^C4-log10(RH))+rd^3}^(1/3)
dp = r_w*2
**Correction factor, C**
k_B = 1.38*10^-23
Temp = V1
P = 101325
d_air = 3.72*10^-10
lambda = (k_B*Temp)/(sqrt(2)*3.1416*P*d_air^2)
C = 1+(2*lambda/dp)*(1.257+0.4*exp(-0.55*dp/lambda))
dyn.vis = V2
rho = 2000
Vg = (rho*(dp)^2*9.81*C)/(18*dyn.vis)
**Compute aerodynamic resistance Ra:**
z = 39
L = V6
x = z/L
shi_H2 = -5*x
zR = 56
z0 = 1.6
u_star = V4
k_c = 0.41
Ra = (log(zR/z0)-shi_H1)/(k_c*u_star)
**Compute surface resistance Rs:**
e_0 = 3
R1 =1
kin.vis = V3
```

```
gamma = 0.56
D = (C*k_B*Temp)/(3*3.1416*dyn.vis*dp)
Sc = (kin.vis/D)
E_B = Sc^(-gamma)
**Compute E_IM (collection efficiency from impaction)**
alpha = 0.80
beta =  2
A = 5/1000
St = (Vg*u_star)/(9.81*A)
E_IM = {St/(alpha+St)}^beta
**Compute E_IN (collection efficiency from interception)**
E_IN = 0.5*(dp/A)^2
Rs = 1/{(e_0*u_star)*(E_B+E_IM+E_IN)*R1}
**Compute Dry deposition velocity**
Vd =   Vg+(1/(Ra+Rs));Vd

**Accuracy Evaluation: Water**
**Dry deposition parameterization by Zhang et al. (2001)**
attach(Caffrey_etal_1998)
C1 = 0.2789
C2 = 3.115
C3 = 5.415*10^-11
C4 = -1.399
RH = 79/100
dp_i = 0.005
dp_d = dp_i*10^-6
rd = dp_d/2
r_w = {(C1*rd^C2)/(C3*rd^C4-log10(RH))+rd^3}^(1/3)
dp = r_w*2
**Correction factor, C**
k_B = 1.38*10^-23
Temp = 273.15+22
P = 101325
d_air = 3.72*10^-10
lambda = (k_B*Temp)/(sqrt(2)*3.1416*P*d_air^2)
C = 1+(2*lambda/dp)*(1.257+0.4*exp(-0.55*dp/lambda))
dyn.vis = ((5*10^-8)*Temp)+4*10^-6
rho = 2000
Vg = (rho*(dp)^2*9.81*C)/(18*dyn.vis)
**Compute aerodynamic resistance Ra:**
z = 8/100
L = 50
x = z/L
**Compute shi_H (stability function)**
shi_H.1 = 2*log(0.5*{1+(1-16*x)^0.5})
shi_H.2 = -5*x
shi_H =ifelse(x <= 0, shi_H.1 , shi_H.2)
zR = 5
u_star = 13.5/100
z0_1 = 0.021*(u_star)^3.32
z0_2 = 0.00098*(u_star)^1.65
z0 = ifelse(u_star<= 0.16, z0_1, z0_2)
k_c = 0.41
Ra = (log(zR/z0)-shi_H)/(k_c*u_star)  # m/s
**Compute surface resistance Rs:**
e_0 = 3
R1 = 1
kin.vis = ((9*10^-8)*Temp)+10^-5
gamma = 0.50
D = (C*k_B*Temp)/(3*3.1416*dyn.vis*dp)
Sc = (kin.vis/D)
E_B = Sc^(-gamma)
alpha = 100
beta =  2
A = 2/1000
St = (Vg*u_star^2)/(kin.vis)
E_IM = {St/(alpha+St)}^beta
Rs = 1/{(e_0*u_star)*(E_B+E_IM)*R1}
**Compute Dry deposition velocity**
Vd =   Vg+(1/(Ra+Rs));Vd
```

```
**Accuracy Evaluation: Ice/snow**
**Dry deposition parameterization by Zhang et al. (2001)**
attach(Ibrahim_1983)
C1 = 0.2789
C2 = 3.115
C3 = 5.415*10^-11
C4 = -1.399
RH = 60/100
dp_i = c(0.22, 0.73)
dp_d = dp_i*10^-6
rd = dp_d/2
r_w = {(C1*rd^C2)/(C3*rd^C4-log10(RH))+rd^3}^(1/3)
dp = r_w*2
**Correction factor, C**
k_B = 1.38*10^-23
Temp = 273.15+25
P = 101325
d_air = 3.72*10^-10
lambda = (k_B*Temp)/(sqrt(2)*3.1416*P*d_air^2)
C = 1+(2*lambda/dp)*(1.257+0.4*exp(-0.55*dp/lambda))
dyn.vis = ((5*10^-8)*Temp)+4*10^-6
rho = 1500
Vg = (rho*(dp)^2*9.81*C)/(18*dyn.vis)
**Compute aerodynamic resistance Ra:**
**z =**
**L =**
**x = z/L**
x = 0.2
**Compute shi_H (stability function)**
shi_H.1 = 2*log(0.5*{1+(1-16*x)^0.5})
shi_H.2 = -5*x
shi_H =ifelse(x <= 0, shi_H.1 , shi_H.2)
zR = 5
u_star = 0.12
z0 = 0.1/100
k_c = 0.41
Ra = (log(zR/z0)-shi_H)/(k_c*u_star)
**Compute surface resistance Rs:**
e_0 = 3
R1 = 1
kin.vis = ((9*10^-8)*Temp)+10^-5
gamma = 0.54
D = (C*k_B*Temp)/(3*3.1416*dyn.vis*dp)
Sc = (kin.vis/D)
E_B = Sc^(-gamma)
alpha = 50
beta =  2
A = 2/1000
St = (Vg*u_star^2)/(kin.vis)
E_IM = {St/(alpha+St)}^beta
Rs = 1/{(e_0*u_star)*(E_B+E_IM)*R1}
**Compute Dry deposition velocity**
Vd =  Vg+(1/(Ra+Rs));Vd
```

**Codes for evaluation of model accuracy using Petroff and Zhang (2010) parameterization**

```
**Accuracy Evaluation: Grass**
**Dry deposition velocity parameterization by Petroff and Zhang (2010)**
attach(Allen_etal_1991)
C1 = 0.2789
C2 = 3.115
C3 = 5.145*10^-11
C4 = -1.399
RH = 90/100
dp_i = 0.48
dp_d = dp_i*10^-6
rd = dp_d/2
r_w = {(C1*rd^C2)/(C3*rd^C4-log10(RH))+rd^3}^(1/3)
dp = r_w*2
**Correction factor, C**
k_B = 1.38*10^-23
Temp = 273.15+25
P = 101325
```

```
d_air = 3.7208*10^-10
lambda = (k_B*Temp)/(sqrt(2)*3.1416*P*d_air^2)
C = 1+(2*lambda/dp)*(1.257+0.4*exp(-0.55*dp/lambda))
dyn.vis = ((5*10^-8)*Temp)+4*10^-6
rho = 1500
Tau = (rho*(dp)^2*C)/(18*dyn.vis)
Ws = Tau*9.81
Vphor = 0
Vdrift = Ws+Vphor
**Compute aerodynamic resistance (Ra):**
z = 2
L = 200
x = z/L
**Compute stability function (shi_H)**
shi_H.1 = 2*log(0.5*{1+(1-16*x)^0.5})
shi_H.2 = -5*x
shi_H = ifelse(x <= 0, shi_H.1 , shi_H.2)
zR = 3.5
z0 = 0.01
u_star = 0.5
k_c = 0.41
Ra = (log(zR/z0)-shi_H)/(k_c*u_star)
**Compute surface resistance (Rs)**
kin.vis = ((9*10^-8)*Temp)+10^-5
D = (C*k_B*Temp)/(3*3.1416*dyn.vis*dp)
Sc = (kin.vis/D)
FSc = (Sc^(1/3))/2.9
Egb = (Sc^(-2/3)/14.5)*{1/6*log(1+FSc)^2/(1-FSc+FSc^2)+1/sqrt(3)*atan((2*FSc-
1)/sqrt(3))+3.1416/6*sqrt(3)}^-1
cd = 1/6
kx = 0.216
LAI = 4
h =  0.07
d  = 0.04
phi_H.1 = (1-16*x)^(-0.5)
phi_H.2 = 1+5*x
phi_H = ifelse(x<=0, phi_H.1, phi_H.2)
phi_M.1 = (1-16*x)^(-0.25)
phi_M.2 = 1+5*x
phi_M = ifelse(x<=0, phi_M.1, phi_M.2)
lmp = (0.41*(z-d))/(phi_H*(z-d)/abs(L))
lmp_h = (0.41*(h-d))/(phi_M*(h-d)/abs(L))
alphaPZ = {(kx*LAI)/(12*k_c^2*(1-d/h)^2)}^(1/3)*phi_M^(2/3)*{(h-d)/abs(L)}
C_IT = 0.056
Tau_phplus.1 = (Tau*u_star^2)/kin.vis
Tau_phplus.2 = C_IT
Tau_phplus = ifelse(Tau_phplus.1<20,Tau_phplus.1, Tau_phplus.2)
E_t.1 = 2.5*10^-3*C_IT*(Tau_phplus)^2
E_t.2 = C_IT
E_t = ifelse(Tau_phplus.1<20, E_t.1, E_t.2)
u_starf = u_star*exp(-alphaPZ)
Tau_phplus.f1 = (Tau*u_star^2)/kin.vis
Tau_phplus.f2 = 0.14
Tau_phplusf = ifelse(Tau_phplus.f1<20,Tau_phplus.f1, Tau_phplus.f2)
E_gt1 = 2.5*10^-3*0.14*(Tau_phplusf)^2
E_gt2 = 0.14
Egt = ifelse(Tau_phplus.f1<20, E_gt1, E_gt2)
Eg = Egb + Egt
**Compute Qg (non-dimensional number)**
Qg = Eg*h/lmp_h
**Compute Q**
U_z = 2
U_h = U_z/(exp(alphaPZ*(z/h-1)))
**Compute E_B (Brownian diffusion)**
L_obs = 0.01
C_B = 0.996
Re_h = (U_h*L_obs)/(kin.vis)
E_B = C_B*(Sc^(-2/3))*(Re_h^(-1/2))
**Compute E_IN (Interception)**
C_IN = 0.162
E_IN = C_IN*(dp/L_obs)
**Compute E_IM (Impaction)**
C_IM = 0.081
beta_IM = 0.47
```

```
St_h = (Tau*U_h)/L_obs
E_IM = C_IM*(St_h/(St_h+beta_IM))^2
E_T = (U_h/u_star)*(E_B+E_IN+E_IM)+E_t
Q   = LAI*E_T*h/(lmp_h)
**Compute etaPZ**
etaPZ = (alphaPZ^2/4+Q)^0.5
Vds = u_star*Eg*{(1+Q/Qg-alphaPZ/2)*tanh(etaPZ)/etaPZ}/{(1+Q+alphaPZ/2)*tanh(etaPZ)/etaPZ}
Vd = Vdrift+1/(Ra+1/Vds);Vd

**Accuracy Evaluation: Coniferous forest**
**Dry deposition velocity parameterization by Petroff and Zhang (2010)**
attach(Rannik_etal_2000)
C1 = 0.2789
C2 = 3.115
C3 = 5.145*10^-11
C4 = -1.399
RH= 0.90
dp_i = V1
dp_d = dp_i*10^-6
rd = dp_d/2
r_w = {(C1*rd^C2)/(C3*rd^C4-log10(RH))+rd^3}^(1/3)
dp = r_w*2
**Correction factor, C**
k_B = 1.38*10^-23
Temp =   273.15+25
P = 101325
d_air = 3.7208*10^-10
lambda = (k_B*Temp)/(sqrt(2)*3.1416*P*d_air^2)
C = 1+(2*lambda/dp)*(1.257+0.4*exp(-0.55*dp/lambda))
dyn.vis = 1.891*10^-5
rho = 1500
Tau = (rho*(dp)^2*C)/(18*dyn.vis)
Ws = Tau*9.81
Vphor = 0
Vdrift = Ws+Vphor
**Compute aerodynamic resistance (Ra):**
z = 23.7
L = 200
x = z/L
**Compute stability function (shi_H)**
shi_H.1 = 2*log(0.5*{1+(1-16*x)^0.5})
shi_H.2 = -5*x
shi_H =ifelse(x <= 0, shi_H.1 , shi_H.2)
zR = 26
z0 = 1.2
u_star = V2
k_c = 0.41
Ra = (log(zR/z0)-shi_H)/(k_c*u_star)
kin.vis = 1.683*10^-5
D = (C*k_B*Temp)/(3*3.1416*dyn.vis*dp)
Sc = (kin.vis/D)
FSc = (Sc^(1/3))/2.9
Egb = (Sc^(-2/3)/14.5)*{1/6*log(1+FSc)^2/(1-FSc+FSc^2)+1/sqrt(3)*atan((2*FSc-1)/sqrt(3))
        +3.1416/6*sqrt(3)}^-1
cd = 1/6
kx = 0.216
LAI = 6
h =   13
d   = 9.75
phi_H.1 = (1-16*x)^(-0.5)
phi_H.2 = 1+5*x
phi_H = ifelse(x<=0, phi_H.1, phi_H.2)
phi_M.1 = (1-16*x)^(-0.25)
phi_M.2 = 1+5*x
phi_M = ifelse(x<=0, phi_M.1, phi_M.2)
lmp = (0.41*(z-d))/(phi_H*(z-d)/abs(L))
lmp_h = (0.41*(h-d))/(phi_M*(h-d)/abs(L))
alphaPZ = {(kx*LAI)/(12*k_c^2*(1-d/h)^2)}^(1/3)*phi_M^(2/3)*{(h-d)/abs(L)}
C_IT = 0
Tau_phplus.1 = (Tau*u_star^2)/kin.vis
Tau_phplus.2 = C_IT
Tau_phplus = ifelse(Tau_phplus.1<20,Tau_phplus.1, Tau_phplus.2)
E_t.1 = 2.5*10^-3*C_IT*(Tau_phplus)^2
```

```
E_t.2 = C_IT
E_t = ifelse(Tau_phplus.1<20, E_t.1, E_t.2)
u_starf = u_star*exp(-alphaPZ)
Tau_phplus.f1 = (Tau*u_star^2)/kin.vis
Tau_phplus.f2 = 0.14
Tau_phplusf = ifelse(Tau_phplus.f1<20,Tau_phplus.f1, Tau_phplus.f2)
E_gt1 = 2.5*10^-3*0.14*(Tau_phplusf)^2
E_gt2 = 0.14
Egt = ifelse(Tau_phplus.f1<20, E_gt1, E_gt2)
Eg = Egb + Egt
**Compute Qg (non-dimensional number)**
Qg = Eg*h/lmp_h
**Compute Q**
U_z = V3
U_h = U_z/(exp(alphaPZ*(z/h-1)))
L_obs = 0.15
C_B = 0.887
Re_h = (U_h*L_obs)/(kin.vis)
E_B = C_B*(Sc^(-2/3))*(Re_h^(-1/2))
C_IN = 0.810
E_IN = C_IN*(dp/L_obs)
**Compute E_IM (Impaction)**
C_IM = 0.162
beta_IM = 0.60
St_h = (Tau*U_h)/L_obs
E_IM = C_IM*(St_h/(St_h+beta_IM))^2
E_T = (U_h/u_star)*(E_B+E_IN+E_IM)+E_t
Q   = LAI*E_T*h/(lmp_h)
**Compute etaPZ**
etaPZ = (alphaPZ^2/4+Q)^0.5
Vds = u_star*Eg*{(1+Q/Qg-alphaPZ/2)*tanh(etaPZ)/etaPZ}/{(1+Q+alphaPZ/2)*tanh(etaPZ)/etaPZ}
Vd = Vdrift+1/(Ra+1/Vds);Vd

**Accuracy Evaluation: Deciduous forest**
**Dry deposition parameterization by Petroff and Zhang (2010)**
attach(Wesely_etal_1983)
C1 = 0.2789
C2 = 3.115
C3 = 5.145*10^-11
C4 = -1.399
RH= 95/100
dp_i = 0.4
dp_d = dp_i*10^-6
rd = dp_d/2
r_w = {(C1*rd^C2)/(C3*rd^C4-log10(RH))+rd^3}^(1/3)
dp = r_w*2
k_B = 1.38*10^-23
Temp = V1
P = 101325
d_air = 3.7208*10^-10
lambda = (k_B*Temp)/(sqrt(2)*3.1416*P*d_air^2)
C = 1+(2*lambda/dp)*(1.257+0.4*exp(-0.55*dp/lambda))
dyn.vis = V2
rho = 2000
Tau = (rho*(dp)^2*C)/(18*dyn.vis)
Ws = Tau*9.81
Vphor = 0
Vdrift = Ws+Vphor
**Compute aerodynamic resistance (Ra):**
z = 39
L = (1*V6)
x = z/L
**Compute stability function (shi_H)**
shi_H.1 = 2*log(0.5*{1+(1-16*x)^0.5})
shi_H.2 = -5*x
shi_H =ifelse(x <= 0, shi_H.1 , shi_H.2)
zR = 56
z0 = 1.6
u_star = V4
k_c = 0.41
Ra = (log(zR/z0)-shi_H)/(k_c*u_star)
kin.vis = V3
D = (C*k_B*Temp)/(3*3.1416*dyn.vis*dp)
```

```
Sc = (kin.vis/D)
FSc = (Sc^(1/3))/2.9
Egb = (Sc^(-2/3)/14.5)*{1/6*log(1+FSc)^2/(1-FSc+FSc^2)+1/sqrt(3)*atan((2*FSc-1)/sqrt(3))
        +3.1416/6*sqrt(3)}^-1
cd = 1/6
kx = 0.216
LAI = 0.2
h =  28
d  = 21
phi_H.1 = (1-16*x)^(-0.5)
phi_H.2 = 1+5*x
phi_H = ifelse(x<=0, phi_H.1, phi_H.2)
phi_M.1 = (1-16*x)^(-0.25)
phi_M.2 = 1+5*x
phi_M = ifelse(x<=0, phi_M.1, phi_M.2)
lmp = (0.41*(z-d))/(phi_H*(z-d)/abs(L))
lmp_h = (0.41*(h-d))/(phi_M*(h-d)/abs(L))
alphaPZ = {(kx*LAI)/(12*k_c^2*(1-d/h)^2)}^(1/3)*phi_M^(2/3)*{(h-d)/abs(L)}
C_IT = 0.056
Tau_phplus.1 = (Tau*u_star^2)/kin.vis
Tau_phplus.2 = C_IT
Tau_phplus = ifelse(Tau_phplus.1<20,Tau_phplus.1, Tau_phplus.2)
E_t.1 = 2.5*10^-3*C_IT*(Tau_phplus)^2
E_t.2 = C_IT
E_t = ifelse(Tau_phplus.1<20, E_t.1, E_t.2)
u_starf = u_star*exp(-alphaPZ)
Tau_phplus.f1 = (Tau*u_star^2)/kin.vis
Tau_phplus.f2 = 0.14
Tau_phplusf = ifelse(Tau_phplus.f1<20,Tau_phplus.f1, Tau_phplus.f2)
E_gt1 = 2.5*10^-3*0.14*(Tau_phplusf)^2
E_gt2 = 0.14
Egt = ifelse(Tau_phplus.f1<20, E_gt1, E_gt2)
Eg = Egb + Egt
**Compute Qg (non-dimensional number)**
Qg = Eg*h/lmp_h
**Compute Q**
U_z = V5
U_h = U_z/(exp(alphaPZ*(z/h-1)))
L_obs = 0.03
C_B = 1.262
Re_h = (U_h*L_obs)/(kin.vis)
E_B = C_B*(Sc^(-2/3))*(Re_h^(-1/2))
C_IN = 0.216
E_IN = C_IN*(dp/L_obs)*(2+log(4*L_obs/dp))
C_IM = 0.130
beta_IM = 0.47
St_h = (Tau*U_h)/L_obs
E_IM = C_IM*(St_h/(St_h+beta_IM))^2
E_T = (U_h/u_star)*(E_B+E_IN+E_IM)+E_t
Q  = LAI*E_T*h/(lmp_h)
**Compute etaPZ**
etaPZ = (alphaPZ^2/4+Q)^0.5
Vds = u_star*Eg*{(1+Q/Qg-alphaPZ/2)*tanh(etaPZ)/etaPZ}/{(1+Q+alphaPZ/2)*tanh(etaPZ)/etaPZ}
Vd = Vdrift+1/(Ra+1/Vds);Vd

**Accuracy Evaluation: Water**
**Dry deposition parameterization by Petroff and Zhang (2010)**
attach(Moller_Schumann_1970)
C1 = 0.2789
C2 = 3.115
C3 = 5.415*10^-11
C4 = -1.399
RH = 90/100
dp_i = V1
dp_d = dp_i*10^-6
rd = dp_d/2
r_w = {(C1*rd^C2)/(C3*rd^C4-log10(RH))+rd^3}^(1/3)
dp = r_w*2
k_B = 1.38*10^-23
Temp = 273.15+25
P = 101325
d_air = 3.7208*10^-10
lambda = (k_B*Temp)/(sqrt(2)*3.1416*P*d_air^2)
```

```
C = 1+(2*lambda/dp)*(1.257+0.4*exp(-0.55*dp/lambda))
dyn.vis = ((5*10^-8)*Temp)+4*10^-6
rho = 1500
Tau = (rho*(dp)^2*C)/(18*dyn.vis)
Ws = Tau*9.81
Vphor = (5*10^-3)/100
Vdrift = Ws+Vphor
**Compute aerodynamic resistance (Ra):**
z = 8/100
L = 50
x = z/L
shi_H.1 = 2*log(0.5*{1+(1-16*x)^0.5})
shi_H.2 = -5*x
shi_H = ifelse(x <= 0, shi_H.1 , shi_H.2)
zR = 5
u_star = 0.4
z0_1 = 0.021*(u_star)^3.32
z0_2 = 0.00098*(u_star)^1.65
z0 = ifelse(u_star<= 0.16, z0_1, z0_2)
k_c = 0.41
Ra = (log(zR/z0)-shi_H)/(k_c*u_star)
**Compute surface resistance (Rs)**
kin.vis = ((9*10^-8)*Temp)+10^-5
D = (C*k_B*Temp)/(3*3.1416*dyn.vis*dp)
Sc = (kin.vis/D)
FSc = (Sc^(1/3))/2.9
Egb = (Sc^(-2/3)/14.5)*{1/6*log(1+FSc)^2/(1-FSc+FSc^2)+1/sqrt(3)*atan((2*FSc-
1)/sqrt(3))+3.1416/6*sqrt(3)}^-1
Eg = Egb
Vd = Vdrift+1/(Ra+(1/(Eg*u_star)));Vd

**Accuracy Evaluation: Ice/snow**
**Dry deposition velocity parameterization by Petroff and Zhang (2010)**
attach(Ibrahim_1983)
C1 = 0.2789
C2 = 3.115
C3 = 5.415*10^-11
C4 = -1.399
RH = 60/100
dp_i = c(0.22, 0.73)
dp_d = dp_i*10^-6
rd = dp_d/2
r_w = {(C1*rd^C2)/(C3*rd^C4-log10(RH))+rd^3}^(1/3)
dp = r_w*2
**Correction factor, C**
k_B = 1.38*10^-23
**Temp_1 = 25**
Temp = 273.15+3
P = 101325
d_air = 3.7208*10^-10
lambda = (k_B*Temp)/(sqrt(2)*3.1416*P*d_air^2)
C = 1+(2*lambda/dp)*(1.257+0.4*exp(-0.55*dp/lambda))
dyn.vis = ((5*10^-8)*Temp)+4*10^-6
rho = 1500
Tau = (rho*(dp)^2*C)/(18*dyn.vis)
Ws = Tau*9.81
Vphor = (2*10^-4)/100
Vdrift = Ws+Vphor
**Compute aerodynamic resistance (Ra):**
**z =**
**L = z/L**
**x = z/L**
x = 0.2
**Compute stability function (shi_H)**
shi_H.1 = 2*log(0.5*{1+(1-16*x)^0.5})
shi_H.2 = -5*x
shi_H = ifelse(x <= 0, shi_H.1 , shi_H.2)
zR = 10
u_star = 0.12
z0 = 0.1/100
k_c = 0.41
Ra = (log(zR/z0)-shi_H)/(k_c*u_star)  # m/s
**Compute surface resistance (Rs)**
```

```
kin.vis = ((9*10^-8)*Temp)+10^-5
D = (C*k_B*Temp)/(3*3.1416*dyn.vis*dp)
Sc = (kin.vis/D)
FSc = (Sc^(1/3))/2.9
Egb = (Sc^(-2/3)/14.5)*{1/6*log(1+FSc)^2/(1-FSc+FSc^2)+1/sqrt(3)*atan((2*FSc-1)/sqrt(3))+3.1416/6*sqrt(3)}^-1
Eg = Egb
Vd = Vdrift+1/(Ra+(1/(Eg*u_star)));Vd
```

**Codes for evaluation of model accuracy using Kouznetsov and Sofiev (2012) parameterization**

```
**Accuracy Evaluation: Grass**
**Dry deposition parameterization by Kouznetsov and Sofiev (2012)**
attach(Allen_etal_1990)
C1 = 0.4809
C2 = 3.082
C3 = 3.110*10^-11
C4 = -1.428
RH = 90/100
dp_a = 0.48
dp_i = dp_a*10^-6
rd = dp_i/2
r_w = {(C1*rd^C2)/(C3*rd^C4-log10(RH))+rd^3}^(1/3)
dp = r_w*2
**Correction factor, C**
k_B = 1.38*10^-23
Temp_1 = V1
Temp = 273.15+V1
P = 101325
d_air = 3.7208*10^-10
lambda = (k_B*Temp)/(sqrt(2)*3.1416*P*d_air^2)
C = 1+(2*lambda/dp)*(1.257+0.4*exp(-0.55*dp/lambda))
dyn.vis = ((5*10^-8)*Temp)+4*10^-6
rho =  2000
Tau = (rho*(dp)^2*C)/(18*dyn.vis)
V_s = Tau*9.81
u_star = V2
a   =  2*10^-3
kin.vis = ((9*10^-8)*Temp)+10^-5
D = (C*k_B*Temp)/(3*3.1416*dyn.vis*dp)
Sc = (kin.vis/D)
Re_star = (u_star*a)/kin.vis
**Compute V_diff (velocity for diffusion)**
V_diff = 2*(Re_star^(-0.5))*Sc^(-2/3) #m/s
**Compute V_int (velocity for interception)**
V_int = 80*u_star*((dp/a)^2)*(Re_star^(0.5))
**Compute V_imp**
C_S = 0.003
C_R = 0.3
LAI = 4
CsCR = (C_S+C_R/LAI)^0.5
u.Uh = 0.3
u_star.by.U_h = min(u.Uh, CsCR)
**Compute Re_c**
Re_c = ((u_star.by.U_h)^-1)^2*Re_star
**Calculate St**
St = (Tau*u_star)/a
**Calculate St_e**
St_e = St - Re_c^(-0.5)
eta_impSt.e1 = exp((-0.1/(St_e - 0.15 ) )-(1/sqrt(St_e -0.15)))
eta_impSt.e2 = 0
eta_impSt.e = ifelse(St_e>0.15,eta_impSt.e1,eta_impSt.e2)
V_imp = ((2*u_star.by.U_h)/u_star)*eta_impSt.e*(St-u_star.by.U_h*Re_star^-0.5)
**Dry deposition velocity**
Vd = V_diff+V_int+V_imp+V_s;Vd
```

```r
**Accuracy Evaluation: Coniferous forest**
**Dry deposition parameterization by Kouznetsov and Sofiev (2012)**
attach(Rannik_etal_2000)
C1 = 0.2789
C2 = 3.115
C3 = 5.145*10^-11
C4 = -1.399
RH = 0.90
dp_a = V1
dp_i = dp_a*10^-6
rd = dp_i/2
r_w = {(C1*rd^C2)/(C3*rd^C4-log10(RH))+rd^3}^(1/3)
dp = r_w*2
**Correction factor, C**
k_B = 1.38*10^-23
Temp =  273.15+25
P = 101325
d_air = 3.7208*10^-10
lambda = (k_B*Temp)/(sqrt(2)*3.1416*P*d_air^2)
C = 1+(2*lambda/dp)*(1.257+0.4*exp(-0.55*dp/lambda))
dyn.vis = 1.891*10^-5
**dyn.vis = V2**
rho = 1500
Tau = (rho*(dp)^2*C)/(18*dyn.vis)
V_s = Tau*9.81
**Need to compute Sc, Re_star**
u_star = V2
a   =  0.7*10^-3
kin.vis = 1.683*10^-5
D = (C*k_B*Temp)/(3*3.1416*dyn.vis*dp)
Sc = (kin.vis/D)
Re_star = (u_star*a)/kin.vis
**Compute V_diff (velocity for diffusion)**
V_diff = 2*(Re_star^(-0.5))*Sc^(-2/3)
**Compute V_int (velocity for interception)**
V_int = 80*u_star*((dp/a)^2)*(Re_star^(0.5))
**Compute V_imp**
C_S = 0.003
C_R = 0.3
LAI = 6
CsCR = (C_S+C_R/LAI)^0.5
u.Uh = 0.3
u_star.by.U_h = min(u.Uh, CsCR)
**Compute Re_c**
Re_c = ((u_star.by.U_h)^-1)^2*Re_star
**Calculate St**
St = (Tau*u_star)/a
**Calculate St_e**
St_e = St - Re_c^(-0.5)
eta_impSt.e1 = exp((-0.1/(St_e - 0.15 ) )-(1/sqrt(St_e -0.15)))
eta_impSt.e2 = 0
eta_impSt.e = ifelse(St_e>0.15,eta_impSt.e1,eta_impSt.e2)
V_imp = ((2*u_star.by.U_h)/u_star)*eta_impSt.e*(St-u_star.by.U_h*Re_star^-0.5)
**Dry deposition velocity**
Vd = V_diff+V_int+V_imp+V_s;Vd

**Accuracy Evaluation: Deciduous forest**
**Dry deposition parameterization by Kouznetsov and Sofiev (2012)**
attach(Wesely_etal_1983)
C1 = 0.2789
C2 = 3.115
C3 = 5.145*10^-11
C4 = -1.399
RH = 0.95
dp_a = 0.4
dp_i = dp_a*10^-6
rd = dp_i/2
r_w = {(C1*rd^C2)/(C3*rd^C4-log10(RH))+rd^3}^(1/3)
dp = r_w*2
**Correction factor, C**
k_B = 1.38*10^-23
Temp = V1
P = 101325
```

```
d_air = 3.7208*10^-10
lambda = (k_B*Temp)/(sqrt(2)*3.1416*P*d_air^2)
C = 1+(2*lambda/dp)*(1.257+0.4*exp(-0.55*dp/lambda))
dyn.vis = V2
rho = 2000
Tau = (rho*(dp)^2*C)/(18*dyn.vis)
V_s = Tau*9.81
**Need to compute Sc, Re_star**
u_star = V4
a  = 0.7*10^-3
kin.vis = 1.597*10^-5
D = (C*k_B*Temp)/(3*3.1416*dyn.vis*dp)
Sc = (kin.vis/D)
Re_star = (u_star*a)/kin.vis
**Compute V_diff (velocity for diffusion)**
V_diff = 2*(Re_star^(-0.5))*Sc^(-2/3) #m/s
**Compute V_int (velocity for interception)**
V_int = 80*u_star*((dp/a)^2)*(Re_star^(0.5))
**Compute V_imp**
C_S = 0.003
C_R = 0.3
LAI = 0.2
CsCR = (C_S+C_R/LAI)^0.5
u.Uh = 0.3
u_star.by.U_h = min(u.Uh, CsCR)
**Compute Re_c**
Re_c = ((u_star.by.U_h)^-1)^2*Re_star
**Calculate St**
St = (Tau*u_star)/a
**Calculate St_e**
St_e = St - Re_c^(-0.5)
eta_impSt.e1 = exp((-0.1/(St_e - 0.15 ) )-(1/sqrt(St_e -0.15)))
eta_impSt.e2 = 0
eta_impSt.e = ifelse(St_e>0.15,eta_impSt.e1,eta_impSt.e2)
V_imp = ((2*u_star.by.U_h)/u_star)*eta_impSt.e*(St-u_star.by.U_h*Re_star^-0.5)
**Dry deposition velocity**
Vd = V_diff+V_int+V_imp+V_s;Vd

**Accuracy Evaluation: Ice/snow**
**Dry deposition parameterization by Kouznetsov and Sofiev (2012)**
attach(Ibrahim_1983)
C1 = 0.4809
C2 = 3.082
C3 = 3.110*10^-11
C4 = -1.428
RH = 0.90
dp_a = c(0.22, 0.73)
dp_i = dp_a*10^-6
rd = dp_i/2
r_w = {(C1*rd^C2)/(C3*rd^C4-log10(RH))+rd^3}^(1/3)
dp = r_w*2
**Correction factor, C**
k_B = 1.38*10^-23
Temp = 273.15+3
P = 101325
d_air = 3.7208*10^-10
lambda = (k_B*Temp)/(sqrt(2)*3.1416*P*d_air^2)
C = 1+(2*lambda/dp)*(1.257+0.4*exp(-0.55*dp/lambda))
dyn.vis = ((5*10^-8)*Temp)+4*10^-6
rho =  1500
Tau = (rho*(dp)^2*C)/(18*dyn.vis)
V_s = Tau*9.81
u_star = 0.12
a  =  0.5*10^-3
kin.vis = ((9*10^-8)*Temp)+10^-5
D = (C*k_B*Temp)/(3*3.1416*dyn.vis*dp)
Sc = (kin.vis/D)
Re_star = (u_star*a)/kin.vis
**Compute V_diff (velocity for diffusion)**
V_diff = 2*(Re_star^(-0.5))*Sc^(-2/3) #m/s
**Compute V_int (velocity for interception)**
V_int = 80*u_star*((dp/a)^2)*(Re_star^(0.5))
**Compute V_imp**
```

```
C_S = 0.003
C_R = 0.3
LAI = 0
CsCR = (C_S+C_R/LAI)^0.5
u.Uh = 0.3
u_star.by.U_h = min(u.Uh, CsCR)
**Compute Re_c**
Re_c = ((u_star.by.U_h)^-1)^2*Re_star
**Calculate St**
St = (Tau*u_star)/a
**Calculate St_e**
St_e = St - Re_c^(-0.5)
eta_impSt.e1 = exp((-0.1/(St_e - 0.15 ) )-(1/sqrt(St_e -0.15)))
eta_impSt.e2 = 0
eta_impSt.e = ifelse(St_e>0.15,eta_impSt.e1,eta_impSt.e2)
**V_imp = ((2*u_star.by.U_h)/u_star)*eta_impSt.e*(St-u_star.by.U_h*Re_star^-0.5)**
V_imp = (2*u_star.by.U_h*eta_impSt.e*(St-(u_star.by.U_h*Re_star^-0.5)))*u_star
**Dry deposition velocity**
Vd = V_diff+V_int+V_imp+V_s;Vd

**Accuracy Evaluation: Water**
**Dry deposition parameterization by Kouznetsov and Sofiev (2012)**
C1 = 0.2789
C2 = 3.115
C3 = 5.415*10^-11
C4 = -1.399
dp_i = c(0.22, 0.73)
dp = dp_i*10^-6
rho = 1500
RH = 0.60
Temp = 25+273.15
u_star = 0.12
z0 =0.1/100
L = 40
kin.vis = ((9*10^-8)*Temp)+10^-5
dyn.vis = ((5*10^-8)*Temp)+4*10^-6
d_air = 3.7208*10^-10
k_B = 1.38*10^-23
P = 101325
lambda = (k_B*Temp)/(sqrt(2)*3.1416*P*d_air^2)
C = 1+((2*lambda/dp)*(1.257+0.4*exp(-0.55*dp/lambda)))
tau_p = (rho*(dp)^2*C)/(18*dyn.vis)
D = (C*k_B*Temp)/(3*3.1416*dyn.vis*dp)
v_s = 9.81*tau_p
taup = (tau_p*u_star^2)/(kin.vis)
Sc = (kin.vis/D)
vsplus = v_s/u_star
rplus = (dp*u_star)/(2*kin.vis)
R_s = 0
Rsplus = u_star*R_s
z_meas = 8
zpmax = (z_meas*u_star)/(kin.vis)
MOplus = (kin.vis)/(u_star*L)
**Fixed parameters**
Zbuf = 3
Ztf = 18                          #turbophoretic sublayer height
taultf = 5                        #Lagrangian time in turbophoretic layer
Nutp_Ztf= (0.4*(Ztf)^3)/(Ztf^2+200)  #Dimensionless eddy viscosity of air
It_Ztf = (2.5*log10(Ztf ))-(100/Ztf^2)
It_Zbuf = (2.5*log10(Zbuf))-(100/Zbuf^2)
S = Sc^(1/3)
Zl = 20/S
fTmp = 2.5/Sc
fTmp1 = (fTmp^3/27+(fTmp*(100+5*sqrt(8*fTmp/27)+400)))^(1/3)
Zl_1 = fTmp1+((fTmp*fTmp)/(9*fTmp1))+(1/3)*fTmp         #Zl updated as zl_1
fTmp_1 = (Zl_1^2)/(Zl_1^2+200)                          #fTmp updated as fTmp_1
fTmp_2 = 1.2*(fTmp_1)-0.8*fTmp_1^2                      #fTmp_1 updated as fTmp_2
fTmp1_1 = 1/(Sc*fTmp_2)                                 #fTmp1 updated as fTmp1_1
fTmp_3 = 1/Sc                                           #ftmp_2 updated as fTmp_3
x_1 = Zl_1 - fTmp1_1
x_2 = Zl_1
x_3 = Zl_1+fTmp1_1
Nutp_x_1 = (0.4*x_1^3)/(x_1^2+200)
```

```
Nutp_x_2 = (0.4*x_2^3)/(x_2^2+200)
Nutp_x_3 = (0.4*x_3^3)/(x_3^2+200)
fIvd = Rsplus+(Zl_1-
fTmp1_1)*Sc+0.3333*fTmp1_1*(1/fTmp_3+((0.4*x_1^3)/(x_1^2+200)))+4/(fTmp_3+((0.4*x_2^3)/(x_2^2+200)))+1/(fTmp_3+((0
.4*x_3^3)/(x_3^2+200)))
x_4 = zpmax
x_5 = Zl_1 + fTmp1_1
It_x_4 = 2.5*log10(x_4) - 100/(x_4^2)
It_x_5 = 2.5*log10(x_5) - 100/(x_5^2)
s = 2.35*(zpmax*MOplus+abs(zpmax*MOplus))
u1 = 0.5*(abs(zpmax*MOplus)-zpmax*MOplus)
u = -4*u1/(2.65*sqrt(u1*sqrt(u1))+1)
fu_Psi = s+u
fTmp_4 = It_x_4 - It_x_5 + 2.5*fu_Psi
fIvd_1 = fIvd+0.5*(fTmp_4+abs(fTmp_4))      #fIvd updated as fIvd_1
fu_vdplus_smooth_1 = 1/fIvd_1               #mind fu_vdplus_smooth is denoted as _1
Il_input1= Zl*S/7.92
Il_input2= rplus*S/7.92
Il_1 = -0.16667*log10(Il_input1^2-Il_input1+1)+0.57735*atan((2*Il_input1-1)*0.57735)+0.3333*log10(Il_input1+1)
Il_2 = -0.16667*log10(Il_input2^2-Il_input2+1)+0.57735*atan((2*Il_input2-1)*0.57735)+0.3333*log10(Il_input2+1)
R = 7.92*S^2*(Il_1-Il_2)  #laminar resistance
R_1 = 0.5*(R+abs(R))      #R updated as R_1 #should be zero if rplus > Zl
Zl_2 = 0.5*(rplus+Zl+abs(rplus-Zl))   #Zl updated as Zl_2, not to be confused with Zl_1 as if statement is in use
It_Zbuf = 2.5*log10(Zbuf) - (100/(Zbuf^2))
It_Zl_2 = 2.5*log10(Zl_2) - (100/(Zl_2^2))
R1 = It_Zbuf - It_Zl_2
R_2 = R_1+0.5*(R1+abs(R1))                   #R_1 updated as R_2
fTmp_5 = vsplus*R_2                          #fTmp used as fTmp_5. Not updated as previous fTmp_4 is for
different condition
fTmp_6 = exp(-fTmp_5)                        #fTmp_5 updated as fTmp_6
fIvd_2  = Rsplus*fTmp_6+(1-fTmp_6)/vsplus    #fIvd denoted as fIvd_2 NOT updated previous fIvd_1 is for different
condition
fIvd_3 = Rsplus+R_2                          #fIvd_3 used; NOT updated
fIvd_23 = ifelse(abs(fTmp_5)>0.001, fIvd_2, fIvd_3 )
**Use above values for the following calculations for turbophoretic layer**
V  = 0.81*taup/(Ztf-Zbuf)/(1+taup/taultf) + vsplus   #chcek for sign of vsplus
It_Ztf = 2.5*log10(Ztf) - 100/(Ztf^2)
R_3  = ((It_Ztf)-(It_Zbuf))*(1+taup/taultf)    #R_2 updated as R_3
fTmp_7 = V*R_3                               #fTmp_6 updated as fTmp_7
fTmp_8 = exp(-fTmp_7)                        #fTmp_7 updated as fTmp_8
fIvd_4 = (fIvd_23*fTmp_8)+(1-fTmp_8)/V       #fIvd_23 updated as fIvd_4
fIvd_5 = fIvd_3+R_3                          #fIvd_5 used; NOT updated
fIvd_45 = ifelse(abs(fTmp_7)>0.001, fIvd_4, fIvd_5)
**Now calculations for the Lagrangian turbophoretic layer**
Ztf2 = 2*taup
V_1 = 0.4+vsplus
R_4 =
0.1667*((1+taup/(0.5*Ztf))/(Nutp_Ztf)+4*(1+taup/(0.25*(Ztf+Ztf2)))/((0.4*(0.5*(Ztf+Ztf2))^3)/((0.5*(Ztf+Ztf2))^2+2
00))+(1+taup/(0.5*Ztf2))/((0.4*Ztf2 ^3)/(Ztf2^2+200)))*(Ztf2-Ztf)
fTmp_9 = V_1*R_4
fTmp_10 = exp(-fTmp_9)
fIvd_6 = fIvd_45*fTmp_10 + (1-fTmp_10)/V_1   #fIvd_5 or fIvd_4 (fIvd_45)
fIvd_7 = fIvd_45+R_3                         #Either fIvd_4 or fIvd_5, R_3 used not R_4 since outside if statement
fIvd_67 = ifelse(abs(fTmp_9) > 0.001, fIvd_6, fIvd_7)
Ztf2_2 = Ztf
fIvd67_Ztf2 = ifelse(Ztf2>Ztf, fIvd_67, Ztf2_2)

**Following calculations are for aerodynamic layer**
It_zpmax = 2.5*log10(zpmax) - 100/(zpmax^2)
Ztf22 = ifelse(Ztf2>Ztf, Ztf2, Ztf2_2)
It_Ztf22  = 2.5*log10(Ztf22) - 100/(Ztf22^2)       #be careful which Ztf2 to be used here based on previous
condition
R_5 = It_zpmax - It_Ztf22  + 2.5*fu_Psi            #R_5 used, NOT updated from R_4
R_6 = 0.5*(R_5+abs(R_5))                           #R_5 updated as R_6
fTmp_11  = vsplus*R_6                              #fTmp_11 used, NOT updated from fTmp_10
fTmp_12 = exp(-fTmp_11)                            #fTmp_11 updated as fTmp_12
fIvd_4567 = ifelse(Ztf2>Ztf, fIvd_67,fIvd_45)
fIvd_8 = (fIvd_4567* fTmp_12)+(1-fTmp_12)/vsplus   # Caution:fIvd_45 or fIvd_67 may be used
fIvd_9 = fIvd_4567 + R_6                           #fIvd_9 used, NOT updated. Caution:fIvd_5 or fIvd_4 may be used
fIvd_10 = ifelse(abs(fTmp_11) > 0.001, fIvd_8, fIvd_9)
fu_vdplus_smooth_3 = 1/fIvd_10                     #fu_vdplus_smooth_3 used. NOT updated. fIvd_8 could be used
fu_vdplus_smooth = ifelse(Zl>Zbuf, fu_vdplus_smooth_1, fu_vdplus_smooth_3)
Vd_smooth = fu_vdplus_smooth*u_star;Vd_smooth
```

**Codes for evaluation of model accuracy using Zhang and He (2014) parameterization**

```
**Accuracy Evaluation: Grass (code is similar for coniferous forest)**
**Dry deposition parameterization by Zhang and He (2014)**
attach(Allen_etal_1991)
C1 = 0.4809
C2 = 3.082
C3 = 3.110*10^-11
C4 = -1.428
RH = 90/100
dp_a = 0.48
dp_i = dp_a*10^-6
rd = dp_i/2
r_w = {(C1*rd^C2)/(C3*rd^C4-log10(RH))+rd^3}^(1/3)
dp = r_w*2
**Correction factor, C**
k_B = 1.38*10^-23
Temp_1 = V1
Temp = 273.15+V1
P = 101325
d_air = 3.7208*10^-10
lambda = (k_B*Temp)/(sqrt(2)*3.1416*P*d_air^2)
C = 1+(2*lambda/dp)*(1.257+0.4*exp(-0.55*dp/lambda))
dyn.vis = ((5*10^-8)*Temp)+4*10^-6          # kg/m*s (temp. corrected viscosity coeff. of air)
rho = 2000
Tau = (rho*(dp)^2*C)/(18*dyn.vis)
V_g = Tau*9.81
Rg = 1/V_g
u_star = V2
a1 = 4.8*10^-3
z = 2
L = V4
x = z/L
**Compute stability function (shi_H)**
shi_H.1 = 2*log(0.5*{1+(1-16*x)^0.5})
shi_H.2 = -5*x
shi_H =ifelse(x <= 0, shi_H.1 , shi_H.2)
zR = 3.5
z0 = 0.01
k_c = 0.41
Ra = (log(zR/z0)-shi_H)/(k_c*u_star)
**Calculate Vds = 1/Rs**
**For PM2.5**
Vds_PM2.5 = (a1*u_star)
Rds_PM2.5 = (1/Vds_PM2.5)
**For PM2.5-10**
**b1= -1.6*10^-1**
**b2= 1.5*10^0**
**b3 = 7.8*10^-1**
**c1= 1.8**
**c2 = -2.0*10^-1**
**c3 = -5.3*10^-1**
**k = c1*u_star+c2*u_star^2+c3*u_star^3**
**LAI =**
**LAImax =**
**Vds_PM10 = (b1*u_star+b2*u_star^2+b3*u_star^3)  #*exp(k*(LAI/LAImax)-1)**
**Rds_PM10 = 1/Vds_PM10**
**For PM10+**
**d1= -2.2**
**d2= 3.9*10^1**
**d3 = -6.7**
**f1= 6.2**
**f2 = -1.2*10^1**
**f3 = 6.1**
**k = f1*u_star+f2*u_star^2+f3*u_star^3**
**LAI =**
**LAImax =**
**Vds_PM10Plus = (d1*u_star+d2*u_star^2+d3*u_star^3)*exp(k*(LAI/LAImax)-1)**
**Rds_PM10Plus = (1/Vds_PM10Plus)**
**Compute Vd**
Vd = 1/Rg+(1/(Ra+Rds_PM2.5));Vd
```

```r
**Accuracy Evaluation: Deciduous forest**
**Dry deposition parameterization by Zhang and He (2014)**
attach(Rannik_etal_2000)
C1 = 0.2789
C2 = 3.115
C3 = 5.145*10^-11
C4 = -1.399
RH = 90/100
dp_a = V1
dp_i = dp_a*10^-6
rd = dp_i/2
r_w = {(C1*rd^C2)/(C3*rd^C4-log10(RH))+rd^3}^(1/3)
dp = r_w*2
k_B = 1.38*10^-23
Temp = 273.15+25
P = 101325
d_air = 3.7208*10^-10
lambda = (k_B*Temp)/(sqrt(2)*3.1416*P*d_air^2)
C = 1+(2*lambda/dp)*(1.257+0.4*exp(-0.55*dp/lambda))
dyn.vis = 1.891*10^-5
rho = 1500
Tau = (rho*(dp)^2*C)/(18*dyn.vis)
V_g = Tau*9.81
Rg = 1/V_g
u_star = V2
a1 = 4.3*10^-3
z = 23.7
L = 200
x = z/L
**Compute stability function (shi_H)**
shi_H.1 = 2*log(0.5*{1+(1-16*x)^0.5})
shi_H.2 = -5*x
shi_H =ifelse(x <= 0, shi_H.1 , shi_H.2)
zR = 26
z0 = 1.2
k_c = 0.41
Ra = (log(zR/z0)-shi_H)/(k_c*u_star)
**Calculate Vds = 1/Rs**
**For PM2.5**
Vds_PM2.5 = (a1*u_star)
Rds_PM2.5 = (1/Vds_PM2.5)
**For PM2.5-10**
**b1= -1.6*10^-1**
**b2= 1.5*10^0**
**b3 = 7.8*10^-1**
**c1= 1.8**
**c2 = -2.0*10^-1**
**c3 = -5.3*10^-1**
k = c1*u_star+c2*u_star^2+c3*u_star^3
LAI = 12
LAImax = 12
Vds_PM10 = (b1*u_star+b2*u_star^2+b3*u_star^3)  #*exp(k*(LAI/LAImax)-1)
Rds_PM10 = 1/Vds_PM10
**For PM10+**
**d1= -2.2**
**d2= 3.9*10^1**
**d3 = -6.7**
**f1= 6.2**
**f2 = -1.2*10^1**
**f3 = 6.1**
**k = f1*u_star+f2*u_star^2+f3*u_star^3**
**LAI =**
**LAImax =**
**Vds_PM10Plus = (d1*u_star+d2*u_star^2+d3*u_star^3)*exp(k*(LAI/LAImax)-1)**
**Rds_PM10Plus = (1/Vds_PM10Plus)**
**Compute Vd**
Vd = 1/Rg+(1/(Ra+Rds_PM2.5));Vd

**Accuracy Evaluation: Water**
**Dry deposition parameterization by Zhang and He (2014)**
attach(Caffrey_etal_1998)
C1 = 0.2789
C2 = 3.115
```

```
C3 = 5.415*10^-11
C4 = -1.399
RH = 79/100
dp_a = V1
dp_i = dp_a*10^-6
rd = dp_i/2
r_w = {(C1*rd^C2)/(C3*rd^C4-log10(RH))+rd^3}^(1/3)
dp = r_w*2
**Correction factor, C**
k_B = 1.38*10^-23
Temp = 273.15+22
P = 101325
d_air = 3.7208*10^-10
lambda = (k_B*Temp)/(sqrt(2)*3.1416*P*d_air^2)
C = 1+(2*lambda/dp)*(1.257+0.4*exp(-0.55*dp/lambda))
dyn.vis = ((5*10^-8)*Temp)+4*10^-6
rho = 2000
Tau = (rho*(dp)^2*C)/(18*dyn.vis)
V_g = Tau*9.81
Rg = 1/V_g
a1 = 6.9*10^-3
z = 8/100
L = 50
x = z/L
**Compute stability function (shi_H)**
shi_H.1 = 2*log(0.5*{1+(1-16*x)^0.5})
shi_H.2 = -5*x
shi_H =ifelse(x <= 0, shi_H.1 , shi_H.2)
zR = 5
u_star = 13.5/100
z0_1 = 0.021*(u_star)^3.32
z0_2 = 0.00098*(u_star)^1.65
z0 = ifelse(u_star<= 0.16, z0_1, z0_2)
k_c = 0.41
Ra = (log(zR/z0)-shi_H)/(k_c*u_star)  # m/s
**Calculate Vds = 1/Rs**
**For PM2.5**
Vds_PM2.5 = (a1*u_star)
Rds_PM2.5 = (1/Vds_PM2.5)
**Compute Vd**
Vd = 1/Rg+(1/(Ra+Rds_PM2.5));Vd

**Accuracy Evaluation: Ice/snow**
**Dry deposition parameterization by Zhang and He (2014)**
**attach(Nilsson_Rannik_2001)**
C1 = 0.2789
C2 = 3.115
C3 = 5.415*10^-11
C4 = -1.399
RH = 60/100
dp_a = c(0.22, 0.73)
dp_i = dp_a*10^-6
rd = dp_i/2
r_w = {(C1*rd^C2)/(C3*rd^C4-log10(RH))+rd^3}^(1/3)
dp = r_w*2
**Correction factor, C**
k_B = 1.38*10^-23
Temp = 273.15+3
P = 101325
d_air = 3.7208*10^-10
lambda = (k_B*Temp)/(sqrt(2)*3.1416*P*d_air^2)
C = 1+(2*lambda/dp)*(1.257+0.4*exp(-0.55*dp/lambda))
dyn.vis = ((5*10^-8)*Temp)+4*10^-6
rho = 1500
Tau = (rho*(dp)^2*C)/(18*dyn.vis)
V_g = Tau*9.81
Rg = 1/V_g
a1 = 4.3*10^-3
**z =**
**L =**
**x = z/L**
x = 0.2
**Compute stability function (shi_H)**
```

```
shi_H.1 = 2*log(0.5*{1+(1-16*x)^0.5})
shi_H.2 = -5*x
shi_H =ifelse(x <= 0, shi_H.1 , shi_H.2)
zR = 10
u_star = 0.12
z0 = 0.1/100
k_c = 0.41
Ra = (log(zR/z0)-shi_H)/(k_c*u_star)  # m/s
**Calculate Vds = 1/Rs**
**For PM2.5**
Vds_PM2.5 = (a1*u_star)
Rds_PM2.5 = (1/Vds_PM2.5)
**Compute Vd**
Vd = 1/Rg+(1/(Ra+Rds_PM2.5));Vd
```

**Codes for evaluation of model accuracy using Zhang and Sao (2014) parameterization**

```
**Accuracy Evaluation: Rough and smooth surfaces**
**Dry deposition parameterization by Zhang and He (2014)**
**attach(Allen_etal_1991)**
C1 = 0.4809
C2 = 3.082
C3 = 3.110*10^-11
C4 = -1.428
RH_1 = 90
RH = 90/100
dp_a = 0.50
dp_i = dp_a*10^-6
rd = dp_i/2
r_w = {(C1*rd^C2)/(C3*rd^C4-log10(RH))+rd^3}^(1/3)
dp = r_w*2
**Correction factor, C**
k_B = 1.38*10^-23
Temp = 273.15+25
P = 101325
d_air = 3.7208*10^-10
lambda = (k_B*Temp)/(sqrt(2)*3.1416*P*d_air^2)
C = 1+(2*lambda/dp)*(1.257+0.4*exp(-0.55*dp/lambda))
Temp = 273.15+25
dyn.vis = ((5*10^-8)*Temp)+4*10^-6
rho = 1500
Tau = (rho*(dp_i)^2*C)/(18*dyn.vis)
Tau_wet = (rho*(dp)^2*C)/(18*dyn.vis)
Wt = Tau*9.81
Vg = Wt
Vg_wet = Tau_wet*9.81
u_star = 0.5
k  =  0.41
z  =  1
zd = 0.20
h_c = 0.23
z0 = 0.002
B1 = 0.45
Sc_T = (1+(Vg^2/u_star^2))^0.5
Ra = (Sc_T/(k*u_star))*(log((z-zd)/(h_c-zd)))  # For rough surface
**Ra = (B1*Sc_T/k*u_star)*log(z/z0)            # For smooth surface**
Rg = 1/Vg
**Calculate surface resistance (Rs)**
U_h = 2
kin.vis = ((9*10^-8)*Temp)+10^-5
d_c = 0.005
D = (C*k_B*Temp)/(3*3.1416*dyn.vis*dp)
Sc = (kin.vis/D)
Re_h = (U_h*d_c)/(kin.vis)
nB = 0.5
C_B = 0.467
E_B = C_B*Sc^(-2/3)*Re_h^(nB-1)
**Compute impaction collection efficiency (E_IM)**
beta_IM = 0.6
St_h = (Tau*u_star)/d_c
E_IM = (St_h/(St_h+beta_IM))^2
**Compute interception efficiency (E_IN)**
Ain = 150
```

```
E_IN = Ain*u_star*(10^(-St_h))*(2*dp/d_c)
**Compute R**
b = 2
R = exp(-b*sqrt(St_h))
**Compute w_dm**
B2 = 3
w_dm = (u_star/U_h*h_c)          #For rough surface
**w_dm = B2*u_star               #For smooth surface**
**compute Tau_c/Tau (ratio of stress)**
Beta = 200
C_1 = 6
C_2 = 0.1
lambda_FAI = 0.4
n_FAI = (lambda_FAI)/(h_c*d_c)
q = (3.1416*d_c^2)/4
eta_BAI = n_FAI*q
lambda_FAIe = ((lambda_FAI)/(1-eta_BAI)^C_2)*exp((-C_1*lambda_FAI)/(1-eta_BAI)^C_2)
Tau_c_BY_Tau = (Beta*lambda_FAIe)/(1+Beta*lambda_FAIe)
**Compute Rs**
E = E_B+E_IM+E_IN
Tau_wetplus = (Tau_wet*u_star^2)/kin.vis
Cd = 1/6
Rs = (R*w_dm*((E*Tau_c_BY_Tau/Cd)+(1+Tau_c_BY_Tau)*Sc^-1+10^(-3/Tau_wetplus))+Vg_wet)^-1
**Compute Vd**
Vd = (Rg+((Rs-Rg)/exp(Ra/Rg)))^-1;Vd
```

**Codes for Monte Carlo uncertainty evaluation for Zhang et al. (2001) parameterization**

```
**Dry deposition parameterization by Zhang et al. (2001)**
**Uncertainty test: Grass**
set.seed(5)
C1 = 0.2789
C2 = 3.115
C3 = 5.145*10^-11
C4 = -1.399
RH = replicate(10000,runif(100,0.76,0.84))
dp_i = 2.0*10^-6
rd = dp_i/2
r_w = {(C1*rd^C2)/(C3*rd^C4-log10(RH))+rd^3}^(1/3)
dp = r_w*2
**Correction factor, C**
k_B = 1.38*10^-23
Temp = (273.15+25)
P = 101325
d_air = 3.72*10^-10
lambda = (k_B*Temp)/(sqrt(2)*3.1416*P*d_air^2)
C = 1+(2*lambda/dp)*(1.257+0.4*exp(-0.55*dp/lambda))
dyn.vis = 1.8908*10^-5
rho = 1500
Vg = (rho*(dp)^2*9.81*C)/(18*dyn.vis)
**Compute aerodynamic resistance Ra:**
z = 5
L = replicate(10000,runif(100,45,55))
x = z/L
**Compute shi_H (stability function)**
shi_H2 = -5*x
zR = 3.5
z0 = replicate(10000,runif(100,0.03, 0.05))
u_star = replicate(10000,runif(100,0.27,0.33))
k_c = 0.41
Ra = (log(zR/z0)-shi_H2)/(k_c*u_star)
**Compute surface resistance Rs:**
e_0 = 3
R1 =1
**Compute E_B (collection efficienty from Brownian diffusion)**
kin.vis = 1.6834*10^-5
gamma = 0.54
D = (C*k_B*Temp)/(3*3.1416*dyn.vis*dp)
Sc = (kin.vis/D)
E_B = Sc^(-gamma)
**Compute E_IM (collection efficiency from impaction)**
alpha = 1.2
beta =  2
```

```r
A = 2/1000
St = (Vg*u_star)/(9.81*A)
E_IM = {St/(alpha+St)}^beta
**Compute E_IN (collection efficiency from interception)**
E_IN = 0.5*(dp/A)^2
Rs = 1/{(e_0*u_star)*(E_B+E_IM+E_IN)*R1}
**Compute Dry deposition velocity**
Vd <- Vg+(1/(Ra+Rs))
quantile(Vd, c(.05, 0.10, .50, 0.95))

**Dry deposition parameterization by Zhang et al. (2001)**
**Uncertainty test: Coniferous forest**
set.seed(5)
C1 = 0.2789
C2 = 3.115
C3 = 5.145*10^-11
C4 = -1.399
RH = replicate(10000,runif(100,0.76,0.84))
dp_i = 2.0*10^-6              #particle dia = 0.005-50 um (assumed, vary)
rd = dp_i/2
r_w = {(C1*rd^C2)/(C3*rd^C4-log10(RH))+rd^3}^(1/3)
dp = r_w*2
**Correction factor, C**
k_B = 1.38*10^-23
Temp = (273.15+25)
P = 101325
d_air = 3.72*10^-10
lambda = (k_B*Temp)/(sqrt(2)*3.1416*P*d_air^2)
C = 1+(2*lambda/dp)*(1.257+0.4*exp(-0.55*dp/lambda))
dyn.vis = 1.8908*10^-5
rho = 1500
Vg = (rho*(dp)^2*9.81*C)/(18*dyn.vis)    #m/s
**Compute aerodynamic resistance Ra:**
L = replicate(10000,runif(100,45,55))
x = z/L
shi_H2 = -5*x
zR = 30
z0 = replicate(10000,runif(100,0.9, 1.5))
u_star = replicate(10000,runif(100,0.27,0.33))
k_c = 0.41
Ra = (log(zR/z0)-shi_H2)/(k_c*u_star)  # m/s
**Compute surface resistance Rs:**
e_0 = 3
R1 =1
**Compute E_B (collection efficiency from Brownian diffusion)**
kin.vis = 1.6834*10^-5
gamma = 0.56
D = (C*k_B*Temp)/(3*3.1416*dyn.vis*dp)
Sc = (kin.vis/D)
E_B = Sc^(-gamma)
**Compute E_IM (collection efficiency from impaction)**
alpha = 1.0
beta =  2
A = 2/1000
St = (Vg*u_star)/(9.81*A)
E_IM = {St/(alpha+St)}^beta
**Compute E_IN (collection efficiency from interception)**
E_IN = 0.5*(dp/A)^2
Rs = 1/{(e_0*u_star)*(E_B+E_IM+E_IN)*R1}  #(m/s)
**Compute Dry deposition velocity**
Vd <- Vg+(1/(Ra+Rs))
quantile(Vd, c(.05, 0.10, .50, 0.95))

**Dry deposition parameterization by Zhang et al. (2001)**
**Uncertainty test: Deciduous forest**
set.seed(5)
C1 = 0.2789
C2 = 3.115
C3 = 5.145*10^-11
C4 = -1.399
RH = replicate(10000,runif(100,0.76,0.84))
dp_i = 2.0*10^-6
```

```r
rd = dp_i/2
r_w = {(C1*rd^C2)/(C3*rd^C4-log10(RH))+rd^3}^(1/3)
dp = r_w*2
**Correction factor, C**
k_B = 1.38*10^-23
Temp = (273.15+25)
P = 101325
d_air = 3.72*10^-10
lambda = (k_B*Temp)/(sqrt(2)*3.1416*P*d_air^2)
C = 1+(2*lambda/dp)*(1.257+0.4*exp(-0.55*dp/lambda))
dyn.vis = 1.8908*10^-5
rho = 1500
Vg = (rho*(dp)^2*9.81*C)/(18*dyn.vis)
**Compute aerodynamic resistance Ra:**
z = 35
L = replicate(10000,runif(100,45,55))
x = 35/L
**Compute shi_H (stability function)**
shi_H2 = -5*x
zR = 50
z0 = replicate(10000,runif(100,1.125, 1.875))
u_star = replicate(10000,runif(100,0.27,0.33))
k_c = 0.41
Ra = (log(zR/z0)-shi_H2)/(k_c*u_star)
**Compute surface resistance Rs:**
e_0 = 3
R1 =1
**Compute E_B (collection efficienty from Brownian diffusion)**
kin.vis = 1.6834*10^-5
gamma = 0.56
D = (C*k_B*Temp)/(3*3.1416*dyn.vis*dp)
Sc = (kin.vis/D)
E_B = Sc^(-gamma)
**Compute E_IM (collection efficienty from impaction)**
alpha = 0.80
beta =  2
A = 5/1000
St = (Vg*u_star)/(9.81*A)
E_IM = {St/(alpha+St)}^beta
**Compute E_IN (collection efficienty from interception)**
E_IN = 0.5*(dp/A)^2
Rs = 1/{(e_0*u_star)*(E_B+E_IM+E_IN)*R1}
**Compute Dry deposition velocity**
Vd <- Vg+(1/(Ra+Rs))
quantile(Vd, c(.05, 0.10, .50, 0.95))

**Dry deposition parameterization by Zhang et al. (2001)**
**Uncertainty test: Water**
set.seed(5)
C1 = 0.2789
C2 = 3.115
C3 = 5.415*10^-11
C4 = -1.399
RH = replicate(10000,runif(100,0.76,0.84))
dp_i = 2.0 #Parameter to vary for MCs
dp_d = dp_i*10^-6
rd = dp_d/2
r_w = {(C1*rd^C2)/(C3*rd^C4-log10(RH))+rd^3}^(1/3)
dp = r_w*2
**Correction factor, C**
k_B = 1.38*10^-23
Temp = 273.15+25
P = 101325
d_air = 3.72*10^-10
lambda = (k_B*Temp)/(sqrt(2)*3.1416*P*d_air^2)
C = 1+(2*lambda/dp)*(1.257+0.4*exp(-0.55*dp/lambda))
dyn.vis = ((5*10^-8)*Temp)+4*10^-6
rho = 1500
Vg = (rho*(dp)^2*9.81*C)/(18*dyn.vis)
**Compute aerodynamic resistance Ra:**
z = 8/100
L = replicate(10000,runif(100,45,55))
x = z/L
```

```
**Compute shi_H (stability function)**
shi_H.1 = 2*log(0.5*{1+(1-16*x)^0.5})
shi_H.2 = -5*x
shi_H =ifelse(x <= 0, shi_H.1 , shi_H.2)
zR = 5
u_star = replicate(10000,runif(100,0.27,0.33))
z0_1 = 0.021*(u_star)^3.32
z0_2 = 0.00098*(u_star)^1.65
z0 = ifelse(u_star<= 0.16, z0_1, z0_2)
k_c = 0.41
Ra = (log(zR/z0)-shi_H)/(k_c*u_star)
**Compute surface resistance Rs:**
e_0 = 3
R1 = 1
kin.vis = ((9*10^-8)*Temp)+10^-5
gamma = 0.50
D = (C*k_B*Temp)/(3*3.1416*dyn.vis*dp)
Sc = (kin.vis/D)
E_B = Sc^(-gamma)
alpha = 100
beta =  2
A = 2/1000
St = (Vg*u_star^2)/(kin.vis)
E_IM = {St/(alpha+St)}^beta
Rs = 1/{(e_0*u_star)*(E_B+E_IM)*R1}  #(m/s)
**Compute Dry deposition velocity**
Vd =   Vg+(1/(Ra+Rs))
quantile(Vd, c(.05, 0.10, .50, 0.95))

**Dry deposition parameterization by Zhang et al. (2001)**
**Uncertainty test: Ice/snow**
set.seed(5)
C1 = 0.2789
C2 = 3.115
C3 = 5.415*10^-11
C4 = -1.399
RH = replicate(10000,runif(100,0.76,0.84))
dp_i = 2.0
dp_d = dp_i*10^-6
rd = dp_d/2
r_w = {(C1*rd^C2)/(C3*rd^C4-log10(RH))+rd^3}^(1/3)
dp = r_w*2
**Correction factor, C**
k_B = 1.38*10^-23
Temp = 273.15+0
P = 101325
d_air = 3.72*10^-10
lambda = (k_B*Temp)/(sqrt(2)*3.1416*P*d_air^2)
C = 1+(2*lambda/dp)*(1.257+0.4*exp(-0.55*dp/lambda))
dyn.vis = ((5*10^-8)*Temp)+4*10^-6
rho = 1500
Vg = (rho*(dp)^2*9.81*C)/(18*dyn.vis)
**Compute aerodynamic resistance Ra:**
z = 5
L = replicate(10000,runif(100,45,55))
x = z/L
**Compute shi_H (stability function)**
shi_H.1 = 2*log(0.5*{1+(1-16*x)^0.5})
shi_H.2 = -5*x
shi_H =ifelse(x <= 0, shi_H.1 , shi_H.2)
zR = 10
u_star = replicate(10000,runif(100,0.27,0.33))
z0 = replicate(10000,runif(100,0.0075,0.0125))
k_c = 0.41
Ra = (log(zR/z0)-shi_H)/(k_c*u_star)
**Compute surface resistance Rs:**
e_0 = 3
R1 = 1
kin.vis = ((9*10^-8)*Temp)+10^-5
gamma = 0.54
D = (C*k_B*Temp)/(3*3.1416*dyn.vis*dp)
Sc = (kin.vis/D)
E_B = Sc^(-gamma)
```

```
alpha = 50
beta =  2
A = 2/1000
St = (Vg*u_star^2)/(kin.vis)
E_IM = {St/(alpha+St)}^beta
Rs = 1/{(e_0*u_star)*(E_B+E_IM)*R1}  #(m/s)
**Compute Dry deposition velocity**
Vd =   Vg+(1/(Ra+Rs))
quantile(Vd, c(.05, 0.10, .50, 0.95))
```

**Codes for Monte Carlo uncertainty evaluation for Petroff and Zhang (2010) parameterization**

```
**Dry deposition parameterization by Petroff and Zhang (2010)**
**Uncertainty test: Grass**
set.seed(5)
C1 = 0.2789
C2 = 3.115
C3 = 5.145*10^-11
C4 = -1.399
RH = replicate(10000,runif(100,0.76,0.84))
dp_a = 2.0
dp_i = dp_a*10^-6
rd = dp_i/2
r_w = {(C1*rd^C2)/(C3*rd^C4-log10(RH))+rd^3}^(1/3)
dp = r_w*2
**Correction factor, C**
k_B = 1.38*10^-23
Temp = (273.15+25)
P = 101325
d_air = 3.7208*10^-10
lambda = (k_B*Temp)/(sqrt(2)*3.1416*P*d_air^2)
C = 1+(2*lambda/dp)*(1.257+0.4*exp(-0.55*dp/lambda))
dyn.vis = 1.8908*10^-5
rho = 1500
Tau = (rho*(dp)^2*C)/(18*dyn.vis)
Ws = Tau*9.81
Vphor = 0
Vdrift = Ws+Vphor
**Compute aerodynamic resistance (Ra):**
z = 5
L = replicate(10000,runif(100,45,55))
x = z/L
**Compute stability function (shi_H)**
shi_H.1 = 2*log(0.5*{1+(1-16*x)^0.5})
shi_H.2 = -5*x
shi_H =ifelse(x <= 0, shi_H.1 , shi_H.2)
zR = 3.5
z0 = replicate(10000,runif(100,0.03,0.05))
u_star = replicate(10000,runif(100,0.27,0.33))
k_c = 0.41
Ra = (log(zR/z0)-shi_H)/(k_c*u_star)
**Compute surface resistance (Rs)**
kin.vis = 1.6834*10^-5
D = (C*k_B*Temp)/(3*3.1416*dyn.vis*dp)
Sc = (kin.vis/D)
FSc = (Sc^(1/3))/2.9
Egb = (Sc^(-2/3)/14.5)*{1/6*log(1+FSc)^2/(1-FSc+FSc^2)+1/sqrt(3)*atan((2*FSc-1)/sqrt(3))
       +3.1416/6*sqrt(3)}^-1
cd = 1/6
kx = 0.216
LAI = replicate(10000,runif(100,3.8,4.2))
h = replicate(10000,runif(100,0.475,0.525))
d = replicate(10000,runif(100,0.225,0.375))
phi_H.1 = (1-16*x)^(-0.5)
phi_H.2 = 1+5*x
phi_H = ifelse(x<=0, phi_H.1, phi_H.2)
phi_M.1 = (1-16*x)^(-0.25)
phi_M.2 = 1+5*x
phi_M = ifelse(x<=0, phi_M.1, phi_M.2)
lmp = (0.41*(z-d))/(phi_H*(z-d)/abs(L))
lmp_h = (0.41*(h-d))/(phi_M*(h-d)/abs(L))
alphaPZ = {(kx*LAI)/(12*k_c^2*(1-d/h)^2)}^(1/3)*phi_M^(2/3)*{(h-d)/abs(L)}
C_IT = 0.042
```

```r
Tau_phplus.1 = (Tau*u_star^2)/kin.vis
Tau_phplus.2 = C_IT
Tau_phplus = ifelse(Tau_phplus.1<20,Tau_phplus.1, Tau_phplus.2)
E_t.1 = 2.5*10^-3*C_IT*(Tau_phplus)^2
E_t.2 = C_IT
E_t = ifelse(Tau_phplus.1<20, E_t.1, E_t.2)
u_starf = u_star*exp(-alphaPZ)
Tau_phplus.f1 = (Tau*u_star^2)/kin.vis
Tau_phplus.f2 = 0.14
Tau_phplusf = ifelse(Tau_phplus.f1<20,Tau_phplus.f1, Tau_phplus.f2)
E_gt1 = 2.5*10^-3*0.14*(Tau_phplusf)^2
E_gt2 = 0.14
Egt = ifelse(Tau_phplus.f1<20, E_gt1, E_gt2)
Eg = Egb + Egt
Qg = Eg*h/lmp_h
U_z = replicate(10000,runif(100,2.91,3.09))
U_h = U_z/(exp(alphaPZ*(z/h-1)))
L_obs = 0.005
C_B = 0.7
Re_h = (U_h*L_obs)/(kin.vis)
E_B = C_B*(Sc^(-2/3))*(Re_h^(-1/2))
C_IN = 0.7
E_IN = C_IN*(dp/L_obs)
C_IM = 0.191
beta_IM = 0.60
St_h = (Tau*U_h)/L_obs
E_IM = C_IM*(St_h/(St_h+beta_IM))^2
E_T = (U_h/u_star)*(E_B+E_IN+E_IM)+E_t
Q   = LAI*E_T*h/(lmp_h)
etaPZ = (alphaPZ^2/4+Q)^0.5
Vds = u_star*Eg*{(1+Q/Qg-alphaPZ/2)*tanh(etaPZ)/etaPZ}/{(1+Q+alphaPZ/2)*tanh(etaPZ)/etaPZ}
Vd = Vdrift+1/(Ra+1/Vds)
quantile(Vd, c(.05, 0.10, .50, 0.95))

**Dry deposition parameterization by Petroff and Zhang (2010)**
**Uncertainty test: Coniferous forest**
set.seed(5)
C1 = 0.2789
C2 = 3.115
C3 = 5.145*10^-11
C4 = -1.399
RH = replicate(10000,runif(100,0.76,0.84))
dp_a = 50
dp_i = dp_a*10^-6
rd = dp_i/2
r_w = {(C1*rd^C2)/(C3*rd^C4-log10(RH))+rd^3}^(1/3)
dp = r_w*2
**Correction factor, C**
k_B = 1.38*10^-23
Temp = (273.15+25)
P = 101325
d_air = 3.7208*10^-10
lambda = (k_B*Temp)/(sqrt(2)*3.1416*P*d_air^2)
C = 1+(2*lambda/dp)*(1.257+0.4*exp(-0.55*dp/lambda))
dyn.vis = 1.8908*10^-5
rho = 1500
Tau = (rho*(dp)^2*C)/(18*dyn.vis)
Ws = Tau*9.81
Vphor = 0
Vdrift = Ws+Vphor
**Compute aerodynamic resistance (Ra):**
z = 35
L = replicate(10000,runif(100,180,200))
x = z/L
**Compute stability function (shi_H)**
shi_H.1 = 2*log(0.5*{1+(1-16*x)^0.5})
shi_H.2 = -5*x
shi_H =ifelse(x <= 0, shi_H.1 , shi_H.2)
zR = 30
z0 = replicate(10000,runif(100,0.9,1.5))
u_star = replicate(10000,runif(100,0.27,0.33))
k_c = 0.41
Ra = (log(zR/z0)-shi_H)/(k_c*u_star)
```

```r
**Compute surface resistance (Rs)**
kin.vis = 1.6834*10^-5
D = (C*k_B*Temp)/(3*3.1416*dyn.vis*dp)
Sc = (kin.vis/D)
FSc = (Sc^(1/3))/2.9
Egb = (Sc^(-2/3)/14.5)*{1/6*log(1+FSc)^2/(1-FSc+FSc^2)+1/sqrt(3)*atan((2*FSc-1)/sqrt(3))
        +3.1416/6*sqrt(3)}^-1
cd = 1/6
kx = 0.216
LAI = replicate(10000,runif(100,9.5,10.5))
h = replicate(10000,runif(100,14.25,15.75))
d = replicate(10000,runif(100,5.25,8.75))
phi_H.1 = (1-16*x)^(-0.5)
phi_H.2 = 1+5*x
phi_H = ifelse(x<=0, phi_H.1, phi_H.2)
phi_M.1 = (1-16*x)^(-0.25)
phi_M.2 = 1+5*x
phi_M = ifelse(x<=0, phi_M.1, phi_M.2)
lmp = (0.41*(z-d))/(phi_H*(z-d)/abs(L))
lmp_h = (0.41*(h-d))/(phi_M*(h-d)/abs(L))
alphaPZ = {(kx*LAI)/(12*k_c^2*(1-d/h)^2)}^(1/3)*phi_M^(2/3)*{(h-d)/abs(L)}
C_IT = 0
Tau_phplus.1 = (Tau*u_star^2)/kin.vis
Tau_phplus.2 = C_IT
Tau_phplus = ifelse(Tau_phplus.1<20,Tau_phplus.1, Tau_phplus.2)
E_t.1 = 2.5*10^-3*C_IT*(Tau_phplus)^2
E_t.2 = C_IT
E_t = ifelse(Tau_phplus.1<20, E_t.1, E_t.2)
u_starf = u_star*exp(-alphaPZ)
Tau_phplus.f1 = (Tau*u_star^2)/kin.vis
Tau_phplus.f2 = 0.14
Tau_phplusf = ifelse(Tau_phplus.f1<20,Tau_phplus.f1, Tau_phplus.f2)
E_gt1 = 2.5*10^-3*0.14*(Tau_phplusf)^2
E_gt2 = 0.14
Egt = ifelse(Tau_phplus.f1<20, E_gt1, E_gt2)
Eg = Egb + Egt
**Compute Qg (non-dimensional number)**
Qg = Eg*h/lmp_h
**Compute Q**
U_z = replicate(10000,runif(100,3.88,4.12))
U_h = U_z/(exp(alphaPZ*(z/h-1)))
**Compute E_B (Brownian diffusion)**
L_obs = 0.0015
C_B = 0.887
Re_h = (U_h*L_obs)/(kin.vis)
E_B = C_B*(Sc^(-2/3))*(Re_h^(-1/2))
**Compute E_IN (Interception)**
C_IN = 0.810
E_IN = C_IN*(dp/L_obs)
**Compute E_IM (Impaction)**
C_IM = 0.162
beta_IM = 0.60
St_h = (Tau*U_h)/L_obs
E_IM = C_IM*(St_h/(St_h+beta_IM))^2
E_T = (U_h/u_star)*(E_B+E_IN+E_IM)+E_t
Q  = LAI*E_T*h/(lmp_h)
**Compute etaPZ**
etaPZ = (alphaPZ^2/4+Q)^0.5
Vds = u_star*Eg*{(1+Q/Qg-alphaPZ/2)*tanh(etaPZ)/etaPZ}/{(1+Q+alphaPZ/2)*tanh(etaPZ)/etaPZ}
Vd = Vdrift+1/(Ra+1/Vds)
quantile(Vd, c(.05, 0.10, .50, 0.95))

**Dry deposition parameterization by Petroff and Zhang (2010)**
**Uncertainty test: Deciduous forest**
set.seed(5)
C1 = 0.2789
C2 = 3.115
C3 = 5.145*10^-11
C4 = -1.399
RH = replicate(10000,runif(100,0.76,0.84))
dp_a = 2.0
dp_i = dp_a*10^-6
rd = dp_i/2
```

```r
r_w = {(C1*rd^C2)/(C3*rd^C4-log10(RH))+rd^3}^(1/3)
dp = r_w*2
**Correction factor, C**
k_B = 1.38*10^-23
Temp = (273.15+25)
P = 101325
d_air = 3.7208*10^-10
lambda = (k_B*Temp)/(sqrt(2)*3.1416*P*d_air^2)
C = 1+(2*lambda/dp)*(1.257+0.4*exp(-0.55*dp/lambda))
dyn.vis = 1.8908*10^-5
rho = 1500
Tau = (rho*(dp)^2*C)/(18*dyn.vis)
Ws = Tau*9.81
Vphor = 0
Vdrift = Ws+Vphor
**Compute aerodynamic resistance (Ra):**
z = 35
L = replicate(10000,runif(100,45,55))
x = z/L
**Compute stability function (shi_H)**
shi_H.1 = 2*log(0.5*{1+(1-16*x)^0.5})
shi_H.2 = -5*x
shi_H =ifelse(x <= 0, shi_H.1 , shi_H.2)
zR = 50
z0 = replicate(10000,runif(100,1.125, 1.875))
u_star = replicate(10000,runif(100,0.54,0.66))
k_c = 0.41
Ra = (log(zR/z0)-shi_H)/(k_c*u_star)
**Compute surface resistance (Rs)**
kin.vis = 1.6834*10^-5
D = (C*k_B*Temp)/(3*3.1416*dyn.vis*dp)
Sc = (kin.vis/D)
FSc = (Sc^(1/3))/2.9
Egb = (Sc^(-2/3)/14.5)*{1/6*log(1+FSc)^2/(1-FSc+FSc^2)+1/sqrt(3)*atan((2*FSc-1)/sqrt(3))
        +3.1416/6*sqrt(3)}^-1
cd = 1/6
kx = 0.216
LAI = replicate(10000,runif(100,9.5,10.5))
h = replicate(10000,runif(100,23.75,26.25))
d = replicate(10000,runif(100,12,20))
phi_H.1 = (1-16*x)^(-0.5)
phi_H.2 = 1+5*x
phi_H = ifelse(x<=0, phi_H.1, phi_H.2)
phi_M.1 = (1-16*x)^(-0.25)
phi_M.2 = 1+5*x
phi_M = ifelse(x<=0, phi_M.1, phi_M.2)
lmp = (0.41*(z-d))/(phi_H*(z-d)/abs(L))
lmp_h = (0.41*(h-d))/(phi_M*(h-d)/abs(L))
alphaPZ = {(kx*LAI)/(12*k_c^2*(1-d/h)^2)}^(1/3)*phi_M^(2/3)*{(h-d)/abs(L)}
C_IT = 0.056
Tau_phplus.1 = (Tau*u_star^2)/kin.vis
Tau_phplus.2 = C_IT
Tau_phplus = ifelse(Tau_phplus.1<20,Tau_phplus.1, Tau_phplus.2)
E_t.1 = 2.5*10^-3*C_IT*(Tau_phplus)^2
E_t.2 = C_IT
E_t = ifelse(Tau_phplus.1<20, E_t.1, E_t.2)
u_starf = u_star*exp(-alphaPZ)
Tau_phplus.f1 = (Tau*u_star^2)/kin.vis
Tau_phplus.f2 = 0.14
Tau_phplusf = ifelse(Tau_phplus.f1<20,Tau_phplus.f1, Tau_phplus.f2)
E_gt1 = 2.5*10^-3*0.14*(Tau_phplusf)^2
E_gt2 = 0.14
Egt = ifelse(Tau_phplus.f1<20, E_gt1, E_gt2)
Eg = Egb + Egt
**Compute Qg (non-dimensional number)**
Qg = Eg*h/lmp_h
U_z = replicate(10000,runif(100,3.88,4.12))
U_h = U_z/(exp(alphaPZ*(z/h-1)))
**Compute E_B (Brownian diffusion)**
L_obs = 0.03
C_B = 1.262
Re_h = (U_h*L_obs)/(kin.vis)
E_B = C_B*(Sc^(-2/3))*(Re_h^(-1/2))
**Compute E_IN (Interception)**
```

```
C_IN = 0.216
E_IN = C_IN*(dp/L_obs)*(2+log(4*L_obs/dp))
**Compute E_IM (Impaction)**
C_IM = 0.130
beta_IM = 0.47
St_h = (Tau*U_h)/L_obs
E_IM = C_IM*(St_h/(St_h+beta_IM))^2
E_T = (U_h/u_star)*(E_B+E_IN+E_IM)+E_t
Q   = LAI*E_T*h/(lmp_h)
**Compute etaPZ**
etaPZ = (alphaPZ^2/4+Q)^0.5
Vds = u_star*Eg*{(1+Q/Qg-alphaPZ/2)*tanh(etaPZ)/etaPZ}/{(1+Q+alphaPZ/2)*tanh(etaPZ)/etaPZ}
Vd = Vdrift+1/(Ra+1/Vds)
quantile(Vd, c(.05, 0.10, .50, 0.95))

**Dry deposition parameterization by Petroff and Zhang (2010)**
**Uncertainty test: Water**
set.seed(5)
C1 = 0.2789
C2 = 3.115
C3 = 5.415*10^-11
C4 = -1.399
RH = replicate(10000,runif(100,0.76,0.84))
dp_i = 2.0
dp_d = dp_i*10^-6
rd = dp_d/2
r_w = {(C1*rd^C2)/(C3*rd^C4-log10(RH))+rd^3}^(1/3)
dp = r_w*2
k_B = 1.38*10^-23
Temp = 273.15+25
P = 101325
d_air = 3.7208*10^-10
lambda = (k_B*Temp)/(sqrt(2)*3.1416*P*d_air^2)
C = 1+(2*lambda/dp)*(1.257+0.4*exp(-0.55*dp/lambda))
dyn.vis = ((5*10^-8)*Temp)+4*10^-6
rho = 1500
Tau = (rho*(dp)^2*C)/(18*dyn.vis)
Ws = Tau*9.81
Vphor = (5*10^-3)/100
Vdrift = Ws+Vphor
**Compute aerodynamic resistance (Ra):**
z = 8/100
L = replicate(10000,runif(100,45,55))
x = z/L
**Compute stability function (shi_H)**
shi_H.1 = 2*log(0.5*{1+(1-16*x)^0.5})
shi_H.2 = -5*x
shi_H = ifelse(x <= 0, shi_H.1 , shi_H.2)
zR = 5
u_star = replicate(10000,runif(100,0.27,0.33))
z0_1 = 0.021*(u_star)^3.32
z0_2 = 0.00098*(u_star)^1.65
z0 = ifelse(u_star<= 0.16, z0_1, z0_2)
k_c = 0.41
Ra = (log(zR/z0)-shi_H)/(k_c*u_star)
**Compute surface resistance (Rs)**
kin.vis = ((9*10^-8)*Temp)+10^-5
D = (C*k_B*Temp)/(3*3.1416*dyn.vis*dp)
Sc = (kin.vis/D)
FSc = (Sc^(1/3))/2.9
Egb = (Sc^(-2/3)/14.5)*{1/6*log(1+FSc)^2/(1-FSc+FSc^2)+1/sqrt(3)*atan((2*FSc-
1)/sqrt(3))+3.1416/6*sqrt(3)}^-1
Eg = Egb
Vd = Vdrift+1/(Ra+(1/(Eg*u_star)))
quantile(Vd, c(.05, 0.10, .50, 0.95))

**Dry deposition parameterization by Petroff and Zhang (2010)**
**Uncertainty test: Ice/snow**
set.seed(5)
C1 = 0.2789
C2 = 3.115
C3 = 5.415*10^-11
```

```
C4 = -1.399
RH = replicate(10000,runif(100,0.76,0.84))
dp_i = 2.0
dp_d = dp_i*10^-6
rd = dp_d/2
r_w = {(C1*rd^C2)/(C3*rd^C4-log10(RH))+rd^3}^(1/3)
dp = r_w*2
**Correction factor, C**
k_B = 1.38*10^-23
Temp = 273.15+0
P = 101325
d_air = 3.7208*10^-10
lambda = (k_B*Temp)/(sqrt(2)*3.1416*P*d_air^2)
C = 1+(2*lambda/dp)*(1.257+0.4*exp(-0.55*dp/lambda))
dyn.vis = ((5*10^-8)*Temp)+4*10^-6
rho = 1500
Tau = (rho*(dp)^2*C)/(18*dyn.vis)
Ws = Tau*9.81
Vphor = (2*10^-4)/100
Vdrift = Ws+Vphor
**Compute aerodynamic resistance (Ra)**
z = 5
L = replicate(10000,runif(100,45,55))
x = z/L
**Compute stability function (shi_H)**
shi_H.1 = 2*log(0.5*{1+(1-16*x)^0.5})
shi_H.2 = -5*x
shi_H = ifelse(x <= 0, shi_H.1 , shi_H.2)
zR = 10
u_star = replicate(10000,runif(100,0.27,0.33))
z0 = replicate(10000,runif(100,0.0075,0.0125))
k_c = 0.41
Ra = (log(zR/z0)-shi_H)/(k_c*u_star)
**Compute surface resistance (Rs)**
kin.vis = ((9*10^-8)*Temp)+10^-5
D = (C*k_B*Temp)/(3*3.1416*dyn.vis*dp)
Sc = (kin.vis/D)
FSc = (Sc^(1/3))/2.9
Egb = (Sc^(-2/3)/14.5)*{1/6*log(1+FSc)^2/(1-FSc+FSc^2)+1/sqrt(3)*atan((2*FSc-1)/sqrt(3))+3.1416/6*sqrt(3)}^-1
Eg = Egb
Vd = Vdrift+1/(Ra+(1/(Eg*u_star)))
quantile(Vd, c(.05, 0.10, .50, 0.95))
```

**Codes for Monte Carlo uncertainty evaluation for Kouznetsov and Sofiev (2012) parameterization**

```
**Dry deposition parameterization by Kouznetsov and Sofiev (2012)**
**Uncertainty test: Grass**
set.seed(5)
C1 = 0.2789
C2 = 3.115
C3 = 5.145*10^-11
C4 = -1.399
RH = replicate(10000,runif(100,0.76,0.84))
dp_a =2.0
dp_i = dp_a*10^-6
rd = dp_i/2
r_w = {(C1*rd^C2)/(C3*rd^C4-log10(RH))+rd^3}^(1/3)
dp = r_w*2
**Correction factor, C**
k_B = 1.38*10^-23
Temp = (273.15+25)
P = 101325
d_air = 3.7208*10^-10
lambda = (k_B*Temp)/(sqrt(2)*3.1416*P*d_air^2)
C = 1+(2*lambda/dp)*(1.257+0.4*exp(-0.55*dp/lambda))
dyn.vis = 1.89*10^-5
rho = 1500
Tau = (rho*(dp)^2*C)/(18*dyn.vis)
V_s = Tau*9.81
u_star = replicate(10000,runif(100,0.27,0.33))
a   = 2*10^-3
kin.vis = 1.68*10^-5
D = (C*k_B*Temp)/(3*3.1416*dyn.vis*dp)
```

```
Sc = (kin.vis/D)
Re_star = (u_star*a)/kin.vis
**Compute V_diff (velocity for diffusion)**
V_diff = 2*(Re_star^(-0.5))*Sc^(-2/3)
**Compute V_int (velocity for interception)**
V_int = 80*u_star*((dp/a)^2)*(Re_star^(0.5))
**Compute V_imp**
C_S = 0.003
C_R = 0.3
LAI = replicate(10000,runif(100,3.8,4.2))
CsCR = (C_S+C_R/LAI)^0.5
u.Uh = 0.3
u_star.by.U_h = min(u.Uh, CsCR)
**Compute Re_c**
Re_c = ((u_star.by.U_h)^-1)^2*Re_star
**Calculate St**
St = (Tau*u_star)/a
**Calculate St_e**
St_e = St - Re_c^(-0.5)
eta_impSt.e1 = exp((-0.1/(St_e - 0.15 ) )-(1/sqrt(St_e -0.15)))
eta_impSt.e2 = 0
eta_impSt.e = ifelse(St_e>0.15,eta_impSt.e1,eta_impSt.e2)
V_imp = ((2*u_star.by.U_h)/u_star)*eta_impSt.e*(St-u_star.by.U_h*Re_star^-0.5)
**Dry deposition velocity**
Vd = V_diff+V_int+V_imp+V_s
quantile(Vd, c(.05, 0.10, .50, 0.95))

**Dry deposition parameterization by Kouznetsov and Sofiev (2012)**
**Uncertainty test: Coniferous forest**
set.seed(5)
C1 = 0.2789
C2 = 3.115
C3 = 5.145*10^-11
C4 = -1.399
RH = replicate(10000,runif(100,0.76,0.84))
dp_a = 2.0
dp_i = dp_a*10^-6
rd = dp_i/2
r_w = {(C1*rd^C2)/(C3*rd^C4-log10(RH))+rd^3}^(1/3)
dp = r_w*2
**Correction factor, C**
k_B = 1.38*10^-23
Temp =  273.15+25
P = 101325
d_air = 3.7208*10^-10
lambda = (k_B*Temp)/(sqrt(2)*3.1416*P*d_air^2)
C = 1+(2*lambda/dp)*(1.257+0.4*exp(-0.55*dp/lambda))
dyn.vis = 1.891*10^-5
rho = 1500
Tau = (rho*(dp)^2*C)/(18*dyn.vis)
V_s = Tau*9.81
u_star = replicate(10000,runif(100,0.27,0.33))
a   = 0.7*10^-3
kin.vis = 1.683*10^-5
D = (C*k_B*Temp)/(3*3.1416*dyn.vis*dp)
Sc = (kin.vis/D)
Re_star = (u_star*a)/kin.vis
**Compute V_diff (velocity for diffusion)**
V_diff = 2*(Re_star^(-0.5))*Sc^(-2/3)
**Compute V_int (velocity for interception)**
V_int = 80*u_star*((dp/a)^2)*(Re_star^(0.5))
**Compute V_imp**
C_S = 0.003
C_R = 0.3
LAI = replicate(10000,runif(100,5.7,6.3))
CsCR = (C_S+C_R/LAI)^0.5
u.Uh = 0.3
u_star.by.U_h = min(u.Uh, CsCR)
**Compute Re_c**
Re_c = ((u_star.by.U_h)^-1)^2*Re_star
**Calculate St**
St = (Tau*u_star)/a
**Calculate St_e**
```

```r
St_e = St - Re_c^(-0.5)
eta_impSt.e1 = exp((-0.1/(St_e - 0.15 ) )-(1/sqrt(St_e -0.15)))
eta_impSt.e2 = 0
eta_impSt.e = ifelse(St_e>0.15,eta_impSt.e1,eta_impSt.e2)
V_imp = ((2*u_star.by.U_h)/u_star)*eta_impSt.e*(St-u_star.by.U_h*Re_star^-0.5)
**Dry deposition velocity**
Vd = V_diff+V_int+V_imp+V_s
quantile(Vd, c(.05, 0.10, .50, 0.95))

**Dry deposition parameterization by Kouznetsov and Sofiev (2012)**
**Uncertainty test: Deciduous forest**
set.seed(5)
C1 = 0.2789
C2 = 3.115
C3 = 5.145*10^-11
C4 = -1.399
RH = replicate(10000,runif(100,0.76,0.84))
dp_a = 2.0
dp_i = dp_a*10^-6
rd = dp_i/2
r_w = {(C1*rd^C2)/(C3*rd^C4-log10(RH))+rd^3}^(1/3)
dp = r_w*2
**Correction factor, C**
k_B = 1.38*10^-23
Temp = (273.15+25)
P = 101325
d_air = 3.7208*10^-10
lambda = (k_B*Temp)/(sqrt(2)*3.1416*P*d_air^2)
C = 1+(2*lambda/dp)*(1.257+0.4*exp(-0.55*dp/lambda))
dyn.vis = 1.89*10^-5
rho = 1500
Tau = (rho*(dp)^2*C)/(18*dyn.vis)
V_s = Tau*9.81
u_star = replicate(10000,runif(100,0.27,0.33))
a   = 7*10^-3
kin.vis = 1.68*10^-5
D = (C*k_B*Temp)/(3*3.1416*dyn.vis*dp)
Sc = (kin.vis/D)
Re_star = (u_star*a)/kin.vis
**Compute V_diff (velocity for diffusion)**
V_diff = 2*(Re_star^(-0.5))*Sc^(-2/3)
**Compute V_int (velocity for interception)**
V_int = 80*u_star*((dp/a)^2)*(Re_star^(0.5))
**Compute V_imp**
C_S = 0.003
C_R = 0.3
LAI = replicate(10000,runif(100,9.5,10.5))
CsCR = (C_S+C_R/LAI)^0.5
u.Uh = 0.3
u_star.by.U_h = min(u.Uh, CsCR)
**Compute Re_c**
Re_c = ((u_star.by.U_h)^-1)^2*Re_star
**Calculate St**
St = (Tau*u_star)/a
**Calculate St_e**
St_e = St - Re_c^(-0.5)
eta_impSt.e1 = exp((-0.1/(St_e - 0.15 ) )-(1/sqrt(St_e -0.15)))
eta_impSt.e2 = 0
eta_impSt.e = ifelse(St_e>0.15,eta_impSt.e1,eta_impSt.e2)
V_imp = ((2*u_star.by.U_h)/u_star)*eta_impSt.e*(St-u_star.by.U_h*Re_star^-0.5)
**Dry deposition velocity**
Vd = V_diff+V_int+V_imp+V_s
quantile(Vd, c(.05, 0.10, .50, 0.95))

**Dry deposition parameterization by Kouznetsov and Sofiev (2012)**
**Uncertainty test: Smooth surface (water)**
set.seed(5)
C1 = 0.2789
C2 = 3.115
C3 = 5.415*10^-11
C4 = -1.399
dp_i = 0.5
```

```
dp = dp_i*10^-6
rho = 1500
RH = replicate(10000,runif(100,0.76,0.84))
Temp = 25+273.15
u_star = replicate(10000,runif(100,0.27,0.33))
z0_1 = 0.021*(u_star)^3.32
z0_2 = 0.00098*(u_star)^1.65
z0 = ifelse(u_star<= 0.16, z0_1, z0_2)
L = replicate(10000,runif(100,45,55))
kin.vis = ((9*10^-8)*Temp)+10^-5
dyn.vis = ((5*10^-8)*Temp)+4*10^-6
d_air = 3.7208*10^-10
k_B = 1.38*10^-23
P = 101325
lambda = (k_B*Temp)/(sqrt(2)*3.1416*P*d_air^2)
C = 1+((2*lambda/dp)*(1.257+0.4*exp(-0.55*dp/lambda)))
tau_p = (rho*(dp)^2*C)/(18*dyn.vis)
D = (C*k_B*Temp)/(3*3.1416*dyn.vis*dp)
v_s = 9.81*tau_p
taup = (tau_p*u_star^2)/(kin.vis)
Sc = (kin.vis/D)
vsplus = v_s/u_star
rplus = (dp*u_star)/(2*kin.vis)
R_s = 0
Rsplus = u_star*R_s
z_meas = 8/100
zpmax = (z_meas*u_star)/(kin.vis)
MOplus = (kin.vis)/(u_star*L)
**Fixed parameters**
Zbuf = 3
Ztf = 18                            #turbophoretic sublayer height
taultf = 5                          #Lagrangian time in turbophoretic layer
Nutp_Ztf= (0.4*(Ztf)^3)/(Ztf^2+200) #Dimensionless eddy viscosity of air
It_Ztf = (2.5*log10(Ztf ))-(100/Ztf^2)
It_Zbuf = (2.5*log10(Zbuf))-(100/Zbuf^2)
S = Sc^(1/3)
Zl = 20/S
fTmp = 2.5/Sc
fTmp1 = (fTmp^3/27+(fTmp*(100+5*sqrt(8*fTmp/27)+400)))^(1/3)
Zl_1 = fTmp1+((fTmp*fTmp)/(9*fTmp1))+(1/3)*fTmp
fTmp_1 = (Zl_1^2)/(Zl_1^2+200)
fTmp_2 = 1.2*(fTmp_1)-0.8*fTmp_1^2
fTmp1_1 = 1/(Sc*fTmp_2)
fTmp_3 = 1/Sc
x_1 = Zl_1 - fTmp1_1
x_2 = Zl_1
x_3 = Zl_1+fTmp1_1
Nutp_x_1 = (0.4*x_1^3)/(x_1^2+200)
Nutp_x_2 = (0.4*x_2^3)/(x_2^2+200)
Nutp_x_3 = (0.4*x_3^3)/(x_3^2+200)
fIvd = Rsplus+(Zl_1-
fTmp1_1)*Sc+0.3333*fTmp1_1*(1/fTmp_3+((0.4*x_1^3)/(x_1^2+200)))+4/(fTmp_3+((0.4*x_2^3)/(x_2^2+200)))+1/(fTmp_3+(
(0.4*x_3^3)/(x_3^2+200)))
x_4 = zpmax
x_5 = Zl_1 + fTmp1_1
It_x_4 = 2.5*log10(x_4) - 100/(x_4^2)
It_x_5 = 2.5*log10(x_5) - 100/(x_5^2)
**Now calculate fu_Psi(zpmax*MOplus)**
s = 2.35*(zpmax*MOplus+abs(zpmax*MOplus))
u1 = 0.5*(abs(zpmax*MOplus)-zpmax*MOplus)
u = -4*u1/(2.65*sqrt(u1*sqrt(u1))+1)
fu_Psi = s+u
fTmp_4 = It_x_4 - It_x_5 + 2.5*fu_Psi
fIvd_1 = fIvd+0.5*(fTmp_4+abs(fTmp_4))
fu_vdplus_smooth_1 = 1/fIvd_1
Il_input1= Zl*S/7.92
Il_input2= rplus*S/7.92
Il_1 = -0.16667*log10(Il_input1^2-Il_input1+1)+0.57735*atan((2*Il_input1-1)*0.57735)+0.3333*log10(Il_input1+1)
Il_2 = -0.16667*log10(Il_input2^2-Il_input2+1)+0.57735*atan((2*Il_input2-1)*0.57735)+0.3333*log10(Il_input2+1)
R = 7.92*S^2*(Il_1-Il_2)
R_1 = 0.5*(R+abs(R))
Zl_2 = 0.5*(rplus+Zl+abs(rplus-Zl))
It_Zbuf = 2.5*log10(Zbuf) - (100/(Zbuf^2))
It_Zl_2 = 2.5*log10(Zl_2) - (100/(Zl_2^2))
```

```r
R1 = It_Zbuf - It_Zl_2
R_2 = R_1+0.5*(R1+abs(R1))
fTmp_5 = vsplus*R_2
fTmp_6 = exp(-fTmp_5)
fIvd_2  = Rsplus*fTmp_6+(1-fTmp_6)/vsplus
fIvd_3 = Rsplus+R_2
fIvd_23 = ifelse(abs(fTmp_5)>0.001, fIvd_2, fIvd_3 )
V   = 0.81*taup/(Ztf-Zbuf)/(1+taup/taultf) + vsplus    #chcek for sign of vsplus
It_Ztf = 2.5*log10(Ztf) - 100/(Ztf^2)
R_3  = ((It_Ztf)-(It_Zbuf))*(1+taup/taultf)
fTmp_7 = V*R_3
fTmp_8 = exp(-fTmp_7)
fIvd_4 = (fIvd_23*fTmp_8)+(1-fTmp_8)/V
fIvd_5 = fIvd_3+R_3
fIvd_45 = ifelse(abs(fTmp_7)>0.001, fIvd_4, fIvd_5)
**Now calculations for the Lagrangian turbophoretic layer**
Ztf2 = 2*taup
V_1 = 0.4+vsplus
R_4 =
0.1667*((1+taup/(0.5*Ztf))/(Nutp_Ztf)+4*(1+taup/(0.25*(Ztf+Ztf2)))/((0.4*(0.5*(Ztf+Ztf2))^3)/((0.5*(Ztf+Ztf2))^2
+200))+(1+taup/(0.5*Ztf2))/((0.4*Ztf2 ^3)/(Ztf2^2+200)))*(Ztf2-Ztf)
fTmp_9 = V_1*R_4
fTmp_10 = exp(-fTmp_9)
fIvd_6 = fIvd_45*fTmp_10 + (1-fTmp_10)/V_1
fIvd_7 = fIvd_45+R_3
fIvd_67 = ifelse(abs(fTmp_9) > 0.001, fIvd_6, fIvd_7)
Ztf2_2 = Ztf
fIvd67_Ztf2 = ifelse(Ztf2>Ztf, fIvd_67, Ztf2_2)
**Following calculations are for aerodynamic layer**
It_zpmax = 2.5*log10(zpmax) - 100/(zpmax^2)
Ztf22 = ifelse(Ztf2>Ztf, Ztf2, Ztf2_2)
It_Ztf22  = 2.5*log10(Ztf22) - 100/(Ztf22^2)
R_5 = It_zpmax - It_Ztf22  + 2.5*fu_Psi
R_6 = 0.5*(R_5+abs(R_5))
fTmp_11  = vsplus*R_6
fTmp_12 = exp(-fTmp_11)
fIvd_4567 = ifelse(Ztf2>Ztf, fIvd_67,fIvd_45)
fIvd_8 = (fIvd_4567* fTmp_12)+(1-fTmp_12)/vsplus
fIvd_9 = fIvd_4567 + R_6
fIvd_10 = ifelse(abs(fTmp_11) > 0.001, fIvd_8, fIvd_9)
fu_vdplus_smooth_3 = 1/fIvd_10
fu_vdplus_smooth = ifelse(Zl>Zbuf, fu_vdplus_smooth_1, fu_vdplus_smooth_3)
Vd_smooth = fu_vdplus_smooth*u_star
quantile(Vd_smooth, c(.05, 0.10, .50, 0.95))

**Dry deposition parameterization by Kouznetsov and Sofiev (2012)**
**Uncertainty test: Ice/snow**
set.seed(5)
C1 = 0.2789
C2 = 3.115
C3 = 5.415*10^-11
C4 = -1.399
RH = replicate(10000,runif(100,0.76,0.84))
dp_a = 2.0
dp_i = dp_a*10^-6
rd = dp_i/2
r_w = {(C1*rd^C2)/(C3*rd^C4-log10(RH))+rd^3}^(1/3)
dp = r_w*2
**Correction factor, C**
k_B = 1.38*10^-23
Temp = 273.15+0
P = 101325
d_air = 3.7208*10^-10
lambda = (k_B*Temp)/(sqrt(2)*3.1416*P*d_air^2)
C = 1+(2*lambda/dp)*(1.257+0.4*exp(-0.55*dp/lambda))
dyn.vis = ((5*10^-8)*Temp)+4*10^-6
rho =  1500
Tau = (rho*(dp)^2*C)/(18*dyn.vis)
V_s = Tau*9.81
**Need to compute Sc, Re_star**
u_star = replicate(10000,runif(100,0.27,0.33))
**u_star = replicate(10000,runif(100,0.27,0.33))**
a   = 0.5*10^-3
```

```
kin.vis = ((9*10^-8)*Temp)+10^-5
D = (C*k_B*Temp)/(3*3.1416*dyn.vis*dp)
Sc = (kin.vis/D)
Re_star = (u_star*a)/kin.vis
**Compute V_diff (velocity for diffusion)**
V_diff = 2*(Re_star^(-0.5))*Sc^(-2/3)
**Compute V_int (velocity for interception)**
V_int = 80*u_star*((dp/a)^2)*(Re_star^(0.5))
**Compute V_imp**
C_S = 0.003
C_R = 0.3
LAI = 0
CsCR = (C_S+C_R/LAI)^0.5
u.Uh = 0.3
u_star.by.U_h = min(u.Uh, CsCR)
**Compute Re_c**
Re_c = ((u_star.by.U_h)^-1)^2*Re_star
**Calculate St**
St = (Tau*u_star)/a
**Calculate St_e**
St_e = St - Re_c^(-0.5)
eta_impSt.e1 = exp((-0.1/(St_e - 0.15 ) )-(1/sqrt(St_e -0.15)))
eta_impSt.e2 = 0
eta_impSt.e = ifelse(St_e>0.15,eta_impSt.e1,eta_impSt.e2)
V_imp = (2*u_star.by.U_h*eta_impSt.e*(St-(u_star.by.U_h*Re_star^-0.5)))*u_star
**Dry deposition velocity**
Vd = V_diff+V_int+V_imp+V_s
quantile(Vd, c(.05, 0.10, .50, 0.95))
```

**Codes for Monte Carlo uncertainty evaluation for Zhang and He (2014) parameterization**

```
**Dry deposition parameterization by Zhang and He (2014)**
**Uncertainty test: Grass**
set.seed(5)
C1 = 0.2789
C2 = 3.115
C3 = 5.145*10^-11
C4 = -1.399
RH = replicate(10000,runif(100,0.76,0.84))
dp_a = 2.0
dp_i = dp_a*10^-6
rd = dp_i/2
r_w = {(C1*rd^C2)/(C3*rd^C4-log10(RH))+rd^3}^(1/3)
dp = r_w*2
**Correction factor, C**
k_B = 1.38*10^-23
Temp = (273.15+25)
P = 101325
d_air = 3.7208*10^-10
lambda = (k_B*Temp)/(sqrt(2)*3.1416*P*d_air^2)
C = 1+(2*lambda/dp)*(1.257+0.4*exp(-0.55*dp/lambda))
dyn.vis = 1.89*10^-5
rho = 1500
Tau = (rho*(dp)^2*C)/(18*dyn.vis)
V_g = Tau*9.81
Rg = 1/V_g
u_star = replicate(10000,runif(100,0.27,0.33))
a1 = 5.4*10^-3
z = 5
L = replicate(10000,runif(100,45,55))
x = z/L
**Compute stability function (shi_H)**
shi_H.1 = 2*log(0.5*{1+(1-16*x)^0.5})
shi_H.2 = -5*x
shi_H =ifelse(x <= 0, shi_H.1 , shi_H.2)
zR = 3.5
z0 = replicate(10000,runif(100,0.03,0.05))
k_c = 0.41
Ra = (log(zR/z0)-shi_H)/(k_c*u_star)
**Calculate Vds = 1/Rs**
**For PM2.5**
Vds_PM2.5 = (a1*u_star)
Rds_PM2.5 = (1/Vds_PM2.5)
```

```
**For PM2.5-10**
**b1=**
**b2=**
**b3 =**
**c1=**
**c2 =**
**c3 =**
**k = c1*u_star+c2*u_star^2+c3*u_star^3**
**LAI = replicate(10000,runif(100,3.8,4.2))**
**LAImax =**
**Vds_PM10 = (b1*u_star+b2*u_star^2+b3*u_star^3)*exp(k*(LAI/LAImax)-1)**
**Vds_PM10 = (b1*u_star+b2*u_star^2+b3*u_star^3)**
**Rds_PM10 = 1/Vds_PM10**
**For PM10+**
**d1=**
**d2=**
**d3 =**
**f1=**
**f2 =**
**f3 =**
**k = f1*u_star+f2*u_star^2+f3*u_star^3**
**LAI =**
**LAImax =**
**Vds_10plus = (d1*u_star+d2*u_star^2+d3*u_star^3)*exp(k*(LAI/LAImax)-1)**
**Vds_10plus = (d1*u_star+d2*u_star^2+d3*u_star^3)**
**Rds_PM2.5 = (1/Vds_PM2.5)**
**Compute Vd**
Vd = 1/Rg+(1/(Ra+Rds_PM2.5))
quantile(Vd, c(.05, 0.10, .50, 0.95))

**Dry deposition parameterization by Zhang and He (2014)**
**Uncertainty test: Coniferous forest**
set.seed(5)
C1 = 0.2789
C2 = 3.115
C3 = 5.145*10^-11
C4 = -1.399
RH = replicate(10000,runif(100,0.76,0.84))
dp_a = 2.0
dp_i = dp_a*10^-6
rd = dp_i/2
r_w = {(C1*rd^C2)/(C3*rd^C4-log10(RH))+rd^3}^(1/3)
dp = r_w*2
**Correction factor, C**
k_B = 1.38*10^-23
Temp = (273.15+25)
P = 101325
d_air = 3.7208*10^-10
lambda = (k_B*Temp)/(sqrt(2)*3.1416*P*d_air^2)
C = 1+(2*lambda/dp)*(1.257+0.4*exp(-0.55*dp/lambda))
dyn.vis = 1.89*10^-5
rho = 1500
Tau = (rho*(dp)^2*C)/(18*dyn.vis)
V_g = Tau*9.81
Rg = 1/V_g
u_star = replicate(10000,runif(100,0.27,0.33))
a1 = 4.3*10^-3
z = 35
L = replicate(10000,runif(100,45,55))
x = z/L
**Compute stability function (shi_H)**
shi_H.1 = 2*log(0.5*{1+(1-16*x)^0.5})
shi_H.2 = -5*x
shi_H =ifelse(x <= 0, shi_H.1 , shi_H.2)
zR = 30
z0 = replicate(10000,runif(100,0.9,1.5))
k_c = 0.41
Ra = (log(zR/z0)-shi_H)/(k_c*u_star)
**Calculate Vds = 1/Rs**
**For PM2.5**
Vds_PM2.5 = (a1*u_star)
Rds_PM2.5 = (1/Vds_PM2.5)
**For PM2.5-10**
```

```r
**b1=**
**b2=**
**b3 =**
**c1=**
**c2 =**
**c3 =**
**k = c1*u_star+c2*u_star^2+c3*u_star^3**
**LAI =**
**LAImax =**
**Vds_PM10 =**
(b1*u_star+b2*u_star^2+b3*u_star^3)*exp(k*(LAI/LAImax)-
1)
**Vds_PM10 = (b1*u_star+b2*u_star^2+b3*u_star^3)**
**Rds_PM10 = 1/Vds_PM10**
**For PM10+**
**d1=**
**d2=**
**d3 =**
**f1=**
**f2 =**
**f3 =**
**k = f1*u_star+f2*u_star^2+f3*u_star^3**
**LAI =**
**LAImax =**
**Vds_10plus =**
(d1*u_star+d2*u_star^2+d3*u_star^3)*exp(k*(LAI/LAImax)-
1)
**Vds_10plus = (d1*u_star+d2*u_star^2+d3*u_star^3)**
Rds_PM10plus = (1/Vds_PM2.5)
**Compute Vd**
Vd = 1/Rg+(1/(Ra+Rds_PM2.5))
quantile(Vd, c(.05, 0.10, .50, 0.95))

**Dry deposition parameterization by Zhang and He (2014)**
**Uncertainty test: Deciduous forest**
set.seed(5)
C1 = 0.2789
C2 = 3.115
C3 = 5.145*10^-11
C4 = -1.399
RH = replicate(10000,runif(100,0.76,0.84))
dp_a = 2.0
dp_i = dp_a*10^-6
rd = dp_i/2
r_w = {(C1*rd^C2)/(C3*rd^C4-log10(RH))+rd^3}^(1/3)
dp = r_w*2
**Correction factor, C**
k_B = 1.38*10^-23
Temp = (273.15+25)
P = 101325
d_air = 3.7208*10^-10
lambda = (k_B*Temp)/(sqrt(2)*3.1416*P*d_air^2)
C = 1+(2*lambda/dp)*(1.257+0.4*exp(-0.55*dp/lambda))
dyn.vis = 1.89*10^-5
rho = 1500
Tau = (rho*(dp)^2*C)/(18*dyn.vis)
V_g = Tau*9.81 # m/s
Rg = 1/V_g
u_star = replicate(10000,runif(100,0.54,0.66))
a1 = 4.3*10^-3
z = 35
L = replicate(10000,runif(100,45,55))
x = z/L
**Compute stability function (shi_H)**
shi_H.1 = 2*log(0.5*{1+(1-16*x)^0.5})
shi_H.2 = -5*x
shi_H =ifelse(x <= 0, shi_H.1 , shi_H.2)
zR = 50
z0 = replicate(10000,runif(100,1.125, 1.875))
k_c = 0.41
Ra = (log(zR/z0)-shi_H)/(k_c*u_star)
**Calculate Vds = 1/Rs**
**For PM2.5**
```

```r
Vds_PM2.5 = (a1*u_star)
Rds_PM2.5 = (1/Vds_PM2.5)
**For PM2.5-10**
**b1=**
**b2=**
**b3 =**
**c1=**
**c2 =**
**c3 =**
**k = c1*u_star+c2*u_star^2+c3*u_star^3**
**LAI =**
**LAImax =**
**Vds_PM10 =**
(b1*u_star+b2*u_star^2+b3*u_star^3)*exp(k*(LAI/LAImax)-
1)
**Rds_PM10 = 1/Vds_PM10**
**For PM10+**
**d1= -2.2**
**d2= 3.9*10^1**
**d3 = -6.7**
**f1= 6.2**
**f2 = -1.2*10^1**
**f3 = 6.1**
**k = f1*u_star+f2*u_star^2+f3*u_star^3**
**LAI =**
**LAImax =**
**Vds_10plus =**
(d1*u_star+d2*u_star^2+d3*u_star^3)*exp(k*(LAI/LAImax)-
1)
**Rds_10plus = (1/Vds_10plus)**
**Compute Vd**
Vd = 1/Rg+(1/(Ra+Rds_PM2.5))
quantile(Vd, c(.05, 0.10, .50, 0.95))

**Dry deposition parameterization by Zhang and He (2014)**
**Uncertainty test: Water**
set.seed(5)
C1 = 0.2789
C2 = 3.115
C3 = 5.415*10^-11
C4 = -1.399
RH = replicate(10000,runif(100,0.76,0.84))
dp_a =2.0
dp_i = dp_a*10^-6
rd = dp_i/2
r_w = {(C1*rd^C2)/(C3*rd^C4-log10(RH))+rd^3}^(1/3)
dp = r_w*2
**Correction factor, C**
k_B = 1.38*10^-23
Temp = 273.15+25
P = 101325
d_air = 3.7208*10^-10
lambda = (k_B*Temp)/(sqrt(2)*3.1416*P*d_air^2)
C = 1+(2*lambda/dp)*(1.257+0.4*exp(-0.55*dp/lambda))
dyn.vis = ((5*10^-8)*Temp)+4*10^-6
rho = 1500
Tau = (rho*(dp)^2*C)/(18*dyn.vis)
V_g = Tau*9.81
Rg = 1/V_g
a1 = 6.9*10^-3
z = 8/100
L = replicate(10000,runif(100,45,55))
x = z/L
**Compute stability function (shi_H)**
shi_H.1 = 2*log(0.5*{1+(1-16*x)^0.5})
shi_H.2 = -5*x
shi_H =ifelse(x <= 0, shi_H.1 , shi_H.2)
zR = 5
u_star = replicate(10000,runif(100,0.27,0.33))
z0_1 = 0.021*(u_star)^3.32
z0_2 = 0.00098*(u_star)^1.65
z0 = ifelse(u_star<= 0.16, z0_1, z0_2)
k_c = 0.41
```

```r
Ra = (log(zR/z0)-shi_H)/(k_c*u_star)
**Calculate Vds = 1/Rs**
**For PM2.5**
Vds_PM2.5 = (a1*u_star)
Rds_PM2.5 = (1/Vds_PM2.5)
**For PM2.5-10**
**b1= 2.6*10^-1**
**b2= -1.3*10^0**
**b3 = 3.0*10^0**
**c1= 1.8**
**c2 = -2.0*10^-1**
**c3 = -5.3*10^-1**
**k = c1*u_star+c2*u_star^2+c3*u_star^3**
**LAI =**
**LAImax =**
**Vds_PM10 =**
(b1*u_star+b2*u_star^2+b3*u_star^3)*exp(k*(LAI/LAImax)-
1)
**Rds_PM10 = 1/Vds_PM10**
**For PM10+**
**d1= -**
**d2=**
**d3 =**
**f1=**
**f2 =**
**f3 =**
**k = f1*u_star+f2*u_star^2+f3*u_star^3**
**LAI =**
**LAImax =**
**Vds_PM10Plus =**
(d1*u_star+d2*u_star^2+d3*u_star^3)*exp(k*(LAI/LAImax)-
1)
**Rds_PM10Plus = (1/Vds_PM10Plus)**
**Compute Vd**
Vd = 1/Rg+(1/(Ra+Rds_PM2.5))
quantile(Vd, c(.05, 0.10, .50, 0.95))

**Dry deposition parameterization by Zhang and He (2014)**
**Uncertainty test: Ice/Water**
set.seed(5)
C1 = 0.2789
C2 = 3.115
C3 = 5.415*10^-11
C4 = -1.399
RH = replicate(10000,runif(100,0.76,0.84))
dp_a = 2.0
dp_i = dp_a*10^-6
rd = dp_i/2
r_w = {(C1*rd^C2)/(C3*rd^C4-log10(RH))+rd^3}^(1/3)
dp = r_w*2
**Correction factor, C**
k_B = 1.38*10^-23
Temp = 273.15+0
P = 101325
d_air = 3.7208*10^-10
lambda = (k_B*Temp)/(sqrt(2)*3.1416*P*d_air^2)
C = 1+(2*lambda/dp)*(1.257+0.4*exp(-0.55*dp/lambda))
dyn.vis = ((5*10^-8)*Temp)+4*10^-6
rho = 1500
Tau = (rho*(dp)^2*C)/(18*dyn.vis)
V_g = Tau*9.81
Rg = 1/V_g
a1 = 4.3*10^-3
z = 5
L = replicate(10000,runif(100,45,55))
x = z/L
**Compute stability function (shi_H)**
shi_H.1 = 2*log(0.5*{1+(1-16*x)^0.5})
shi_H.2 = -5*x
shi_H =ifelse(x <= 0, shi_H.1 , shi_H.2)
zR = 10
u_star = replicate(10000,runif(100,0.27,0.33))
z0 = replicate(10000,runif(100,0.0075,0.0125))
```

```
k_c = 0.41
Ra = (log(zR/z0)-shi_H)/(k_c*u_star)
**Calculate Vds = 1/Rs#For PM2.5**
Vds_PM2.5 = (a1*u_star)
Rds_PM2.5 = (1/Vds_PM2.5)
**For PM2.5-10**
**b1=**
**b2=**
**b3 =**
**c1=**
**c2 =**
**c3 =**
**k = c1*u_star+c2*u_star^2+c3*u_star^3**
**LAI =**
**LAImax =**
**Vds_PM10 = (b1*u_star+b2*u_star^2+b3*u_star^3)*exp(k*(LAI/LAImax)-1)**
**Rds_PM10 = 1/Vds_PM10**
**For PM10+**
**d1=**
**d2=**
**d3 =**
**f1=**
**f2 =**
**f3 =**
**k = f1*u_star+f2*u_star^2+f3*u_star^3**
**LAI =**
**LAImax =**
**Vds_PM10Plus = (d1*u_star+d2*u_star^2+d3*u_star^3)*exp(k*(LAI/LAImax)-1)**
**Rds_PM10Plus = (1/Vds_PM10Plus)**
**Compute Vd**
Vd = 1/Rg+(1/(Ra+Rds_PM2.5))
quantile(Vd, c(.05, 0.10, .50, 0.95))
```

**Codes for Monte Carlo uncertainty evaluation for Zhang and Sao (2014) parameterization**

```
**Dry deposition parameterization by Zhang and Sao (2014)**
**Uncertainty test: Plant (Grass, coniferous, and deciduous forests)**
set.seed(5)
C1 = 0.2789
C2 = 3.115
C3 = 5.145*10^-11
C4 = -1.399
RH = replicate(10000,runif(100,0.76,0.84))
dp_a = 2.0
dp_i = dp_a*10^-6
rd = dp_i/2
r_w = {(C1*rd^C2)/(C3*rd^C4-log10(RH))+rd^3}^(1/3)
dp = r_w*2
**Correction factor, C**
k_B = 1.38*10^-23
Temp = (273.15+15)
P = 101325
d_air = 3.7208*10^-10
lambda = (k_B*Temp)/(sqrt(2)*3.1416*P*d_air^2)
C = 1+(2*lambda/dp)*(1.257+0.4*exp(-0.55*dp/lambda))
dyn.vis = 1.841*10^-5
rho = 1500
Tau = (rho*(dp_i)^2*C)/(18*dyn.vis)
Tau_wet = (rho*(dp)^2*C)/(18*dyn.vis)
Wt = Tau*9.81
Vg = Wt
Vg_wet = Tau_wet*9.81
u_star = replicate(10000,runif(100,0.27,0.33))
k = 0.41
z = 1
zd = 0.20
h_c = 0.23
z0 = replicate(10000,runif(100,0.0015, 0.0025))
B1 = 0.45
Sc_T = (1+(Vg^2/u_star^2))^0.5
Ra = (Sc_T/(k*u_star))*(log((z-zd)/(h_c-zd)))
Rg = 1/Vg
U_h = replicate(10000,runif(100,3.88,4.12))
```

```
kin.vis = 1.593*10^-5
d_c = 0.005
D = (C*k_B*Temp)/(3*3.1416*dyn.vis*dp)
Sc = (kin.vis/D)
Re_h = (U_h*d_c)/(kin.vis)
nB = 0.5
C_B = 0.467
E_B = C_B*Sc^(-2/3)*Re_h^(nB-1)
**Compute impaction collection efficiency (E_IM)**
beta_IM = 0.6
St_h = (Tau*u_star)/d_c
E_IM = (St_h/(St_h+beta_IM))^2
**Compute interception efficiency (E_IN)**
Ain = 150
E_IN = Ain*u_star*(10^(-St_h))*(2*dp/d_c)
**Compute R**
b = 2
R = exp(-b*sqrt(St_h))
**Compute w_dm**
B2 = 3
w_dm = (u_star/U_h*h_c)        #For rough surface
**compute Tau_c/Tau (ratio of stress)**
Beta = 200
C1 = 6
C2 = 0.1
lambda_FAI = 0.4
n_FAI = (lambda_FAI)/(h_c*d_c)
q = (3.1416*d_c^2)/4
eta_BAI = n_FAI*q
lambda_FAIe = ((lambda_FAI)/(1-eta_BAI)^C2)*exp((-C1*lambda_FAI)/(1-eta_BAI)^C2)
Tau_c_BY_Tau = (Beta*lambda_FAIe)/(1+Beta*lambda_FAIe)
**Compute Rs**
E = E_B+E_IM+E_IN
Tau_wetplus = (Tau_wet*u_star^2)/kin.vis
Cd = 1/6
Rs = (R*w_dm*((E*Tau_c_BY_Tau/Cd)+(1+Tau_c_BY_Tau)*Sc^-1+10^(-3/Tau_wetplus))+Vg_wet)^-1
**Compute Vd**
Vd = (Rg+((Rs-Rg)/exp(Ra/Rg)))^-1
quantile(Vd, c(.05, 0.10, .50, 0.95))

**Dry deposition parameterization by Zhang and Sao (2014)**
**Uncertainty test: Water**
set.seed(5)
C1 = 0.2789
C2 = 3.115
C3 = 5.415*10^-11
C4 = -1.399
RH = replicate(10000,runif(100,0.76,0.84))
dp_a = 2.0
dp_i = dp_a*10^-6
rd = dp_i/2
r_w = {(C1*rd^C2)/(C3*rd^C4-log10(RH))+rd^3}^(1/3)
dp = r_w*2
**Correction factor, C**
k_B = 1.38*10^-23
Temp = 273.15+25
P = 101325
d_air = 3.7208*10^-10
lambda = (k_B*Temp)/(sqrt(2)*3.1416*P*d_air^2)
C = 1+(2*lambda/dp)*(1.257+0.4*exp(-0.55*dp/lambda))
dyn.vis = ((5*10^-8)*Temp)+4*10^-6
rho = 1500
Tau = (rho*(dp_i)^2*C)/(18*dyn.vis)
Tau_wet = (rho*(dp)^2*C)/(18*dyn.vis)
Wt = Tau*9.81
Vg = Wt
Vg_wet = Tau_wet*9.81
u_star = replicate(10000,runif(100,0.27,0.33))
k =  0.41
z0 = 0.3/1000
z = 8/100
U_h = replicate(10000,runif(100,4.85,5.15))
zd = 0
```

```
h_c = 30*z0
B1 = 0.45
Sc_T = (1+(Vg^2/u_star^2))^0.5
Ra = (B1*Sc_T/k*u_star)*log(z/z0)          # For smooth surface
Rg = 1/Vg
**Calculate surface resistance (Rs)**
kin.vis = ((9*10^-8)*Temp)+10^-5
d_c = 0.005
D = (C*k_B*Temp)/(3*3.1416*dyn.vis*dp)
Sc = (kin.vis/D)
Re_h = (U_h*d_c)/(kin.vis)
nB = 0.5
C_B = 0.467
E_B = C_B*Sc^(-2/3)*Re_h^(nB-1)
**Compute impaction collection efficiency (E_IM)**
beta_IM = 0.6
St_h = (Tau*u_star)/d_c
E_IM = (St_h/(St_h+beta_IM))^2
**Compute interception efficiency (E_IN)**
Ain = 100
E_IN = Ain*u_star*(10^(-St_h))*(2*dp/d_c)
**Compute R**
b = 2
R = exp(-b*sqrt(St_h))
**Compute w_dm**
B2 = 3
w_dm = B2*u_star                             #For smooth surface
**compute Tau_c/Tau (ratio of stress)**
Beta = 200
C_1 = 6
C_2 = 0.1
lambda_FAI = 0.538
n_FAI = (lambda_FAI)/(h_c*d_c)
q = (3.1416*d_c^2)/4
eta_BAI = n_FAI*q
lambda_FAIe = ((lambda_FAI)/(1-eta_BAI)^C_2)*exp((-C_1*lambda_FAI)/(1-eta_BAI)^C_2)
Tau_c_BY_Tau = (Beta*lambda_FAIe)/(1+Beta*lambda_FAIe)
**Compute Rs**
E = E_B+E_IM+E_IN
Tau_wetplus = (Tau_wet*u_star^2)/kin.vis
Cd = 1/6
Rs = (R*w_dm*((E*Tau_c_BY_Tau/Cd)+(1+Tau_c_BY_Tau)*Sc^-1+10^(-3/Tau_wetplus))+Vg_wet)^-1
**Compute Vd**
Vd = (Rg+((Rs-Rg)/exp(Ra/Rg)))^-1
quantile(Vd, c(.05, 0.10, .50, 0.95))
```

**Codes for Sobol' sensitivity test for Zhang et al. (2001) parameterization**

```
**Dry deposition parameterization by Zhang et al. (2001)**
**Sobol sensitivity test: Grass (the code is similar for other LUCs)**
**Change LUC dependent parameters for other LUCs**
**Change sensitivity ranges for other LUCs**
set.seed(5)
library(sensitivity)
library(boot)
C1 = 0.2789
C2 = 3.115
C3 = 5.415*10^-11
C4 = -1.399
dp_i = 10
dp_d = dp_i*10^-6
rd = dp_d/2
k_B = 1.38*10^-23
Temp = 273.15+25
P = 101325
d_air = 3.72*10^-10
lambda = (k_B*Temp)/(sqrt(2)*3.1416*P*d_air^2)
dyn.vis = ((5*10^-8)*Temp)+4*10^-6
z = 2
zR = 3.5
k_c = 0.41
e_0 = 3
R1 = 1
```

```r
kin.vis = ((9*10^-8)*Temp)+10^-5
gamma = 0.54
alpha = 1.2
beta =  2
A = 2/1000
model <- function (X) (((((X[,2])*(2*{(C1*rd^C2)/(C3*rd^C4-
log10(X[,1]))+rd^3}^(1/3))^2*9.81*(1+(2*lambda/(2*{(C1*rd^C2)/(C3*rd^C4-
log10(X[,1]))+rd^3}^(1/3)))*(1.257+0.4*exp(-0.55*((2*{(C1*rd^C2)/(C3*rd^C4-
log10(X[,1]))+rd^3}^(1/3)))/lambda))))/(18*dyn.vis)))+
   (1/(((((log(zR/(X[,4]))+5*z/(X[,3]))/(k_c*(X[,5])))))+
        (1/{(e_0*(X[,5]))*
                 (((((kin.vis/(((1+(2*lambda/(2*{(C1*rd^C2)/(C3*rd^C4-log10(X[,1]))+rd^3}^(1/3)))*(1.257+0.4*exp(-
0.55*(2*{(C1*rd^C2)/(C3*rd^C4-
log10(X[,1]))+rd^3}^(1/3)))/lambda)))*k_B*Temp)/(3*3.1416*dyn.vis*(2*{(C1*rd^C2)/(C3*rd^C4-
log10(X[,1]))+rd^3}^(1/3))))))^(-gamma))+
                   ({(((((X[,2])*(2*{(C1*rd^C2)/(C3*rd^C4-
log10(X[,1]))+rd^3}^(1/3))^2*9.81*(1+(2*lambda/(2*{(C1*rd^C2)/(C3*rd^C4-
log10(X[,1]))+rd^3}^(1/3)))*(1.257+0.4*exp(-0.55*((2*{(C1*rd^C2)/(C3*rd^C4-
log10(X[,1]))+rd^3}^(1/3)))/lambda))))/(18*dyn.vis))*(X[,5]))/(9.81*A))/(alpha+((((( X[,2])*(2*{(C1*rd^C2)/(C3*rd^C
4-log10(X[,1]))+rd^3}^(1/3))^2*9.81*(1+(2*lambda/(2*{(C1*rd^C2)/(C3*rd^C4-
log10(X[,1]))+rd^3}^(1/3)))*(1.257+0.4*exp(-0.55*((2*{(C1*rd^C2)/(C3*rd^C4-
log10(X[,1]))+rd^3}^(1/3)))/lambda))))/(18*dyn.vis))*(X[,5]))/(9.81*A)))}^beta)+
                   (0.5*((2*{(C1*rd^C2)/(C3*rd^C4-log10(X[,1]))+rd^3}^(1/3))/A)^2))*R1}))))

N <- 100000
x1 = runif(1*N,0.1,1.0)              #RH
x2 = runif(1*N,1500,2000)            #rho
x3 = runif(1*N,10,100)               #L
x4 = runif(1*N,0.02,0.10)            #z0
x5 = runif(1*N,0.1,0.5)              #u_star
x_1 = runif(1*N,0.1,1.0)             #RH
x_2 = runif(1*N,1500,2000)           #rho
x_3 = runif(1*N,10,100)              #L
x_4 = runif(1*N,0.02,0.10)           #z0
x_5 = runif(1*N,0.1,0.5)             #u_star
Y1 = matrix(c(x1,x2,x3,x4,x5), nrow=N)
X1 = data.frame(matrix(Y1,nrow=N))
Y2 = matrix(c(x_1,x_2,x_3,x_4,x_5), nrow=N)
X2 = data.frame(matrix(Y2, nrow=N))
a = sobol2007(model = model, X1 = X1, X2= X2, nboot = 2000, conf = 0.95);a
```

**Codes for Sobol' sensitivity test for the Petroff and Zhang (2010) parameterization**

```r
**Dry deposition parameterization by Petroff and Zhang (2010)**
**Sobol sensitivity test: Grass (the code is similar for other LUCs)**
**Change LUC dependent parameters for other LUCs**
**Change sensitivity ranges for other LUCs**
set.seed(5)
library(sensitivity)
library(boot)
C1 = 0.2789
C2 = 3.115
C3 = 5.415*10^-11
C4 = -1.399
dp_i = 10
dp_d = dp_i*10^-6
rd = dp_d/2
k_B = 1.38*10^-23
Temp = 273.15+25
P = 101325
d_air = 3.7208*10^-10
lambda = (k_B*Temp)/(sqrt(2)*3.1416*P*d_air^2)
dyn.vis = ((5*10^-8)*Temp)+4*10^-6
kin.vis = ((9*10^-8)*Temp)+10^-5
Vphor = 0
z = 2
zR = 3.5
k_c = 0.41
cd = 1/6
kx = 0.216
C_IT = 0.056
L_obs = 0.01
```

```
C_B = 0.996
C_IN = 0.162
C_IM = 0.081
beta_IM = 0.47

model <- function (X) ((((((X[,2])*(2*({(C1*rd^C2)/(C3*rd^C4-
log10(X[,1]))+rd^3}^(1/3)))^2*(1+(2*lambda/(2*({(C1*rd^C2)/(C3*rd^C4-log10(X[,1]))+rd^3}^(1/3))))*(1.257+0.4*exp(-
0.55*(2*({(C1*rd^C2)/(C3*rd^C4-
log10(X[,1]))+rd^3}^(1/3)))/lambda))))/(18*dyn.vis))*9.81)+Vphor)+1/(((log(zR/(X[,4]))+(5*(z/(X[,3]))))/(k_c*(X[,5
])))+1/((X[,5])*(((((kin.vis/(((1+(2*lambda/(2*({(C1*rd^C2)/(C3*rd^C4-
log10(X[,1]))+rd^3}^(1/3))))*(1.257+0.4*exp(-0.55*(2*({(C1*rd^C2)/(C3*rd^C4-
log10(X[,1]))+rd^3}^(1/3)))/lambda)))*k_B*Temp)/(3*3.1416*dyn.vis*(2*({(C1*rd^C2)/(C3*rd^C4-
log10(X[,1]))+rd^3}^(1/3))))))^(-2/3)/14.5)*{1/6*log(1+((((kin.vis/(((1+(2*lambda/(2*({(C1*rd^C2)/(C3*rd^C4-
log10(X[,1]))+rd^3}^(1/3))))*(1.257+0.4*exp(-0.55*(2*({(C1*rd^C2)/(C3*rd^C4-
log10(X[,1]))+rd^3}^(1/3)))/lambda)))*k_B*Temp)/(3*3.1416*dyn.vis*(2*({(C1*rd^C2)/(C3*rd^C4-
log10(X[,1]))+rd^3}^(1/3)))))))^(1/3))/2.9))^2/(1-((((kin.vis/(((1+(2*lambda/(2*({(C1*rd^C2)/(C3*rd^C4-
log10(X[,1]))+rd^3}^(1/3))))*(1.257+0.4*exp(-0.55*(2*({(C1*rd^C2)/(C3*rd^C4-
log10(X[,1]))+rd^3}^(1/3)))/lambda)))*k_B*Temp)/(3*3.1416*dyn.vis*(2*({(C1*rd^C2)/(C3*rd^C4-
log10(X[,1]))+rd^3}^(1/3)))))))^(1/3))/2.9)+((((kin.vis/(((1+(2*lambda/(2*({(C1*rd^C2)/(C3*rd^C4-
log10(X[,1]))+rd^3}^(1/3))))*(1.257+0.4*exp(-0.55*(2*({(C1*rd^C2)/(C3*rd^C4-
log10(X[,1]))+rd^3}^(1/3)))/lambda)))*k_B*Temp)/(3*3.1416*dyn.vis*(2*({(C1*rd^C2)/(C3*rd^C4-
log10(X[,1]))+rd^3}^(1/3)))))))^(1/3))/2.9)^2)+1/sqrt(3)*atan((2*((((kin.vis/(((1+(2*lambda/(2*({(C1*rd^C2)/(C3*rd
^C4-log10(X[,1]))+rd^3}^(1/3))))*(1.257+0.4*exp(-0.55*(2*({(C1*rd^C2)/(C3*rd^C4-
log10(X[,1]))+rd^3}^(1/3)))/lambda)))*k_B*Temp)/(3*3.1416*dyn.vis*(2*({(C1*rd^C2)/(C3*rd^C4-
log10(X[,1]))+rd^3}^(1/3)))))))^(1/3))/2.9)-1)/sqrt(3))+3.1416/6*sqrt(3)}^-1) + (2.5*10^-
3*0.14*((((((X[,2])*(2*({(C1*rd^C2)/(C3*rd^C4-
log10(X[,1]))+rd^3}^(1/3)))^2*(1+(2*lambda/(2*({(C1*rd^C2)/(C3*rd^C4-log10(X[,1]))+rd^3}^(1/3))))*(1.257+0.4*exp(-
0.55*(2*({(C1*rd^C2)/(C3*rd^C4-log10(X[,1]))+rd^3}^(1/3)))/lambda))))/(18*dyn.vis))*(X[,5])^2)/kin.vis))^2)
)*{(1+((X[,6])*(((((X[,9])/(exp(({(kx*(X[,6]))/(12*k_c^2*(1-
(X[,8])/(X[,7]))^2)}^(1/3)*(1+5*(z/(X[,3])))^(2/3)*{((X[,7])-(X[,8]))/((X[,3]))})*((z)/(X[,7])-
1))))/(X[,5]))*(C_B*((kin.vis/(((1+(2*lambda/(2*({(C1*rd^C2)/(C3*rd^C4-
log10(X[,1]))+rd^3}^(1/3))))*(1.257+0.4*exp(-0.55*(2*({(C1*rd^C2)/(C3*rd^C4-
log10(X[,1]))+rd^3}^(1/3)))/lambda)))*k_B*Temp)/(3*3.1416*dyn.vis*(2*({(C1*rd^C2)/(C3*rd^C4-
log10(X[,1]))+rd^3}^(1/3))))))^(-2/3))*(((((X[,9])/(exp(({(kx*(X[,6]))/(12*k_c^2*(1-
(X[,8])/(X[,7]))^2)}^(1/3)*(1+5*(z/(X[,3])))^(2/3)*{((X[,7])-(X[,8]))/((X[,3]))})*((z)/(X[,7])-
1))))*L_obs)/(kin.vis))^(-1/2)))+(C_IN*((2*({(C1*rd^C2)/(C3*rd^C4-
log10(X[,1]))+rd^3}^(1/3))/L_obs)*(2+log(4*L_obs/(2*({(C1*rd^C2)/(C3*rd^C4-
log10(X[,1]))+rd^3}^(1/3))))))+(C_IM*((((((X[,2])*(2*({(C1*rd^C2)/(C3*rd^C4-
log10(X[,1]))+rd^3}^(1/3)))^2*(1+(2*lambda/(2*({(C1*rd^C2)/(C3*rd^C4-log10(X[,1]))+rd^3}^(1/3))))*(1.257+0.4*exp(-
0.55*(2*({(C1*rd^C2)/(C3*rd^C4-log10(X[,1]))+rd^3}^(1/3)))/lambda))))/(18*dyn.vis)
)*((X[,9])/(exp(({(kx*(X[,6]))/(12*k_c^2*(1-(X[,8])/(X[,7]))^2)}^(1/3)*(1+5*(z/(X[,3])))^(2/3)*{((X[,7])-
(X[,8]))/((X[,3]))})*((z)/(X[,7])-1)))))/L_obs)/((((((X[,2])*(2*({(C1*rd^C2)/(C3*rd^C4-
log10(X[,1]))+rd^3}^(1/3)))^2*(1+(2*lambda/(2*({(C1*rd^C2)/(C3*rd^C4-log10(X[,1]))+rd^3}^(1/3))))*(1.257+0.4*exp(-
0.55*(2*({(C1*rd^C2)/(C3*rd^C4-log10(X[,1]))+rd^3}^(1/3)))/lambda))))/(18*dyn.vis)
)*((X[,9])/(exp(({(kx*(X[,6]))/(12*k_c^2*(1-(X[,8])/(X[,7]))^2)}^(1/3)*(1+5*(z/(X[,3])))^(2/3)*{((X[,7])-
(X[,8]))/((X[,3]))})*((z)/(X[,7])-1)))))/L_obs)+(beta_IM)^2))+(2.5*10^-
3*C_IT*((((X[,2])*(2*({(C1*rd^C2)/(C3*rd^C4-log10(X[,1]))+rd^3}^(1/3)))^2*(1+(2*lambda/(2*({(C1*rd^C2)/(C3*rd^C4-
log10(X[,1]))+rd^3}^(1/3))))*(1.257+0.4*exp(-0.55*(2*({(C1*rd^C2)/(C3*rd^C4-
log10(X[,1]))+rd^3}^(1/3)))/lambda))))/(18*dyn.vis))*(X[,5])^2)/kin.vis)^2)
)*(X[,7])/(((0.41*((X[,7])-(X[,8])))/((1+5*(z/(X[,3])))*((X[,7])-
(X[,8]))/((X[,3])))))/((((((kin.vis/(((1+(2*lambda/(2*({(C1*rd^C2)/(C3*rd^C4-
log10(X[,1]))+rd^3}^(1/3))))*(1.257+0.4*exp(-0.55*(2*({(C1*rd^C2)/(C3*rd^C4-
log10(X[,1]))+rd^3}^(1/3)))/lambda)))*k_B*Temp)/(3*3.1416*dyn.vis*(2*({(C1*rd^C2)/(C3*rd^C4-
log10(X[,1]))+rd^3}^(1/3))))))^(-2/3)/14.5)*{1/6*log(1+((((kin.vis/(((1+(2*lambda/(2*({(C1*rd^C2)/(C3*rd^C4-
log10(X[,1]))+rd^3}^(1/3))))*(1.257+0.4*exp(-0.55*(2*({(C1*rd^C2)/(C3*rd^C4-
log10(X[,1]))+rd^3}^(1/3)))/lambda)))*k_B*Temp)/(3*3.1416*dyn.vis*(2*({(C1*rd^C2)/(C3*rd^C4-
log10(X[,1]))+rd^3}^(1/3)))))))^(1/3))/2.9))^2/(1-((((kin.vis/(((1+(2*lambda/(2*({(C1*rd^C2)/(C3*rd^C4-
log10(X[,1]))+rd^3}^(1/3))))*(1.257+0.4*exp(-0.55*(2*({(C1*rd^C2)/(C3*rd^C4-
log10(X[,1]))+rd^3}^(1/3)))/lambda)))*k_B*Temp)/(3*3.1416*dyn.vis*(2*({(C1*rd^C2)/(C3*rd^C4-
log10(X[,1]))+rd^3}^(1/3)))))))^(1/3))/2.9)+((((kin.vis/(((1+(2*lambda/(2*({(C1*rd^C2)/(C3*rd^C4-
log10(X[,1]))+rd^3}^(1/3))))*(1.257+0.4*exp(-0.55*(2*({(C1*rd^C2)/(C3*rd^C4-
log10(X[,1]))+rd^3}^(1/3)))/lambda)))*k_B*Temp)/(3*3.1416*dyn.vis*(2*({(C1*rd^C2)/(C3*rd^C4-
log10(X[,1]))+rd^3}^(1/3)))))))^(1/3))/2.9)^2)+1/sqrt(3)*atan((2*((((kin.vis/(((1+(2*lambda/(2*({(C1*rd^C2)/(C3*rd
^C4-log10(X[,1]))+rd^3}^(1/3))))*(1.257+0.4*exp(-0.55*(2*({(C1*rd^C2)/(C3*rd^C4-
log10(X[,1]))+rd^3}^(1/3)))/lambda)))*k_B*Temp)/(3*3.1416*dyn.vis*(2*({(C1*rd^C2)/(C3*rd^C4-
log10(X[,1]))+rd^3}^(1/3)))))))^(1/3))/2.9)-1)/sqrt(3))+3.1416/6*sqrt(3)}^-1) + (2.5*10^-
3*0.14*((((((X[,2])*(2*({(C1*rd^C2)/(C3*rd^C4-
log10(X[,1]))+rd^3}^(1/3)))^2*(1+(2*lambda/(2*({(C1*rd^C2)/(C3*rd^C4-log10(X[,1]))+rd^3}^(1/3))))*(1.257+0.4*exp(-
0.55*(2*({(C1*rd^C2)/(C3*rd^C4-
log10(X[,1]))+rd^3}^(1/3)))/lambda))))/(18*dyn.vis))*(X[,5])^2)/kin.vis))^2))*(X[,7])/((0.41*((X[,7])-
(X[,8])))/((1+5*(z/(X[,3])))*((X[,7])-(X[,8]))/((X[,3]))))
)-({kx*(X[,6]))/(12*k_c^2*(1-(X[,8])/(X[,7]))^2)}^(1/3)*(1+5*(z/(X[,3])))^(2/3)*{((X[,7])-
(X[,8]))/((X[,3]))})/2)*tanh(((({(kx*(X[,6]))/(12*k_c^2*(1-
```

```
(X[,8])/(X[,7]))^2)}^(1/3)*(1+5*(z/(X[,3])))^(2/3)*{((X[,7])-
(X[,8]))/((X[,3]))})^2/4+((X[,6])*((((X[,9])/(exp(({(kx*(X[,6]))/(12*k_c^2*(1-
(X[,8])/(X[,7]))^2)}^(1/3)*(1+5*(z/(X[,3])))^(2/3)*{((X[,7])-(X[,8]))/((X[,3]))})*((z)/(X[,7])-
1))))/(X[,5]))*((C_B*(((kin.vis/(((1+(2*lambda/(2*({(C1*rd^C2)/(C3*rd^C4-
log10(X[,1]))+rd^3}^(1/3))))*(1.257+0.4*exp(-0.55*(2*({(C1*rd^C2)/(C3*rd^C4-
log10(X[,1]))+rd^3}^(1/3)))/lambda))*k_B*Temp)/(3*3.1416*dyn.vis*(2*({(C1*rd^C2)/(C3*rd^C4-
log10(X[,1]))+rd^3}^(1/3)))))))^(-2/3))*(((((X[,9])/(exp(({(kx*(X[,6]))/(12*k_c^2*(1-
(X[,8])/(X[,7]))^2)}^(1/3)*(1+5*(z/(X[,3])))^(2/3)*{((X[,7])-(X[,8]))/((X[,3]))})*((z)/(X[,7])-
1))))*L_obs)/(kin.vis))^(-1/2)))+(C_IN*((2*({(C1*rd^C2)/(C3*rd^C4-
log10(X[,1]))+rd^3}^(1/3)))/L_obs)*(2+log(4*L_obs/(2*({(C1*rd^C2)/(C3*rd^C4-
log10(X[,1]))+rd^3}^(1/3))))))+(C_IM*((((((X[,2])*(2*({(C1*rd^C2)/(C3*rd^C4-
log10(X[,1]))+rd^3}^(1/3)))^2*(1+(2*lambda/(2*({(C1*rd^C2)/(C3*rd^C4-log10(X[,1]))+rd^3}^(1/3))))*(1.257+0.4*exp(-
0.55*(2*({(C1*rd^C2)/(C3*rd^C4-log10(X[,1]))+rd^3}^(1/3)))/lambda))))/(18*dyn.vis)
)*((X[,9])/(exp(({(kx*(X[,6]))/(12*k_c^2*(1-(X[,8])/(X[,7]))^2)}^(1/3)*(1+5*(z/(X[,3])))^(2/3)*{((X[,7])-
(X[,8]))/((X[,3]))})*((z)/(X[,7])-1)))))/L_obs)/(((((((X[,2])*(2*({(C1*rd^C2)/(C3*rd^C4-
log10(X[,1]))+rd^3}^(1/3)))^2*(1+(2*lambda/(2*({(C1*rd^C2)/(C3*rd^C4-log10(X[,1]))+rd^3}^(1/3))))*(1.257+0.4*exp(-
0.55*(2*({(C1*rd^C2)/(C3*rd^C4-log10(X[,1]))+rd^3}^(1/3)))/lambda))))/(18*dyn.vis)
)*((X[,9])/(exp(({(kx*(X[,6]))/(12*k_c^2*(1-(X[,8])/(X[,7]))^2)}^(1/3)*(1+5*(z/(X[,3])))^(2/3)*{((X[,7])-
(X[,8]))/((X[,3]))})*((z)/(X[,7])-1))))))/L_obs)+(beta_IM))^2))+(2.5*10^-
3*C_IT*((((X[,2])*(2*({(C1*rd^C2)/(C3*rd^C4-log10(X[,1]))+rd^3}^(1/3)))^2*(1+(2*lambda/(2*({(C1*rd^C2)/(C3*rd^C4-
log10(X[,1]))+rd^3}^(1/3))))*(1.257+0.4*exp(-0.55*(2*({(C1*rd^C2)/(C3*rd^C4-
log10(X[,1]))+rd^3}^(1/3)))/lambda))))/(18*dyn.vis))*(X[,5])^2)/kin.vis)^2)
)*(X[,7])/(((0.41*((X[,7])-(X[,8])))/((1+5*(z/(X[,3])))*((X[,7])-
(X[,8]))/((X[,3])))))))^0.5)/((((({(kx*(X[,6]))/(12*k_c^2*(1-
(X[,8])/(X[,7]))^2)}^(1/3)*(1+5*(z/(X[,3])))^(2/3)*{((X[,7])-
(X[,8]))/((X[,3]))})^2/4+((X[,6])*((((X[,9])/(exp(({(kx*(X[,6]))/(12*k_c^2*(1-
(X[,8])/(X[,7]))^2)}^(1/3)*(1+5*(z/(X[,3])))^(2/3)*{((X[,7])-(X[,8]))/((X[,3]))})*((z)/(X[,7])-
1))))/(X[,5]))*((C_B*(((kin.vis/(((1+(2*lambda/(2*({(C1*rd^C2)/(C3*rd^C4-
log10(X[,1]))+rd^3}^(1/3))))*(1.257+0.4*exp(-0.55*(2*({(C1*rd^C2)/(C3*rd^C4-
log10(X[,1]))+rd^3}^(1/3)))/lambda)))*k_B*Temp)/(3*3.1416*dyn.vis*(2*({(C1*rd^C2)/(C3*rd^C4-
log10(X[,1]))+rd^3}^(1/3)))))))^(-2/3))*(((((X[,9])/(exp(({(kx*(X[,6]))/(12*k_c^2*(1-
(X[,8])/(X[,7]))^2)}^(1/3)*(1+5*(z/(X[,3])))^(2/3)*{((X[,7])-(X[,8]))/((X[,3]))})*((z)/(X[,7])-
1))))*L_obs)/(kin.vis))^(-1/2)))+(C_IN*((2*({(C1*rd^C2)/(C3*rd^C4-
log10(X[,1]))+rd^3}^(1/3)))/L_obs)*(2+log(4*L_obs/(2*({(C1*rd^C2)/(C3*rd^C4-
log10(X[,1]))+rd^3}^(1/3))))))+(C_IM*((((((X[,2])*(2*({(C1*rd^C2)/(C3*rd^C4-
log10(X[,1]))+rd^3}^(1/3)))^2*(1+(2*lambda/(2*({(C1*rd^C2)/(C3*rd^C4-log10(X[,1]))+rd^3}^(1/3))))*(1.257+0.4*exp(-
0.55*(2*({(C1*rd^C2)/(C3*rd^C4-log10(X[,1]))+rd^3}^(1/3)))/lambda))))/(18*dyn.vis)
)*((X[,9])/(exp(({(kx*(X[,6]))/(12*k_c^2*(1-(X[,8])/(X[,7]))^2)}^(1/3)*(1+5*(z/(X[,3])))^(2/3)*{((X[,7])-
(X[,8]))/((X[,3]))})*((z)/(X[,7])-1)))))/L_obs)/((((((X[,2])*(2*({(C1*rd^C2)/(C3*rd^C4-
log10(X[,1]))+rd^3}^(1/3)))^2*(1+(2*lambda/(2*({(C1*rd^C2)/(C3*rd^C4-log10(X[,1]))+rd^3}^(1/3))))*(1.257+0.4*exp(-
0.55*(2*({(C1*rd^C2)/(C3*rd^C4-log10(X[,1]))+rd^3}^(1/3)))/lambda))))/(18*dyn.vis)
)*((X[,9])/(exp(({(kx*(X[,6]))/(12*k_c^2*(1-(X[,8])/(X[,7]))^2)}^(1/3)*(1+5*(z/(X[,3])))^(2/3)*{((X[,7])-
(X[,8]))/((X[,3]))})*((z)/(X[,7])-1))))))/L_obs)+(beta_IM)))^2))+(2.5*10^-
3*C_IT*((((X[,2])*(2*({(C1*rd^C2)/(C3*rd^C4-log10(X[,1]))+rd^3}^(1/3)))^2*(1+(2*lambda/(2*({(C1*rd^C2)/(C3*rd^C4-
log10(X[,1]))+rd^3}^(1/3))))*(1.257+0.4*exp(-0.55*(2*({(C1*rd^C2)/(C3*rd^C4-
log10(X[,1]))+rd^3}^(1/3)))/lambda))))/(18*dyn.vis))*(X[,5])^2)/kin.vis)^2)
)*(X[,7])/(((0.41*((X[,7])-(X[,8])))/((1+5*(z/(X[,3])))*((X[,7])-
(X[,8]))/((X[,3]))))))))^0.5)/{(1+((X[,6])*((((X[,9])/(exp(({(kx*(X[,6]))/(12*k_c^2*(1-
(X[,8])/(X[,7]))^2)}^(1/3)*(1+5*(z/(X[,3])))^(2/3)*{((X[,7])-(X[,8]))/((X[,3]))})*((z)/(X[,7])-
1))))/(X[,5]))*((C_B*(((kin.vis/(((1+(2*lambda/(2*({(C1*rd^C2)/(C3*rd^C4-
log10(X[,1]))+rd^3}^(1/3))))*(1.257+0.4*exp(-0.55*(2*({(C1*rd^C2)/(C3*rd^C4-
log10(X[,1]))+rd^3}^(1/3)))/lambda)))*k_B*Temp)/(3*3.1416*dyn.vis*(2*({(C1*rd^C2)/(C3*rd^C4-
log10(X[,1]))+rd^3}^(1/3)))))))^(-2/3))*(((((X[,9])/(exp(({(kx*(X[,6]))/(12*k_c^2*(1-
(X[,8])/(X[,7]))^2)}^(1/3)*(1+5*(z/(X[,3])))^(2/3)*{((X[,7])-(X[,8]))/((X[,3]))})*((z)/(X[,7])-
1))))*L_obs)/(kin.vis))^(-1/2)))+(C_IN*((2*({(C1*rd^C2)/(C3*rd^C4-
log10(X[,1]))+rd^3}^(1/3)))/L_obs)*(2+log(4*L_obs/(2*({(C1*rd^C2)/(C3*rd^C4-
log10(X[,1]))+rd^3}^(1/3))))))+(C_IM*((((((X[,2])*(2*({(C1*rd^C2)/(C3*rd^C4-
log10(X[,1]))+rd^3}^(1/3)))^2*(1+(2*lambda/(2*({(C1*rd^C2)/(C3*rd^C4-log10(X[,1]))+rd^3}^(1/3))))*(1.257+0.4*exp(-
0.55*(2*({(C1*rd^C2)/(C3*rd^C4-log10(X[,1]))+rd^3}^(1/3)))/lambda))))/(18*dyn.vis)
)*((X[,9])/(exp(({(kx*(X[,6]))/(12*k_c^2*(1-(X[,8])/(X[,7]))^2)}^(1/3)*(1+5*(z/(X[,3])))^(2/3)*{((X[,7])-
(X[,8]))/((X[,3]))})*((z)/(X[,7])-1))))))/L_obs)/(((((((X[,2])*(2*({(C1*rd^C2)/(C3*rd^C4-
log10(X[,1]))+rd^3}^(1/3)))^2*(1+(2*lambda/(2*({(C1*rd^C2)/(C3*rd^C4-log10(X[,1]))+rd^3}^(1/3))))*(1.257+0.4*exp(-
0.55*(2*({(C1*rd^C2)/(C3*rd^C4-log10(X[,1]))+rd^3}^(1/3)))/lambda))))/(18*dyn.vis)
)*((X[,9])/(exp(({(kx*(X[,6]))/(12*k_c^2*(1-(X[,8])/(X[,7]))^2)}^(1/3)*(1+5*(z/(X[,3])))^(2/3)*{((X[,7])-
(X[,8]))/((X[,3]))})*((z)/(X[,7])-1))))))/L_obs)+(beta_IM)))^2))+(2.5*10^-
3*C_IT*((((X[,2])*(2*({(C1*rd^C2)/(C3*rd^C4-log10(X[,1]))+rd^3}^(1/3)))^2*(1+(2*lambda/(2*({(C1*rd^C2)/(C3*rd^C4-
log10(X[,1]))+rd^3}^(1/3))))*(1.257+0.4*exp(-0.55*(2*({(C1*rd^C2)/(C3*rd^C4-
log10(X[,1]))+rd^3}^(1/3)))/lambda))))/(18*dyn.vis))*(X[,5])^2)/kin.vis)^2)
)*(X[,7])/(((0.41*((X[,7])-(X[,8])))/((1+5*(z/(X[,3])))*((X[,7])-
(X[,8]))/((X[,3]))))))))+({(kx*(X[,6]))/(12*k_c^2*(1-(X[,8])/(X[,7]))^2)}^(1/3)*(1+5*(z/(X[,3])))^(2/3)*{((X[,7])-
(X[,8]))/((X[,3]))}/2)*tanh(((({(kx*(X[,6]))/(12*k_c^2*(1-
(X[,8])/(X[,7]))^2)}^(1/3)*(1+5*(z/(X[,3])))^(2/3)*{((X[,7])-
(X[,8]))/((X[,3]))})^2/4+((X[,6])*((((X[,9])/(exp(({(kx*(X[,6]))/(12*k_c^2*(1-
(X[,8])/(X[,7]))^2)}^(1/3)*(1+5*(z/(X[,3])))^(2/3)*{((X[,7])-(X[,8]))/((X[,3]))})*((z)/(X[,7])-
```

```
1))))/(X[,5]))*((C_B*(((kin.vis/(((1+(2*lambda/(2*({(C1*rd^C2)/(C3*rd^C4-
log10(X[,1]))+rd^3}^(1/3))))*(1.257+0.4*exp(-0.55*(2*({(C1*rd^C2)/(C3*rd^C4-
log10(X[,1]))+rd^3}^(1/3)))/lambda)))*k_B*Temp)/(3*3.1416*dyn.vis*(2*({(C1*rd^C2)/(C3*rd^C4-
log10(X[,1]))+rd^3}^(1/3))))))))^(-2/3))*(((((X[,9])/(exp(({(kx*(X[,6)))/(12*k_c^2*(1-
(X[,8])/(X[,7]))^2)}^(1/3)*(1+5*(z/(X[,3])))^(2/3)*{((X[,7])-(X[,8]))/((X[,3]))})*((z)/(X[,7])-
1))))*L_obs)/(kin.vis))^(-1/2)))+(C_IN*((2*({(C1*rd^C2)/(C3*rd^C4-
log10(X[,1]))+rd^3}^(1/3)))/L_obs)*(2+log(4*L_obs/(2*({(C1*rd^C2)/(C3*rd^C4-
log10(X[,1]))+rd^3}^(1/3)))))+(C_IM*((((((X[,2])*(2*({(C1*rd^C2)/(C3*rd^C4-
log10(X[,1]))+rd^3}^(1/3))^2*(1+(2*lambda/(2*({(C1*rd^C2)/(C3*rd^C4-log10(X[,1]))+rd^3}^(1/3))))*(1.257+0.4*exp(-
0.55*(2*({(C1*rd^C2)/(C3*rd^C4-log10(X[,1]))+rd^3}^(1/3)))/lambda))))/(18*dyn.vis)
)*((X[,9])/(exp(({(kx*(X[,6)))/(12*k_c^2*(1-(X[,8])/(X[,7]))^2)}^(1/3)*(1+5*(z/(X[,3])))^(2/3)*{((X[,7])-
(X[,8]))/((X[,3]))})*((z)/(X[,7])-1)))))/L_obs)/((((((X[,2])*(2*({(C1*rd^C2)/(C3*rd^C4-
log10(X[,1]))+rd^3}^(1/3))^2*(1+(2*lambda/(2*({(C1*rd^C2)/(C3*rd^C4-log10(X[,1]))+rd^3}^(1/3))))*(1.257+0.4*exp(-
0.55*(2*({(C1*rd^C2)/(C3*rd^C4-log10(X[,1]))+rd^3}^(1/3)))/lambda))))/(18*dyn.vis)
)*((X[,9])/(exp(({(kx*(X[,6)))/(12*k_c^2*(1-(X[,8])/(X[,7]))^2)}^(1/3)*(1+5*(z/(X[,3])))^(2/3)*{((X[,7])-
(X[,8]))/((X[,3]))})*((z)/(X[,7])-1))))))/L_obs)+(beta_IM)))^2))+(2.5*10^-
3*C_IT*((((X[,2])*(2*({(C1*rd^C2)/(C3*rd^C4-log10(X[,1]))+rd^3}^(1/3))^2*(1+(2*lambda/(2*({(C1*rd^C2)/(C3*rd^C4-
log10(X[,1]))+rd^3}^(1/3))))*(1.257+0.4*exp(-0.55*(2*({(C1*rd^C2)/(C3*rd^C4-
log10(X[,1]))+rd^3}^(1/3)))/lambda))))/(18*dyn.vis))*(X[,5])^2)/kin.vis)^2)
)*(X[,7])/((((0.41*((X[,7])-(X[,8])))/((1+5*(z/(X[,3])))*((X[,7])-
(X[,8]))/((X[,3]))))))^0.5))/(((({(kx*(X[,6)))/(12*k_c^2*(1-
(X[,8])/(X[,7]))^2)}^(1/3)*(1+5*(z/(X[,3])))^(2/3)*{((X[,7])-
(X[,8]))/((X[,3]))})^2/4+((X[,6])*(((((X[,9])/(exp(({(kx*(X[,6)))/(12*k_c^2*(1-
(X[,8])/(X[,7]))^2)}^(1/3)*(1+5*(z/(X[,3])))^(2/3)*{((X[,7])-(X[,8]))/((X[,3]))})*((z)/(X[,7])-
1))))/(X[,5]))*((C_B*(((kin.vis/(((1+(2*lambda/(2*({(C1*rd^C2)/(C3*rd^C4-
log10(X[,1]))+rd^3}^(1/3))))*(1.257+0.4*exp(-0.55*(2*({(C1*rd^C2)/(C3*rd^C4-
log10(X[,1]))+rd^3}^(1/3)))/lambda)))*k_B*Temp)/(3*3.1416*dyn.vis*(2*({(C1*rd^C2)/(C3*rd^C4-
log10(X[,1]))+rd^3}^(1/3))))))))^(-2/3))*(((((X[,9])/(exp(({(kx*(X[,6)))/(12*k_c^2*(1-
(X[,8])/(X[,7]))^2)}^(1/3)*(1+5*(z/(X[,3])))^(2/3)*{((X[,7])-(X[,8]))/((X[,3]))})*((z)/(X[,7])-
1))))*L_obs)/(kin.vis))^(-1/2)))+(C_IN*((2*({(C1*rd^C2)/(C3*rd^C4-
log10(X[,1]))+rd^3}^(1/3)))/L_obs)*(2+log(4*L_obs/(2*({(C1*rd^C2)/(C3*rd^C4-
log10(X[,1]))+rd^3}^(1/3)))))+(C_IM*((((((X[,2])*(2*({(C1*rd^C2)/(C3*rd^C4-
log10(X[,1]))+rd^3}^(1/3))^2*(1+(2*lambda/(2*({(C1*rd^C2)/(C3*rd^C4-log10(X[,1]))+rd^3}^(1/3))))*(1.257+0.4*exp(-
0.55*(2*({(C1*rd^C2)/(C3*rd^C4-log10(X[,1]))+rd^3}^(1/3)))/lambda))))/(18*dyn.vis)
)*((X[,9])/(exp(({(kx*(X[,6)))/(12*k_c^2*(1-(X[,8])/(X[,7]))^2)}^(1/3)*(1+5*(z/(X[,3])))^(2/3)*{((X[,7])-
(X[,8]))/((X[,3]))})*((z)/(X[,7])-1)))))/L_obs)/((((((X[,2])*(2*({(C1*rd^C2)/(C3*rd^C4-
log10(X[,1]))+rd^3}^(1/3))^2*(1+(2*lambda/(2*({(C1*rd^C2)/(C3*rd^C4-log10(X[,1]))+rd^3}^(1/3))))*(1.257+0.4*exp(-
0.55*(2*({(C1*rd^C2)/(C3*rd^C4-log10(X[,1]))+rd^3}^(1/3)))/lambda))))/(18*dyn.vis)
)*((X[,9])/(exp(({(kx*(X[,6)))/(12*k_c^2*(1-(X[,8])/(X[,7]))^2)}^(1/3)*(1+5*(z/(X[,3])))^(2/3)*{((X[,7])-
(X[,8]))/((X[,3]))})*((z)/(X[,7])-1))))))/L_obs)+(beta_IM)))^2))+(2.5*10^-
3*C_IT*((((X[,2])*(2*({(C1*rd^C2)/(C3*rd^C4-log10(X[,1]))+rd^3}^(1/3))^2*(1+(2*lambda/(2*({(C1*rd^C2)/(C3*rd^C4-
log10(X[,1]))+rd^3}^(1/3))))*(1.257+0.4*exp(-0.55*(2*({(C1*rd^C2)/(C3*rd^C4-
log10(X[,1]))+rd^3}^(1/3)))/lambda))))/(18*dyn.vis))*(X[,5])^2)/kin.vis)^2)
)*(X[,7])/((((0.41*((X[,7])-(X[,8])))/((1+5*(z/(X[,3])))*((X[,7])-(X[,8]))/((X[,3]))))))^0.5)})))

N <- 100000
x1 = runif(1*N,0.1,1.0)              #RH
x2 = runif(1*N,1500,2000)            #rho
x3 = runif(1*N,10,100)               #L
x4 = runif(1*N,0.02,0.10)            #z0
x5 = runif(1*N,0.1,0.5)              #u_star
x6 = runif(1*N,1,4)                  #LAI
x7 = runif(1*N,0.15,0.77)            #h
x8 = runif(1*N,0.10,0.49)            #d
x9 = runif(1*N,1,5)                  #U

x_1 = runif(1*N,0.1,1.0)              #RH
x_2 = runif(1*N,1500,2000)            #rho
x_3 = runif(1*N,10,100)               #L
x_4 = runif(1*N,0.02,0.10)            #z0
x_5 = runif(1*N,0.1,0.5)              #u_star
x_6 = runif(1*N,1,4)                  #LAI
x_7 = runif(1*N,0.15,0.77)            #h
x_8 = runif(1*N,0.10,0.49)            #d
x_9 = runif(1*N,1,5)                  #U

Y1 = matrix(c(x1,x2,x3,x4,x5,x6,x7,x8,x9), nrow=N)
X1 = data.frame(matrix(Y1, nrow=N))
Y2 = matrix(c(x_1,x_2,x_3,x_4,x_5,x_6,x_7,x_8,x_9), nrow=N)
X2 = data.frame(matrix(Y2, nrow=N))
a = sobol2007(model = model, X1 = X1, X2=X2, nboot = 2000, conf= 0.95);a
```

**Codes for Sobol' sensitivity test for Kouznetsov and Sofiev (2012) parameterization**

```r
**Dry deposition parameterization by Kouznetsov and Sofiev (2012)**
**Sobol sensitivity test: Grass (the code is similar for other LUCs)**
**Change LUC dependent parameters for other LUCs**
**Change sensitivity ranges for other LUCs**
set.seed(5)
library(sensitivity)
library(boot)
C1 = 0.2789
C2 = 3.115
C3 = 5.415*10^-11
C4 = -1.399
dp_a = 10
dp_i = dp_a*10^-6
rd = dp_i/2
k_B = 1.38*10^-23
Temp = 273.15+25
P = 101325
d_air = 3.7208*10^-10
lambda = (k_B*Temp)/(sqrt(2)*3.1416*P*d_air^2)
dyn.vis = ((5*10^-8)*Temp)+4*10^-6
kin.vis = ((9*10^-8)*Temp)+10^-5
a   = 2*10^-3
kin.vis = ((9*10^-8)*Temp)+10^-5
C_S = 0.003
C_R = 0.3
u.Uh = 0.3
eta_impSt.e2 = 0

model<-function(X) ((2*(((((X[,3])*a)/kin.vis)^(-0.5))*(((kin.vis/(((1+(2*lambda/(2*({(C1*rd^C2)/(C3*rd^C4-
log10(X[,1]))+rd^3}^(1/3)))))*(1.257+0.4*exp(-0.55*(2*({(C1*rd^C2)/(C3*rd^C4-
log10(X[,1]))+rd^3}^(1/3)))/lambda)))*k_B*Temp)/(3*3.1416*dyn.vis*(2*({(C1*rd^C2)/(C3*rd^C4-
log10(X[,1]))+rd^3}^(1/3)))))))^(-2/3))+(80*(X[,3])*(((2*({(C1*rd^C2)/(C3*rd^C4-
log10(X[,1]))+rd^3}^(1/3)))/(a))^2)*(((((X[,3])*a)/kin.vis)^(0.5)))+(((2*((C_S+C_R/(X[,4]))^0.5))/(X[,3]))*(ifelse(
((((((X[,2])*(2*({(C1*rd^C2)/(C3*rd^C4-log10(X[,1]))+rd^3}^(1/3)))^2*(1+(2*lambda/(2*({(C1*rd^C2)/(C3*rd^C4-
log10(X[,1]))+rd^3}^(1/3))))*(1.257+0.4*exp(-0.55*(2*({(C1*rd^C2)/(C3*rd^C4-
log10(X[,1]))+rd^3}^(1/3)))/lambda)))/(18*dyn.vis))*(X[,3]))/a) - (((((C_S+C_R/(X[,4]))^0.5))^-
1)^2*(((X[,3])*a)/kin.vis))^(-0.5))>0.15,(exp((-0.1/((((((X[,2])*(2*({(C1*rd^C2)/(C3*rd^C4-
log10(X[,1]))+rd^3}^(1/3)))^2*(1+(2*lambda/(2*({(C1*rd^C2)/(C3*rd^C4-log10(X[,1]))+rd^3}^(1/3))))*(1.257+0.4*exp(-
0.55*(2*({(C1*rd^C2)/(C3*rd^C4-log10(X[,1]))+rd^3}^(1/3)))/lambda))))/(18*dyn.vis))*(X[,3]))/a) -
(((((C_S+C_R/(X[,4]))^0.5))^-1)^2*(((X[,3])*a)/kin.vis))^(-0.5)) - 0.15 ) )-
(1/sqrt((((((((X[,2])*(2*({(C1*rd^C2)/(C3*rd^C4-
log10(X[,1]))+rd^3}^(1/3)))^2*(1+(2*lambda/(2*({(C1*rd^C2)/(C3*rd^C4-log10(X[,1]))+rd^3}^(1/3))))*(1.257+0.4*exp(-
0.55*(2*({(C1*rd^C2)/(C3*rd^C4-log10(X[,1]))+rd^3}^(1/3)))/lambda))))/(18*dyn.vis))*(X[,3]))/a) -
(((((C_S+C_R/(X[,4]))^0.5))^-1)^2*(((X[,3])*a)/kin.vis))^(-0.5)-
0.15)))),(eta_impSt.e2)))*((((((X[,2])*(2*({(C1*rd^C2)/(C3*rd^C4-
log10(X[,1]))+rd^3}^(1/3)))^2*(1+(2*lambda/(2*({(C1*rd^C2)/(C3*rd^C4-log10(X[,1]))+rd^3}^(1/3))))*(1.257+0.4*exp(-
0.55*(2*({(C1*rd^C2)/(C3*rd^C4-log10(X[,1]))+rd^3}^(1/3)))/lambda))))/(18*dyn.vis))*(X[,3]))/a)-
((C_S+C_R/(X[,4]))^0.5)*(((X[,3])*a)/kin.vis)^-0.5))+(((((X[,2])*(2*({(C1*rd^C2)/(C3*rd^C4-
log10(X[,1]))+rd^3}^(1/3)))^2*(1+(2*lambda/(2*({(C1*rd^C2)/(C3*rd^C4-log10(X[,1]))+rd^3}^(1/3))))*(1.257+0.4*exp(-
0.55*(2*({(C1*rd^C2)/(C3*rd^C4-log10(X[,1]))+rd^3}^(1/3)))/lambda))))/(18*dyn.vis))*9.81))

N <- 100000
x1 = runif(1*N,0.1,1.0)              #RH
x2 = runif(1*N,1500,2000)            #rho
x3 = runif(1*N,0.1,0.5)              #u_star
x4 = runif(1*N,1,4)                  #LAI
x_1 = runif(1*N,0.1,1.0)             #RH
x_2 = runif(1*N,1500,2000)           #rho
x_3 = runif(1*N,0.1,0.5)             #u_star
x_4 = runif(1*N,1,4)                 #LAI
Y1 = matrix(c(x1,x2,x3,x4), nrow=N)
X1 = data.frame(matrix(Y1,nrow=N))
Y2 = matrix(c(x_1,x_2,x_3,x_4), nrow=N)
X2 = data.frame(matrix(Y2, nrow=N))
a = sobol2007(model = model, X1 = X1, X2=X2, nboot = 2000, conf = 0.95);a
```

**Codes for Sobol' sensitivity test for Zhang and He (2014) parameterization**

```
**Dry deposition parameterization by Zhang and He (2014)**
**Sobol sensitivity test: Grass (the code is similar for other LUCs)**
**Change LUC dependent parameters for other LUCs**
**Change sensitivity ranges for other LUCs**
set.seed(5)
library(sensitivity)
library(boot)
C1 = 0.4809
C2 = 3.082
C3 = 3.110*10^-11
C4 = -1.428
dp_a = 1
dp_i = dp_a*10^-6
rd = dp_i/2
k_B = 1.38*10^-23
Temp = 273.15+25
P = 101325
d_air = 3.7208*10^-10
lambda = (k_B*Temp)/(sqrt(2)*3.1416*P*d_air^2)
dyn.vis = ((5*10^-8)*Temp)+4*10^-6
z = 2
zR = 3.5
k_c = 0.41
**For PM2.5-10**
b1= -7.9*10^-2
b2= 1.0*10^0
b3 = 6.6*10^-1
c1= 5.1*10^0
c2 = -4.2*10^0
c3 = 9.9*10^-1
LAImax = 4

model<- function(X) (1/(1/((((X[,2])*((2*({(C1*rd^C2)/(C3*rd^C4-
log10(X[,1]))+rd^3}^(1/3))))^2*(1+(2*lambda/(2*({(C1*rd^C2)/(C3*rd^C4-
log10(X[,1]))+rd^3}^(1/3))))*(1.257+0.4*exp(-0.55*(2*({(C1*rd^C2)/(C3*rd^C4-
log10(X[,1]))+rd^3}^(1/3)))/lambda))))/(18*dyn.vis))*9.81))+(1/(((log(zR/(X[,5]))-(ifelse((z/(X[,4])) <= 0,
(2*log(0.5*{1+(1-16*(z/(X[,4])))^0.5})),(-
5*(z/(X[,4])))))))/(k_c*(X[,3])))+(1/((b1*(X[,3])+b2*(X[,3])^2+b3*(X[,3])^3)*(exp((c1*(X[,3])+c2*(X[,3])^2+c3*(X[,3
])^3)*(X[,6])/LAImax)-1)))))))

N <- 100000
x1 = runif(1*N,0.1,1.0)            #RH
x2 = runif(1*N,1500,2000)          #rho
x3 = runif(1*N,0.1,0.5)            #u_star
x4 = runif(1*N,10,100)             #L
x5 = runif(1*N,0.02,0.10)          #z0
x6 = runif(1*N,1,4)                #LAI
x_1 = runif(1*N,0.1,1.0)           #RH
x_2 = runif(1*N,1500,2000)         #rho
x_3 = runif(1*N,0.1,0.5)           #u_star
x_4 = runif(1*N,10,100)            #L
x_5 = runif(1*N,0.02,0.10)         #z0
x_6 = runif(1*N,1,4)               #LAI
Y1 = matrix(c(x1,x2,x3,x4,x5,x6), nrow=N)
X1 = data.frame(matrix(Y1,nrow=N))
Y2 = matrix(c(x_1,x_2,x_3,x_4,x_5,x_6), nrow=N)
X2 = data.frame(matrix(Y2, nrow=N))
a = sobol2007(model = model, X1 = X1, X2=X2, nboot = 2000, conf = 0.95);a
```

**Codes for Sobol' sensitivity test for Zhang and Sao (2014) parameterization**

```
**Dry deposition parameterization by Zhang and Sao (2014)**
**Sobol sensitivity test: Grass (the code is similar for other LUCs)**
**Change LUC dependent parameters for other LUCs**
**Change sensitivity ranges for other LUCs**
set.seed(5)
library(sensitivity)
library(boot)
```

```
C1 = 0.2789
C2 = 3.115
C3 = 5.415*10^-11
C4 = -1.399
dp_a = 10
dp_i = dp_a*10^-6
rd = dp_i/2
k_B = 1.38*10^-23
Temp = 273.15+25
P = 101325
d_air = 3.7208*10^-10
lambda = (k_B*Temp)/(sqrt(2)*3.1416*P*d_air^2)
dyn.vis = ((5*10^-8)*Temp)+4*10^-6
kin.vis = ((9*10^-8)*Temp)+10^-5
Temp = 273.15+25
k =  0.41
z =  1
zd = 0.20
z0 = 0.3/1000
h_c = 0.3*z0
B1 = 0.45
d_c = 0.005
nB = 0.5
C_B = 0.467
beta_IM = 0.6
Ain = 150
b = 2
B2 = 3
Beta = 200
C_1 = 6
C_2 = 0.1
lambda_FAI = 0.4
n_FAI = (lambda_FAI)/(h_c*d_c)
q = (3.1416*d_c^2)/4
eta_BAI = n_FAI*q
lambda_FAIe = ((lambda_FAI)/(1-eta_BAI)^C_2)*exp((-C_1*lambda_FAI)/(1-eta_BAI)^C_2)
Tau_c_BY_Tau = (Beta*lambda_FAIe)/(1+Beta*lambda_FAIe)
Cd = 1/6
model<- function(X) (((((1/(((((X[,2])*(dp_i)^2*(1+(2*lambda/(2*({(C1*rd^C2)/(C3*rd^C4-
log10(X[,1]))+rd^3}^(1/3))))*(1.257+0.4*exp(-0.55*(2*({(C1*rd^C2)/(C3*rd^C4-
log10(X[,1]))+rd^3}^(1/3)))/lambda))))/(18*dyn.vis))*9.81)))+(((((exp(-
b*sqrt(((((X[,2])*(dp_i)^2*(1+(2*lambda/(2*({(C1*rd^C2)/(C3*rd^C4-log10(X[,1]))+rd^3}^(1/3))))*(1.257+0.4*exp(-
0.55*(2*({(C1*rd^C2)/(C3*rd^C4-
log10(X[,1]))+rd^3}^(1/3)))/lambda))))/(18*dyn.vis))*(X[,3]))/d_c)))*(((X[,3])/(X[,4])*h_c))*((((((C_B*((kin.vis
/((((1+(2*lambda/(2*({(C1*rd^C2)/(C3*rd^C4-log10(X[,1]))+rd^3}^(1/3))))*(1.257+0.4*exp(-
0.55*(2*({(C1*rd^C2)/(C3*rd^C4-
log10(X[,1]))+rd^3}^(1/3)))/lambda)))*k_B*Temp)/(3*3.1416*dyn.vis*(2*({(C1*rd^C2)/(C3*rd^C4-
log10(X[,1]))+rd^3}^(1/3)))))))^(-2/3)*(((X[,4])*d_c)/(kin.vis))^(nB-
1))+(((((((X[,2])*(dp_i)^2*(1+(2*lambda/(2*({(C1*rd^C2)/(C3*rd^C4-log10(X[,1]))+rd^3}^(1/3))))*(1.257+0.4*exp(-
0.55*(2*({(C1*rd^C2)/(C3*rd^C4-
log10(X[,1]))+rd^3}^(1/3)))/lambda))))/(18*dyn.vis))*(X[,3]))/d_c)/((((((X[,2])*(dp_i)^2*(1+(2*lambda/(2*({(C1*
rd^C2)/(C3*rd^C4-log10(X[,1]))+rd^3}^(1/3))))*(1.257+0.4*exp(-0.55*(2*({(C1*rd^C2)/(C3*rd^C4-
log10(X[,1]))+rd^3}^(1/3)))/lambda))))/(18*dyn.vis))*(X[,3]))/d_c+beta_IM)^2)+(Ain*(X[,3])*(10^((-
((((X[,2])*(dp_i)^2*(1+(2*lambda/(2*({(C1*rd^C2)/(C3*rd^C4-log10(X[,1]))+rd^3}^(1/3))))*(1.257+0.4*exp(-
0.55*(2*({(C1*rd^C2)/(C3*rd^C4-
log10(X[,1]))+rd^3}^(1/3)))/lambda))))/(18*dyn.vis))*(X[,3]))/d_c)))*(2*(2*({(C1*rd^C2)/(C3*rd^C4-
log10(X[,1]))+rd^3}^(1/3))/d_c))))*(Tau_c_BY_Cd))+(1+Tau_c_BY_Tau)*((kin.vis/((((1+(2*lambda/(2*({(C1*rd^C2
)/(C3*rd^C4-log10(X[,1]))+rd^3}^(1/3))))*(1.257+0.4*exp(-0.55*(2*({(C1*rd^C2)/(C3*rd^C4-
log10(X[,1]))+rd^3}^(1/3)))/lambda)))*k_B*Temp)/(3*3.1416*dyn.vis*(2*({(C1*rd^C2)/(C3*rd^C4-
log10(X[,1]))+rd^3}^(1/3))))))^-1+10^((-3/(((((X[,2])*(2*({(C1*rd^C2)/(C3*rd^C4-
log10(X[,1]))+rd^3}^(1/3)))^2*(1+(2*lambda/(2*({(C1*rd^C2)/(C3*rd^C4-
log10(X[,1]))+rd^3}^(1/3))))*(1.257+0.4*exp(-0.55*(2*({(C1*rd^C2)/(C3*rd^C4-
log10(X[,1]))+rd^3}^(1/3)))/lambda))))/(18*dyn.vis))*(X[,3])^2)/kin.vis))))+((((X[,2])*((2*({(C1*rd^C2)/(C3*rd^
C4-log10(X[,1]))+rd^3}^(1/3)))^2*(1+(2*lambda/(2*({(C1*rd^C2)/(C3*rd^C4-
log10(X[,1]))+rd^3}^(1/3))))*(1.257+0.4*exp(-0.55*(2*({(C1*rd^C2)/(C3*rd^C4-
log10(X[,1]))+rd^3}^(1/3)))/lambda))))/(18*dyn.vis))*9.81))^-1)-
(1/(((((X[,2])*(dp_i)^2*(1+(2*lambda/(2*({(C1*rd^C2)/(C3*rd^C4-log10(X[,1]))+rd^3}^(1/3))))*(1.257+0.4*exp(-
0.55*(2*({(C1*rd^C2)/(C3*rd^C4-
log10(X[,1]))+rd^3}^(1/3)))/lambda))))/(18*dyn.vis))*9.81))))/(exp(((((1+(((((X[,2])*((2*({(C1*rd^C2)/(C3*rd^C4
-log10(X[,1]))+rd^3}^(1/3)))^2*(1+(2*lambda/(2*({(C1*rd^C2)/(C3*rd^C4-
log10(X[,1]))+rd^3}^(1/3))))*(1.257+0.4*exp(-0.55*(2*({(C1*rd^C2)/(C3*rd^C4-
log10(X[,1]))+rd^3}^(1/3)))/lambda))))/(18*dyn.vis)*9.81)^2/(X[,3])^2))^0.5)/(k*(X[,3])))*(log((z-zd)/(h_c-
zd))))/(1/(((((X[,2])*(dp_i)^2*(1+(2*lambda/(2*({(C1*rd^C2)/(C3*rd^C4-
```

```
log10(X[,1]))+rd^3}^(1/3))))*(1.257+0.4*exp(-0.55*(2*({(C1*rd^C2)/(C3*rd^C4-
log10(X[,1]))+rd^3}^(1/3)))/lambda))))/(18*dyn.vis))*9.81)))))))^-1)

N <- 100000
x1 = runif(1*N,0.1,1.0)              #RH
x2 = runif(1*N,1500,2000)           #rho
x3 = runif(1*N,0.1,0.5)             #u_star
x4 = runif(1*N,1,5)                 #U
x_1 = runif(1*N,0.1,1.0)            #RH
x_2 = runif(1*N,1500,2000)          #rho
x_3 = runif(1*N,0.1,0.5)            #u_star
x_4 = runif(1*N,1,5)                #U
Y1 = matrix(c(x1,x2,x3,x4), nrow=N)
X1 = data.frame(matrix(Y1, nrow=N))
Y2 = matrix(c(x_1,x_2,x_3,x_4), nrow=N)
X2 = data.frame(matrix(Y2, nrow=N))
a = sobol2002(model = model, X1 = X1, X2=X2, nboot = 2000, conf= 0.95);a
```